# EHRXQA: A Multi-Modal Question Answering Dataset for Electronic Health Records with Chest X-ray Images

**Seongsu Bae**[1][*], **Daeun Kyung**[1][*], **Jaehee Ryu**[1], **Eunbyeol Cho**[1], **Gyubok Lee**[1],
**Sunjun Kweon**[1], **Jeongwoo Oh**[1], **Lei Ji**[2], **Eric I-Chao Chang**[3], **Tackeun Kim**[4], **Edward Choi**[1][†]

KAIST[1]   Microsoft Research Asia[2]   Centre of Perceptual and Interactive Intelligence[3]
Seoul National University Bundang Hospital[4]
{seongsu,kyungdaeun,edwardchoi}@kaist.ac.kr[1]

## Abstract

Electronic Health Records (EHRs), which contain patients' medical histories in various multi-modal formats, often overlook the potential for joint reasoning across imaging and table modalities underexplored in current EHR Question Answering (QA) systems. In this paper, we introduce **EHRXQA**, a novel multi-modal question answering dataset combining structured EHRs and chest X-ray images. To develop our dataset, we first construct two uni-modal resources: 1) The MIMIC-CXR-VQA dataset, our newly created medical visual question answering (VQA) benchmark, specifically designed to augment the imaging modality in EHR QA, and 2) EHRSQL (MIMIC-IV), a refashioned version of a previously established table-based EHR QA dataset. By integrating these two uni-modal resources, we successfully construct a multi-modal EHR QA dataset that necessitates both uni-modal and cross-modal reasoning. To address the unique challenges of multi-modal questions within EHRs, we propose a NeuralSQL-based strategy equipped with an external VQA API. This pioneering endeavor enhances engagement with multi-modal EHR sources and we believe that our dataset can catalyze advances in real-world medical scenarios such as clinical decision-making and research. EHRXQA is available at https://github.com/baeseongsu/ehrxqa.

## 1 Introduction

Electronic Health Records (EHRs) are large-scale databases that store the entire medical history of patients, including but not limited to structured medical records (*e.g.*, diagnosis, procedure, medication), medical images (*e.g.*, chest X-ray, MRI, CT), and clinical text (*e.g.*, discharge summary, nursing note). This wealth of patient information reveals tremendous clinical knowledge about individual patients and cohorts, marking them as an indispensable resource for healthcare professionals (*e.g.*, physicians, nurses, administrators) in routine clinical practice.

Recent years have seen an upsurge in research [33, 34, 44, 46, 53] into question answering (QA) systems for EHRs. These systems are designed to effectively retrieve information from EHRs, each specializing in a different information modality within the records. For instance, table-based EHR QA systems can easily retrieve specific information from structured databases and answer questions like "Did patient 42 undergo a left heart cardiac catheterization procedure in the last hospital visit?" (see EHRSQL part in Figure 1) by executing an SQL query on the relational database. On the other hand, image-based EHR QA (*i.e.*, medical visual question answering) models are designed to handle questions related to individual medical images. For instance, given a question such as "List all

---

[*]These authors contributed equally
[†]Corresponding author

37th Conference on Neural Information Processing Systems (NeurIPS 2023) Track on Datasets and Benchmarks.

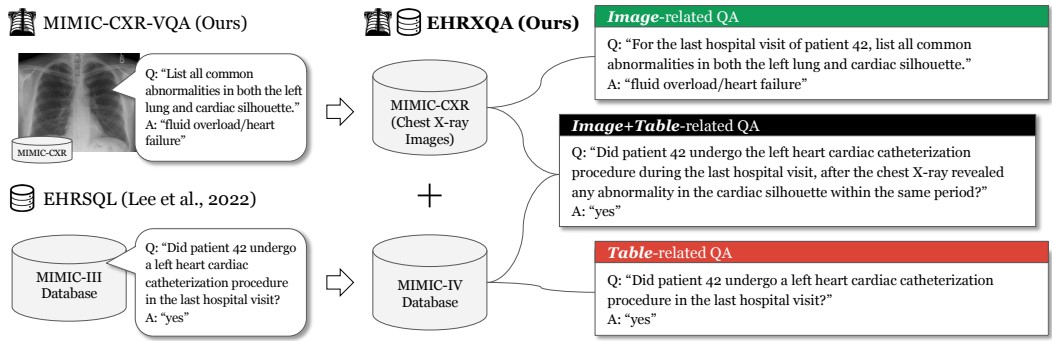

Figure 1: Our EHRXQA dataset is constructed from three uni-modal resources: MIMIC-IV for the *table* modality, MIMIC-CXR for the *image* modality, and Chest ImaGenome as a high-quality annotated version of MIMIC-CXR (not shown in the figure). Our dataset features questions for individual EHR modalities and those requiring multi-modal reasoning. It encompasses three types of QA scope: *Image-*, *Table-*, and *Image+Table*-related QA.

common abnormalities in both the left lung and cardiac silhouette." (see MIMIC-CXR-VQA part in Figure 1) along with a patient's chest radiograph, these models generate a response, thereby serving as an effective aid for radiologists. However, despite their undeniable utility, a main challenge in the current landscape of EHR QA systems lies in their focus on a single information modality, overlooking EHRs' inherently multi-modal nature. To fully utilize EHRs' potential, it is crucial to develop QA systems capable of seamlessly navigating across these multiple modalities such as "Did patient 42 undergo the left heart cardiac catheterization procedure during the last hospital visit, after the chest X-ray revealed any abnormality in the cardiac silhouette within the same period?" (see EHRXQA part in Figure 1). This capability significantly enhances our ability to build a comprehensive model of a patient's status, thereby improving the quality of the clinical decision-making process.

The progression from uni-modal to multi-modal EHR QA is a promising and self-evident step in the healthcare domain. Currently, however, only one multi-modal EHR QA dataset [4] integrates structured EHRs with clinical text. On the other hand, the integration of table modalities with imaging modalities, such as chest X-rays (CXR), remains unexplored [37]. Our research aims to bridge this gap. This has the potential to unlock significant clinical benefits, enhance cross-modal analysis, and catalyze advances in medical research.

To sum up, our contributions are threefold:

- To address the lack of publicly accessible image-based EHR QA datasets that can be combined with structured EHRs, we present MIMIC-CXR-VQA (Sec. 3.2.2). This is a complex, diverse, and large-scale visual question answering dataset in the medical domain. We not only use its questions as a basis for multi-modal EHR questions, but also exploit this dataset to benchmark existing medical VQA approaches.

- We present EHRXQA (Sec. 4), the first multi-modal EHR QA dataset for table and image modality. By leveraging uni-modal resources (*i.e.*, data sources & question templates), we integrate patients' structured databases with their aligned chest X-ray images, thereby creating a comprehensible set of QA pairs covering *Image*-related, *Table*-related, *Image+Table*-related questions.

- We propose a NeuralSQL-based approach (Sec. 5) that integrates Large Language Models (LLMs) with an external VQA application programming interface (API) to handle multi-modal questions over a structured database with images. Despite facing unique challenges of reasoning based on single or multiple images, or even a combination of images and tables, our approach effectively extracts relevant information from multi-modal EHRs in response to natural language queries.

## 2 Related Work

**Image-based EHR Question Answering**  Image-based EHR QA [23–25, 31] is a distinct subset of medical visual question answering (VQA) [1, 5, 21, 22, 32, 38], given that it focuses on answering questions related to a specific patient's single medical image, primarily within the radiography domain. Despite intriguing research directions in existing datasets such as patient-centric QA [24] or dialogue [31], there remains a noticeable gap in efforts to view patient images as an integral part of the EHR database or to synchronize them effectively with the structured tabular data in EHRs.

Presently, MIMIC-CXR [28] is the only publicly available imaging resource that links to patient IDs (*i.e.*, subject IDs) in the MIMIC-IV database [27], offering a comprehensive perspective on EHRs. Although there exist two medical VQA datasets [23, 31] based on MIMIC-CXR, neither is publicly available. Moreover, their question templates are less complex (*i.e.*, they lack complex set operations or logical operations) and are largely encompassed by our question template scope.[3]

**Table-based EHR Question Answering**   Table-based EHR QA [3, 17, 33, 46, 49, 51, 53] focuses on extracting structured information from a hospital's relational database. The task is typically approached through semantic parsing [6], where natural language utterances are translated into either a query language [35, 55, 57] or domain-specific logical forms [46, 49]. Wang *et al.* [53] introduced MIMICSQL dataset for the text-to-SQL generation task on MIMIC-III, employing slot-filling for pre-defined templates and using crowd-sourced paraphrasing. Pampari *et al.* [44] constructed emrKBQA dataset, a large-scale text-to-logical form dataset tailored for patient-specific QA on MIMIC-III, drawing from the logical forms identified in emrQA [44]. Recently, Lee *et al.* [33] introduced a novel text-to-SQL dataset, EHRSQL, associated with both MIMIC-III and eICU [45]. This dataset presents unique challenges, including time-sensitive questions and unanswerable queries.

**Question Answering over Multi-Modal Knowledge Sources**   Recent research [10, 12, 14, 16, 39, 48, 50, 52, 56] has delved into generating responses to queries using multi-modal knowledge sources. However, the major challenge when dealing with multi-modal databases, such as EHRs [4], is integrating rich unstructured data (*e.g.*, image, text) into a structured database (*e.g.*, table) and effectively leveraging this information within the QA system. Urban *et al.* [52] introduced MMDBs, a new category of database systems, which allow seamless querying of text and tables using SQL. Similarly, Chen *et al.* [14] proposed Symphony, a QA system for multi-modal data lakes, particularly designed to handle text and tables by using a unified representation for multi-modal datasets. Drawing inspiration from recent studies like Binder [15], a training-free neural-symbolic framework that uses GPT-3 Codex [11] to map task inputs to programs, our research broadens the SQL syntax to create a QA system specifically intended for image processing within the database.

## 3   Preliminary: Ingredients for Multi-Modal EHR QA

### 3.1   Uni-Modal Data Resources

To construct a comprehensive EHR database that integrates both *table* and *image* modalities, we need uni-modal resources that meet our criteria: (i) publicly accessible; (ii) presence of common patients across datasets; (iii) contain high-quality image annotations. After careful consideration, we strategically select three datasets: MIMIC-IV [27] for *table* modality, MIMIC-CXR [28] for *image* modality, and Chest ImaGenome [54] as a high-quality annotated version of MIMIC-CXR. Note that all datasets share a significant number of patient IDs (19,264), while incompatible patient IDs exist due to the varying data collection periods. We briefly introduce each of the source datasets.[4]

- **MIMIC-IV (v2.2)** [27] is a large, freely accessible relational database of deidentified health-related data (*e.g.*, diagnoses, procedures, and treatments) associated with 50,920 patients who stayed in critical care units of Beth Israel Deaconess Medical Center (BIDMC) between 2008-2019.

- **MIMIC-CXR (v2.0.0)** [28] is a large-scale publicly available dataset of 377,110 chest radiographs associated with 227,827 imaging studies sourced from the BIDMC between 2011-2016. MIMIC-CXR can be linked to MIMIC-IV using lookup tables that connect patient identifiers.

- **Chest ImaGenome (v1.0.0)** [54], organized with scene graphs for 242,072 frontal images sourced from MIMIC-CXR, illustrates the relationships between anatomical locations and their corresponding attributes within each image. This dataset comprises two primary subsets: the *silver*[5] dataset with automatically generated scene graphs for each chest X-ray image, and the *gold* dataset containing a subset that has been manually validated and corrected by clinicians, serving as a reliable held-out set for research derived from 500 unique patients.

---

[3]In comparison: Hu *et al.* [23] provides around 15 templates across 6 types, Kovaleva *et al.* [31] offers 1 template across 1 type, while our dataset presents 48 templates across 7 types.

[4]All three datasets are publicly accessible through the PhysioNet platform (https://physionet.org/), with users required to request and obtain credentialed access under its established procedure.

[5]For the *silver* dataset, given the high inter-annotator agreement score (0.984 for 500 reports) [54], the reliability is strongly suggested. This score substantiates the decision to use the *silver* dataset for building our MIMIC-CXR-VQA and EHRXQA, providing confidence in the accuracy and quality of the derived information.

## 3.2 Uni-Modal EHR QA datasets

We aim to build a multi-modal EHR QA dataset featuring questions for each modality individually, as well as those that require cross-modal reasoning. To achieve this, we utilize uni-modal QA datasets based on MIMIC nature. For *table* modality, we take the existing questions templates from EHRSQL [33], and adapt them to MIMIC-IV. For *image* modality, to address the lack of diverse question templates and the absence of accessible VQA datasets based on MIMIC-CXR, we craft our templates and further construct a medical VQA dataset called **MIMIC-CXR-VQA** (Sec. 3.2.2).

### 3.2.1 Table-based EHR QA: EHRSQL

EHRSQL [33] is a text-to-SQL dataset curated for structured EHRs, assembled from the responses of various hospital staff. EHRSQL provides (Question, SQL) samples for two publicly accessible EHR datasets, namely MIMIC-III [29] and eICU [45], and samples consist of both *answerable* and *unanswerable* questions. Since our research scope primarily focuses on building a multi-modal QA dataset, we have selected only the *answerable* question templates from EHRSQL for MIMIC-III. These templates were converted to align with our MIMIC-IV setting, while maintaining their comprehensive template schema, including multiple value slots (*e.g.*, operation and condition value slots) and time filter slots. For more details about the conversion process of question templates from MIMIC-III to MIMIC-IV, please refer to Appendix B.2.1.

### 3.2.2 Image-based EHR QA: MIMIC-CXR-VQA

**Data Preprocessing** We use MIMIC-CXR [28] as our image source and Chest ImaGenome [54] for label information. In MIMIC-CXR, each patient can have multiple studies arranged in chronological order, and each study can contain multiple CXR images. From each study, we select one representative frontal view (*i.e.*, AP, PA) image. We then assign labels to these images derived from the Chest ImaGenome *silver*/*gold* datasets. As a result, each CXR image features 563 distinct *relations* among 36 *objects*, each linked to several attributes from a pool of 68 *attributes* (across 5 *categories*[6]). As illustrated in Figure 2, each *relation* indicates the presence (1) or absence (0) of an *attribute* (*e.g.*, lung cancer) within a *category* (*e.g.*, disease), linked to an *object* (*e.g.*, left lung). For data splitting, we use the machine-generated *silver* label dataset for training and validation, with a 95:5 split, while the human-labeled *gold* dataset serves as the testing dataset. For more details of data preprocessing, please refer to Appendix B.2.2.

Figure 2: Upper: Scene graphs of multiple CXR studies derived from the Chest ImaGenome. Lower: Our processed CXR features, obtained from these scene graphs. Due to spatial constraints, only a subset of the original Chest ImaGenome labels is displayed.

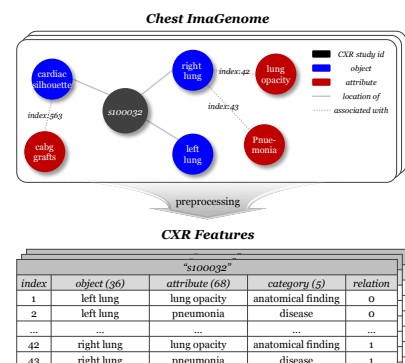

**Question Template Construction** We started by analyzing existing medical VQA datasets [1, 5, 21, 22, 32, 38] and templatized their questions to match our preprocessed data schema (*i.e.*, object, attribute, category), thus handcrafting our initial seed templates. We drew inspiration from general VQA datasets [2, 19, 26, 30], enhancing these seed templates using logical and set operations to create a more diverse and complex set of question templates. We further incorporated clinically relevant factors [32] into our templates, such as the patient's gender, CXR view position, and size-related features (*i.e.*, width ratio between two anatomical locations). As a result, we defined a total of 48 templates, all of which were evaluated by a medical expert for clinical importance. For more details about template construction including a list of our templates, please refer to Appendix B.2.2.

**VQA dataset generation** We generated our VQA dataset by sampling (image *I*, question *Q*, answer *A*) triples. For example, consider the template "Is there ${attribute} in the ${object}?". We filled this template using sampled arguments (*e.g.*, ${object}='left lung', ${attribute}='lung cancer'), which led to the creation of the question *Q*: "Is there lung cancer in the left lung?". Next, we sampled an image

---

[6]The 5 categories include 'anatomical finding', 'disease', 'device', 'tubes/lines', 'technical assessment'.

Table 1: Sample questions in EHRXQA, categorized by **modality-based** (*Image*, *Table*, *Image+Table*) and **patient-based** scope (*none*, *single*, *group*), illustrating our dataset's diversity and complexity.

| modality-based | patient-based | | Sample question |
|---|---|---|---|
| *Image* | *single* | 1-*image* | Given the last study of patient 15439, which anatomical finding is associated with the right lower lung zone, pneumothorax or vascular redistribution? |
| | | 2-*image* | Enumerate all diseases that are newly detected based on the last study of patient 19290 in 2103 compared to the previous study. |
| | | N-*image* | How many times has the chest X-ray of patient 18489 shown linear/patchy atelectasis in the left lung on the current hospital visit? |
| | *group* | | Count the number of patients whose chest X-ray studies this year showed any abnormalities in the mediastinum. |
| *Table* | *none* | | What's the cost of a drug named lopinavir-ritonavir? |
| | *single* | | Did patient 16164 receive any magnesium lab tests last year? |
| | *group* | | What was the top three diagnosis that had the highest two year mortality rate? |
| *Image+Table* | *single* | | Did a chest X-ray study for patient 15110 reveal any anatomical findings within 2 month after the prescription of hydralazine since 2102? |
| | *group* | | Provide the ids of patients in the 20s whose chest X-ray showed low lung volumes in the right lung this month. |

$I$ and executed a predefined program[7] to generate an answer $A$. To enrich linguistic diversity while preserving focus on the medical domain [41], we devised a paraphrasing strategy (an average of 16.5 paraphrases for each template) using carefully designed prompts based on GPT-4 [43]. Finally, we present **MIMIC-CXR-VQA**, a dataset composed of 377,391 unique $(I, Q, A)$ triples across seven content types[8]. For a deeper dive into the statistics of MIMIC-CXR-VQA and its comparisons to other medical VQA datasets, please refer to Appendix B.2.2.

## 4 EHRXQA: A Multi-Modal EHR Question Answering Dataset

### 4.1 Dataset Construction

In this section, we outline the construction process for the EHRXQA dataset. We begin by integrating CXR images from MIMIC-CXR and tables from MIMIC-IV into our EHRXQA database (see Sec. 4.1.1). Next, we detail the creation of question templates (see Sec. 4.1.2), and the incorporation of the corresponding SQL/NeuralSQL annotations (see Sec. 4.1.3). Finally, we discuss our systematic data generation process (see Sec. 4.1.4) employed to build our EHRXQA dataset.

### 4.1.1 Database Construction

**CXR Integration into MIMIC-IV** To cross-reference CXR images with structured EHRs (*e.g.*, to find CXR images of patients who have been prescribed a specific drug), an integrated database system is crucial. To achieve this, we developed an image reference table named `TB_CXR`. This table comprises six columns: `subject_id`, `hadm_id`, `study_id`, `image_id`, `studydatetime`, and `viewposition`, connecting patient-related identifiers with CXR images of MIMIC-CXR. Through this table, patient CXR images can be retrieved alongside other table data (*e.g.*, diagnosis, procedure, and prescriptions) from MIMIC-IV using the `subject_id` or `hadm_id`. For more details on the database construction process, please refer to Appendix C.1.

**Timeframe Adjustment** We condensed the event times in each patient's records, which originally spanned from 2100 to 2200 due to the de-identification process in MIMIC-IV [27], to a more realistic timeframe (2100-2105). This adjustment was performed while preserving the integrity of CXR images and individual medical event timelines. To enable relative time expressions like 'last year', we set '2105-12-31 23:59:00' as the *current time* and excluded any records beyond this point. We consider patients without hospital discharge times, due to this exclusion, as currently admitted.

**Building Silver/Gold Databases** The Chest ImaGenome [54] dataset includes two types of cohorts based on image information: *silver* (*i.e.*, machine-generated) and *gold* (*i.e.*, human-labeled). We selected subsets of patients from each cohort to create two distinct databases: the *silver* database, comprising 800 patients, and the *gold* database, comprising 400 patients. These databases are utilized for different purposes: the *silver* database is used for training and validating the QA dataset, while the *gold* database is used for testing the QA dataset.

---

[7]For each template, we define a program to produce an answer $A$ using the given question $Q$ and relationship information from the preprocessed data (see Sec. 3.2.2) of the image $I$.

[8]Questions are divided into 7 categories based on the content of the question: 'presence', 'anatomy', 'attribute', 'abnormality', 'size', 'plane', 'gender'.

### 4.1.2 Question Template Construction

We define the scope of our question templates using two key criteria: **modality-based** and **patient-based** scopes. The **modality-based** scope classifies templates into three categories, *Image*-related, *Table*-related, and *Image+Table*-related, depending on the type of data modality they require. The **patient-based** scope classifies templates according to whether they relate to a *single* patient, a *group* of patients, or *none* (*i.e.*, do not relate to specific patients). To accommodate these scopes with diverse and comprehensive question templates, we employ existing uni-modal question resources discussed in Sec. 3.2: MIMIC-CXR-VQA for *image* modality and EHRSQL for *table* modality. Examples of our *modality*- and *patient*-based question templates, which illustrate the diversity and complexity of EHRXQA dataset, can be found in Table 1.

Recognizing the critical role of time expressions in real-world questions in the hospital workplace [33], we further refined our question templates. We adopted the time filter concept from EHRSQL and applied it to all question templates. This enhancement allows our question templates to better meet the specific needs in clinical practice. Note that these time filters can be categorized into three types: 1) [time_filter_global] restricts the time range of interest, such as 'last year' or 'in 2022'; 2) [time_filter_within], incorporating the keyword 'within', pinpoints events happening within specific temporal boundaries, such as 'within the same hospital visit' or 'within the same day'; 3) [time_filter_exact] refers to a precise temporal point, such as the 'last CXR study' or a specific date and time like '2105-12-26 15:00:00'.

Our template construction process included 1) clinical needs across both image and table modalities via consulting a medical expert, 2) grounding our templates in these needs for both CXR images and EHR tables, and 3) ensuring clinical relevance. Note that the entire process of designing templates was validated by a board-certified medical expert from the department of neurosurgery to ensure clinical utility. For a full list or an in-depth discussion on template construction strategy, please refer to Appendix C.2. The following details how we tailored question templates for each modality.

***Image*-related** Questions related to *image* modality can be defined as inquiries requiring pixel-level information from CXR images retrieved from EHR, which can aid in analyzing visual diagnoses for individual or cohort patient conditions in real-world medical scenarios. To cater to these queries, we used the 48 MIMIC-CXR-VQA templates (*e.g.*, "List all diseases.") and integrated with expressions to specify our target images (*e.g.*, "The last study of patient 42"). This integration (*e.g.*, "Given the last study of patient 42, list all diseases.") enables retrieval of CXR images from the EHR and subsequent analysis based on natural language requests. We further enhanced the templates focusing on a *single* patient to include queries that compare two consecutive CXR studies (*e.g.*, "Given the last study of patient 42, are there any newly detected diseases compared to the previous study?") or multiple studies (*e.g.*, "Has patient 42 had any chest X-ray study indicating any anatomical findings in 2023?") from the same patient. This process resulted in 168 templates for the *image* modality.

***Table*-related** The *table* modality, a significant part of EHRs, covers questions primarily requiring structured information from EHR tables. These questions relate to patient demographics, diagnoses, procedures, medications, and other clinical details typically recorded in structured EHR formats. EHRSQL, which offers a wealth of questions seeking information from EHR tables, proves to be an invaluable resource in this context. Considering the substantial overlap between the MIMIC-III and MIMIC-IV schemas, we leveraged the question templates from EHRSQL's MIMIC-III templates, adapting them appropriately to fit the MIMIC-IV schema with minimal modifications. This process resulted in 174 templates for the *table* modality.

***Image+Table*-related** In the *image+table* modality, all templates are designed to require multi-modal information from both CXR images and structured data from EHRs. We leveraged both MIMIC-CXR-VQA and EHRSQL templates to build multi-modal question templates. Since we recognize the essential role of temporal analysis in multi-modal medical events, we designed templates to capture three primary scenarios: 1) Co-occurring table and CXR events. (*e.g.*, "On the same visit, did patient 42 receive nitroglycerin and have a CXR showing any abnormality in the cardiac silhouette?"); 2) A CXR event following a table event. (*e.g.*, "After being prescribed nitroglycerin, did patient 42 have a CXR during the same visit revealing any abnormality in the cardiac silhouette?") 3) A table event following a CXR event. (*e.g.*, "Was patient 42 prescribed nitroglycerin during the same visit after a CXR showed cardiac silhouette abnormalities?"). These templates allow for comprehensive analysis of combined events, the cause-and-effect relationships in CXR diagnosis,

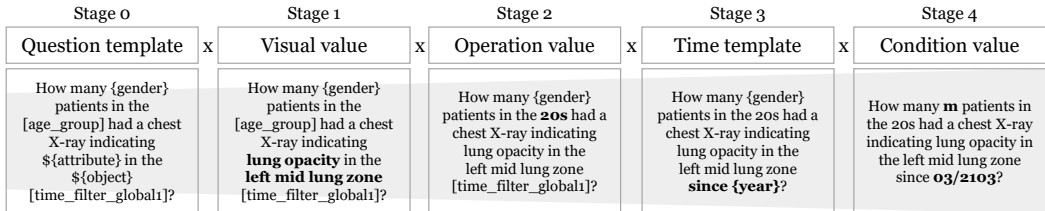

Figure 3: QA data generation process

Table 3: A comparison of EHRXQA with other EHR QA datasets based on the MIMIC database.

| Data source | | Image | Table | Text | Patient scope | # of tables / DB | # of questions | compositional |
|---|---|---|---|---|---|---|---|---|
| Mimic-VQA [23] | MIMIC-CXR | ✓ | - | - | single | - | 297,723 | - |
| MIMIC-CXR-VQA (ours) | MIMIC-CXR, Chest ImaGenome | ✓ | - | - | single | - | 377,726 | ✓ |
| MIMICSQL [53] | MIMIC-III | - | ✓ | - | none, single, group | 5 | 10,000 | ✓ |
| EHRSQL [33] | MIMIC-III, eICU | - | ✓ | - | none, single, group | 13.5 | 24,411 | ✓ |
| DrugEHRQA [4] | MIMIC-III | - | ✓ | ✓ | single | 3 | 70,381 | - |
| EHRXQA (ours) | MIMIC-IV, MIMIC-CXR, Chest ImaGenome | ✓ | ✓ | - | none, single, group | 18 | 46,152 | ✓ |

and relevant follow-up measures related to the CXR diagnosis. To eliminate confusion arising from overlapping information between the CXR and diagnoses/procedures tables, we ensure that questions explicitly specify when a 'CXR study' is necessary. This led to 75 templates for the *image+table* modality, enabling simulations across diverse scenarios.

### 4.1.3 SQL/NeuralSQL Annotation

Standard SQL queries are effective for retrieving structured data from EHRs [53, 33], such as demographic information or lab results stored in tables. However, they are not designed to handle unstructured data, such as CXR images, which also contain valuable patient information. This limitation prevents us from using SQL to retrieve answers for complex, multi-modal questions that span both structured and unstructured data. To overcome this limitation, we adopt NeuralSQL, which is inspired by the Binder approach [15]. NeuralSQL acts as an executable representation, extending SQL's capabilities to process unstructured image data. NeuralSQL utilizes a pretrained neural model to extract features from medical images, turning them into a structured format suitable for SQL queries. For more details about our NeuralSQL-based strategy, please refer to Sec. 5.

For *Table*-related question templates, we utilize the SQL annotations provided by EHRSQL and modify them to be compatible with the MIMIC-IV schema. For question templates related to *Image* or *Image+Table*, we annotate them using NeuralSQL representation. The entire SQL/NeuralSQL annotation process was manually undertaken by four graduate students over a span of two months, involving iterative revisions. During this process, the students transformed question templates into their corresponding SQL or NeuralSQL formats.

### 4.1.4 Data Generation

The question generation process, illustrated in Figure 3, begins with choosing a template at Stage 0, followed by a four-step systematic process (Stages 1-4) that specifies semantics of the template. These steps involve the sampling of *visual value* (Stage 1), *operation value* (Stage 2), *time template* (Stage 3), and *condition value* (Stage 4). In Stage 1, we augment the question with *visual values* by filling in object, attribute, and category slots (described in Sec. 3.2.2), tailored specifically for CXR images. Stage 2 involves sampling *operation values* (*e.g.*, 20s) from a predefined set of options such as [age_group] = (20s, 30s, 40s, 50s, 60 or above), which are independent of the database schema or records. Stage 3 incorporates *time templates*, translated into natural language expressions to establish a temporal context within the questions. Lastly, Stage 4 incorporates *condition value* sampling, filling placeholders such as {gender} and {year} to provide context-specific conditions to the question.

The corresponding SQL/NeuralSQL query also contains these slots, filled with the same values during the question creation process, thereby completing the (Question, SQL/NeuralSQL) pair. These (Question, SQL/NeuralSQL) pairs are only added to the data pool if the sampled SQL/NeuralSQL query yields a valid answer when executed. To enhance linguistic diversity, we use GPT-4 to paraphrase each question. These paraphrases are then manually reviewed by our team to ensure quality. Further details can be found in the Appendix C.3.

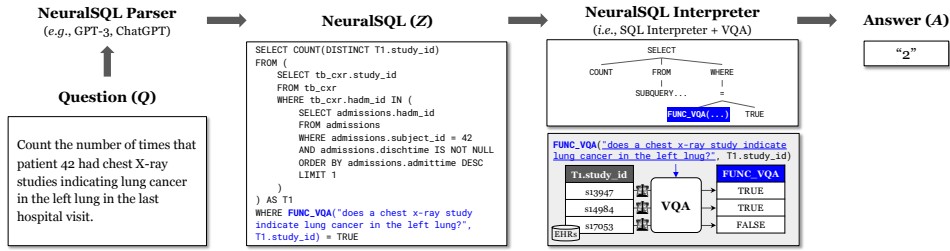

Figure 4: Overview of our NeuralSQL-based Approach.

## 4.2 Data Statistics and Comparisons with other EHR QA datasets

EHRXQA consists of a total of 46,152 samples including 16,366 *image*-related samples, 16,529 *table*-related samples, and 13,257 samples involving both *images* and *tables*. Overall Statistics are summarized in Table 2. For a comprehensive breakdown of the dataset's distribution across various modalities and patient scopes, please refer to Appendix C.4.

Table 2: Overall statistics of EHRXQA including the number of samples for each modality.

|  | train | valid | test |
|---|---|---|---|
| *image*-related QA | 12,860 | 1,838 | 1,668 |
| *table*-related QA | 12,961 | 1,852 | 1,716 |
| *image+table*-related QA | 10,353 | 1,480 | 1,424 |
| total # of samples | 36,174 | 5,170 | 4,808 |

Table 3 provides a comparison of EHRXQA with other EHR QA datasets based on the MIMIC database. Compared to other image-based EHR QA datasets (rows 1-2), EHRXQA incorporates information from EHR tables. This allows for more complex queries about images, such as comparing the clinical condition of two specific images. This feature extends beyond the existing VQA scope and aims to maximize EHR data utilization. Compared with table-based EHR QA datasets (rows 3-5), EHRXQA shows the most complex data structure, featuring up to 18 tables per database and a comprehensive range of patients. These features broaden the spectrum of potential questions that can be posed. To the best of our knowledge, EHRXQA is the first attempt to merge image and tabular modalities in medical QA.

## 5 NeuralSQL with Visual Question Answering

EHRXQA presents three unique challenges for EHR QA systems that handle both image and table modalities: 1) retrieving and analyzing a single image from the database solely based on natural language expressions; 2) handling multiple images, which include comparative queries across multiple studies; 3) and reasoning across multi-modal data over tables and images. To overcome these challenges, we introduce a NeuralSQL-based approach, inspired by the Binder [15] framework. Our approach integrates a large language model (LLM)-based parser with an external VQA API module, effectively handling both structured information and images. As depicted in Figure 4, the NeuralSQL-based approach consists of two stages:

1. **NeuralSQL Parsing**: Given a database $D$ and question $Q$, the parser model translates the question $Q$ to an executable NeuralSQL query $Z$. Note that for all *Image*-related and *Image+Table*-related questions, we annotated the corresponding NeuralSQL query, as discussed in Sec. 4.1.3. This query features a specific VQA API call function (FUNC_VQA), which handles image-related queries by calling an external VQA model. This API function requires two arguments: (1) a subquestion, $q_I$, which seeks information related to the image, and (2) the relevant image identifier, $c_I$, linking to the study_id column in TB_CXR.

2. **NeuralSQL Execution**: This execution stage involves parsing the NeuralSQL query into an abstract syntax tree (AST), guided by the extended grammar. During this process, the interpreter executes the parsed tree in sequence, including any API calls. Upon encountering a VQA API call, the interpreter employs an internal image loader for the corresponding image(s) $I$ based on $c_I$. These image(s) are then fed into the VQA model, which infers the information based on the provided question $q_I$ and image(s) $I$. The output of the API call is preserved as a column data object, making it compatible with the standard SQL grammar. This allows the NeuralSQL interpreter to execute the program seamlessly and derive the final answer $A$.

## 6 Experiments

In this section, we evaluate medical visual question answering methods on our MIMIC-CXR-VQA dataset (Sec. 6.1). Subsequently, we use the best-performing model as an external VQA API for benchmarking our EHRXQA dataset (Sec. 6.2).

## 6.1 MIMIC-CXR-VQA

**Task & Evaluation** We define the VQA task as a multi-label classification with 110 distinct answer labels. This includes 36 objects, 68 attributes, and 4 extras (*i.e.*, 'M', 'F', 'AP', 'PA'), as well as 'yes' and 'no' responses.

In MIMIC-CXR-VQA, *verify* questions (*i.e.*, "Is there ${attribute} in ${object}?") test a model's basic perception, while other questions demand a logical combination of corresponding percep-

Table 4: Performance of five baselines on MIMIC-CXR-VQA. To ensure a fair comparison, we pre-trained VLP models (indicated by ∗) using the same corpus.

| Model | Valid set | | Test set | | |
|---|---|---|---|---|---|
| | Acc | F1 (micro) | Acc | F1 (micro) | $AUC_{rel}$ |
| Prior (Most) [2] | 26.8 | 0.27 | 25.4 | 0.25 | - |
| Prior (Question) [2] | 34.3 | 0.34 | 32.4 | 0.32 | - |
| PubMedCLIP [18] | $55.1 \pm 1.7$ | $0.56 \pm 0.02$ | $54.9 \pm 1.3$ | $0.54 \pm 0.02$ | $0.82 \pm 0.09$ |
| PubMedCLIP* | $56.6 \pm 1.9$ | $0.58 \pm 0.02$ | $56.5 \pm 2.1$ | $0.56 \pm 0.02$ | $0.83 \pm 0.09$ |
| MedViLL* [40] | $64.7 \pm 0.2$ | $0.69 \pm 0.00$ | $63.6 \pm 0.1$ | $0.67 \pm 0.00$ | $0.98 \pm 0.08$ |
| M³AE [13] | $68.9 \pm 0.2$ | $0.73 \pm 0.00$ | $68.9 \pm 0.3$ | $0.72 \pm 0.00$ | $1.02 \pm 0.08$ |
| **M³AE*** | $70.2 \pm 0.1$ | $0.74 \pm 0.00$ | $69.2 \pm 0.4$ | $0.73 \pm 0.00$ | $1.05 \pm 0.09$ |

tion abilities. Therefore, both perception and logical combination are necessary to solve our QA dataset. However, unlike logical operations with clear answers, even radiologists cannot achieve perfect perception accuracy in CXRs [7, 8]. Thus, it is very likely that the upper bound QA performance of MIMIC-CXR-VQA is lower than 100%. We thus aim to estimate the highest achievable perception accuracy for single-image *verify* questions as a reference score. To simplify the problem, we design a reference model as a classification model that can answer our basic *verify* questions. We propose the performance of this model as a reference score for perception performance and introduce a new metric by comparing this reference score with the performance of the VQA model. For each object-attribute pair $(o, a)$, $m_{rel}(o, a) = \frac{m_{VQA}(o,a)}{m_{ref}(o,a)}$ where $o$ and $a$ denote a specific object and attribute. $m_{VQA}$ and $m_{ref}$ denote the metric scores of the VQA model and the reference model, and $m_{rel}$ is our proposed relative metric. We use Area Under the Receiver Operating Characteristic (AUROC) as our measure $m$ (denote as $AUROC_{rel}$). We provide a comprehensive evaluation of the model, not only our relative score, but also standard metrics like accuracy and F1 score. For further details on the reference model, please refer to Appendix E.1.

**VQA Baselines** We evaluate five VQA baselines: two prior models [2], PubMedCLIP [18], MedViLL [40], and M³AE [13]. Prior (Most) or Prior (Question) returns the most probable answer estimated from the entire training set or the corresponding question. PubMedCLIP, MedViLL, and M³AE are vision-language pre-training (VLP) models, each leveraging unique pre-training objectives and architectures. To ensure a fair comparison, we pre-trained all models on the same MIMIC-CXR (image, report) pre-training corpus, with those models denoted by an asterisk (∗). For more details, please refer to Appendix E.1.

**Results and Findings** Table 4 presents the baseline results on MIMIC-CXR-VQA dataset. The model Prior (Question), which depends solely on language, yields an accuracy of around 30%. This result attests to the reduced language bias in our dataset, emphasizing the importance of multi-modal reasoning. Among the models evaluated, M³AE achieves the best performance, likely due to its more fine-grained pre-training objectives compared to PubMedCLIP and MedViLL.

## 6.2 EHRXQA

**Task** We use semantic parsing to bridge natural language and machine-executable language. The *Image*-related and *Image+Table*-related QA scopes are formulated as a Text-to-NeuralSQL task, facilitating complex queries across images and tables. The *Table*-related QA scope, focusing solely on tabular data, is tackled as a Text-to-SQL task.

**Evaluation** We employ three metrics to assess the effectiveness of the parsing and execution stages described in Sec. 5, as well as the overall performance of the QA system: 1) *Logical Form Accuracy* ($Acc_{LF}$) evaluates the performance of the parsing stage ($Q \rightarrow Z$). It computes the accuracy by performing an exact match comparison between the logical form of the predicted program $\hat{Z}$ and that of the ground truth program $Z$; 2) *Ground-truth Execution Accuracy* ($Acc_{EX|gt}$) assesses the accuracy of the execution stage ($Z \rightarrow A$) by comparing the result of the ground truth program $Z$ with the ground truth answer $A$. For *Table*-related QA in EHRXQA, this metric yields 100% accuracy. For *Image*-related QA and *Image+Table*-related QA, this equates to measuring the VQA performance; 3) *Prediction Execution Accuracy* ($Acc_{EX|pred}$) evaluates the accuracy of execution with the predicted program $\hat{Z}$, providing an assessment of the overall system performance, including both parsing and execution stages.

Table 5: Comparison of ChatGPT (`gpt-3.5-turbo-0613`) with M$^3$AE model on EHRXQA dataset using two different prompting strategies for *Image*-, *Table*-, and *Image+Table*-related QA.

| Model | Prompt | *Image*-related | | | *Table*-related | | | *Image+Table*-related | | |
|---|---|---|---|---|---|---|---|---|---|---|
| | | $Acc_{LF}$ | $Acc_{EX\|gt}$ | $Acc_{EX\|pred}$ | $Acc_{LF}$ | $Acc_{EX\|gt}$ | $Acc_{EX\|pred}$ | $Acc_{LF}$ | $Acc_{EX\|gt}$ | $Acc_{EX\|pred}$ |
| ChatGPT | Fixed | 1.1 | 49.4 | 17.4 | 4.9 | 100.0 | 30.0 | 4.8 | 68.8 | 35.7 |
| + M$^3$AE | BM25 (train) | 87.3 | 49.4 | 48.2 | 73.0 | 100.0 | 92.9 | 72.5 | 68.8 | 65.9 |

**Baselines**  We build a strong QA baseline by combining ChatGPT [42] and M$^3$AE [13], which are outperforming models in the fields of semantic parsing (*e.g.*, Text-to-Query) and medical VQA (*e.g.*, MIMIC-CXR-VQA), respectively. For ChatGPT, we conduct in-context learning [9] (ICL) through two different prompt strategies: 1) Fixed: using fixed `N`-shot (Question, Query) pairs; 2) BM25 (train) [47]: retrieving `N` relevant (Question, Query) pairs from the training QA dataset for a given question. These retrieved pairs are then used as few-shot examples. Here, we use `N` as 10. For M$^3$AE, we first train it on our MIMIC-CXR-VQA and then deploy it as our external VQA API, integrated within NeuralSQL. For more detailed implementations, please refer to Appendix E.3.

**Results and Findings**  Table 5 shows the performance of EHRXQA, with three metrics for each *modality*-based scope. The first row of the table shows the performance when using a fixed prompt for all questions, while the second row shows the performance when given a different prompt for each question using BM25. As shown in the Table 5, giving relevant few-shot examples using BM25 significantly boosts performance. In the case of *Table*-related questions, our model achieves 92.9% $Acc_{EX|pred}$ score with 73.0% $Acc_{LF}$ score. However, when it comes to the remaining questions that rely on image information, our model demonstrates a relatively low performance, even though it maintains a high $Acc_{LF}$ score. Specifically, for *Image*-related questions, the $Acc_{LF}$ is 87.3% as compared to the $Acc_{EX|pred}$ of 48.2%. For *Image+Table*-related questions, the model achieves an $Acc_{LF}$ of 72.5%, while the $Acc_{EX|pred}$ is 65.9%.

Notably, the model's performance at the execution stage ($Acc_{EX|pred}$) is affected by the number of images that the model (*i.e.*, VQA model) needs to process. For example, in the context of *Image*-related QA, we observed that the $Acc_{EX|pred}$ drops to 39.6% when the model has to process multiple images (*i.e.*, (Image, single, N-image) scope described in Table 1) within a *single* patient QA scope. The situation worsens in a *group* QA scope where the model faces the challenge of accurately predicting a large number of image results, leading to an $Acc_{EX|pred}$ of 1.7%. This observed trend contributes to the relatively reduced performance for *Image*-related (48.2%) and *Image+Table*-related questions (65.9%), even when considering the model's peak overall test set performance (69.2%) as detailed in Table 4.

This trend also explains the model showing superior performance ($Acc_{EX|pred}$) on *Image+Table*-related questions (65.9%) than on *Image*-related questions (48.2%). Given the complex conditions present in *Image+Table*-related questions, the scope of images becomes more specified. This leads to a lower number of images to process in comparison to *Image*-related scenarios, resulting in a relatively higher performance for these multi-modal queries. Overall, the huge gap between $Acc_{LF}$ and $Acc_{EX|pred}$ suggests visual perception could be a bigger roadblock to AI models being deployed in clinical practice than logical reasoning, and future research should put as much emphasis on perception as complex logical reasoning.

## 7   Discussion

**Limitations**  Though we have carefully designed the dataset, several limitations exist: 1) Since our dataset is based on the MIMIC database, it potentially limits its generalizability. 2) Due to the constrained label scope of Chest Imagenome, our dataset lacks the capability to address more detailed visual questions, such as identifying specific tumor sizes from chest X-rays. 3) Unlike EHRSQL, our model does not include unanswerable questions, an aspect that, if addressed, could enhance our model's comprehensiveness and applicability. Future work should aim to address these constraints.
**Future Direction**  Our study signifies a substantial step forward in multi-modal EHR QA systems, but notable potential for refinement remains. Key future directions include: 1) Enlarging the scope of our dataset by enhancing the multi-modal dialogue system [36]; 2) Incorporating mechanisms to address unanswerable questions or ambiguous images, which is crucial for real-world applications [33]; and 3) Broadening our modality by evolving our dataset to support tri-modal question answering [20, 50]. These forward-looking endeavors will leverage our dataset as a valuable resource, laying the groundwork for more comprehensive and practical healthcare solutions.

## Acknowledgments and Disclosure of Funding

We are grateful to Jiho Kim, Jiyoung Lee, Youngjune Lee, JongHak Moon, Hyunseung Chung, and Seungho Kim for their fruitful comments and inspiration. We would like to thank three anonymous reviewers for their time and insightful comments. This work was (partially) supported by Microsoft Research Asia, Institute of Information & Communications Technology Planning & Evaluation (IITP) grant (No.2019-0-00075, RS-2022-00155958), National Research Foundation of Korea (NRF) grant (NRF-2020H1D3A2A03100945), and the Korea Health Industry Development Institute (KHIDI) grant (No.HR21C0198), funded by the Korea government (MSIT, MOHW).

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
