# Supplementary Contents

# A  Datasheet for Datasets

## A.1  Motivation

- **For what purpose was the dataset created?**

  We created EHRXQA to provide a valuable resource for advancing machine learning applications in multi-modal question answering systems on structured electronic health records (EHRs) and chest X-ray images. As an affiliated dataset, we created MIMIC-CXR-VQA to provide a benchmark for medical visual question answering systems.

- **Who created the dataset (e.g., which team, research group) and on behalf of which entity (e.g., company, institution, organization)?**

  The authors of this paper.

- **Who funded the creation of the dataset? If there is an associated grant, please provide the name of the grantor and the grant name and number.**

  This work was (partially) supported by Microsoft Research Asia, Institute of Information & Communications Technology Planning & Evaluation (IITP) grant (No.2019-0-00075, RS-2022-00155958), National Research Foundation of Korea (NRF) grant (NRF-2020H1D3A2A03100945), and the Korea Health Industry Development Institute (KHIDI) grant (No.HR21C0198), funded by the Korea government (MSIT, MOHW).

## A.2  Composition

- **What do the instances that comprise the dataset represent (e.g., documents, photos, people, countries)?**

  EHRXQA contains natural questions and corresponding SQL/NeuralSQL queries (text). MIMIC-CXR-VQA contains the image ID of the MIMIC-CXR dataset and their related natural questions.

- **How many instances are there in total (of each type, if appropriate)?**

  In EHRXQA, there are about 46.2K instances (16,366 image-related samples, 16,529 table-related samples, and 13,257 image+table-related samples). In MIMIC-CXR-VQA, there are about 377.4K instances.

- **Does the dataset contain all possible instances or is it a sample (not necessarily random) of instances from a larger set?**

  We will provide all instances in our GitHub repository for EHRXQA[9] and MIMIC-CXR-VQA[10].

- **What data does each instance consist of?**

  EHRXQA contains (Question, SQL/NeuralSQL, Answer) pair for each instance. MIMIC-CXR-VQA contains (Question, CXR image ID, Answer) pair for each instance.

- **Is there a label or target associated with each instance?**

  The answer (label) is provided for each question.

- **Is any information missing from individual instances? If so, please provide a description, explaining why this information is missing (e.g., because it was unavailable). This does not include intentionally removed information, but might include, e.g., redacted text.**

  No.

- **Are relationships between individual instances made explicit (e.g., users' movie ratings, social network links)?**

  No.

- **Are there recommended data splits (e.g., training, development/validation, testing)?**

  See Appendix B.2.2, and Appendix C.3.3.

---

[9] https://github.com/baeseongsu/ehrxqa
[10] https://github.com/baeseongsu/mimic-cxr-vqa

- **Are there any errors, sources of noise, or redundancies in the dataset?**

  Questions are created by filling the slots in the templates with pre-defined values and records from the database. Thus, some questions can be grammatically incorrect but not critical (*e.g.*, verb tense).

- **Is the dataset self-contained, or does it link to or otherwise rely on external resources (e.g., websites, tweets, other datasets)?**

  EHRXQA depends on three open-source databases: MIMIC-IV[11], MIMIC-CXR[12], and Chest ImaGenome[13], which are accessible via PhysioNet[14]. MIMIC-CXR-VQA depends on two open-source databases: MIMIC-CXR and Chest ImaGenome.

- **Does the dataset contain data that might be considered confidential (e.g., data that is protected by legal privilege or by doctor-patient confidentiality, data that includes the content of individuals' non-public communications)?**

  No.

- **Does the dataset contain data that, if viewed directly, might be offensive, insulting, threatening, or might otherwise cause anxiety?**

  No.

- **Does the dataset relate to people?**

  Yes.

- **Does the dataset identify any subpopulations (e.g., by age, gender)?**

  No.

- **Does the dataset contain data that might be considered sensitive in any way (e.g., data that reveals race or ethnic origins, sexual orientations, religious beliefs, political opinions or union memberships, or locations; financial or health data; biometric or genetic data; forms of government identification, such as social security numbers; criminal history)?**

  No. The source datasets are already de-identified.

## A.3 Collection process

- **How was the data associated with each instance acquired?** To collect diverse questions, we constructed question templates and their associated query (SQL, NerualSQL) by analyzing existing resources (for image, medical VQA datasets, and for table, table-based EHR QA datasets). Then, we sampled QA samples from source databases (MIMIC-IV, MIMIC-CXR, Chest ImaGenome) for each question template.

- **What mechanisms or procedures were used to collect the data (e.g., hardware apparatuses or sensors, manual human curation, software programs, software APIs)?**

  We mainly used Excel, Google Sheets, and Python to collect, process and label the data. In addition, we used OpenAI's ChatGPT (GPT-4) to generate paraphrases for each question template.

- **If the dataset is a sample from a larger set, what was the sampling strategy (e.g., deterministic, probabilistic with specific sampling probabilities)?**

  When it needs random sampling such as data splitting, patient sampling, CXR image sampling for each (question, answer) pair, we fixed the random seed and randomly selected a fixed number of samples from the larger set.

- **Who was involved in the data collection process (e.g., students, crowd workers, contractors) and how were they compensated (e.g., how much were crowd workers paid)?**

  The data collection and construction process, which included SQL/NeuralSQL labeling, was performed exclusively by the authors of the study. No crowd workers were involved due to the sensitive nature of the data and the specialized knowledge required for query labeling.

---

[11] https://physionet.org/content/mimiciv/2.2/
[12] https://physionet.org/content/mimic-cxr/2.0.0/
[13] https://physionet.org/content/chest-imagenome/1.0.0/
[14] https://physionet.org/

- **Over what timeframe was the data collected?**

  The EHRXQA and MIMIC-CXR-VQA datasets were constructed in 2023. They were built using data from the MIMIC-CXR and MIMIC-IV databases. The MIMIC-IV data was collected between 2008 and 2019, and the MIMIC-CXR data was collected between 2011 and 2016.

- **Were any ethical review processes conducted (e.g., by an institutional review board)?**

  N/A.

- **Does the dataset relate to people?**

  Yes.

- **Did you collect the data from the individuals in question directly, or obtain it via third parties or other sources (e.g., websites)?**

  N/A.

- **Were the individuals in question notified about the data collection?**

  N/A.

- **Did the individuals in question consent to the collection and use of their data?**

  N/A.

- **If consent was obtained, were the consenting individuals provided with a mechanism to revoke their consent in the future or for certain uses?**

  N/A.

- **Has an analysis of the potential impact of the dataset and its use on data subjects (e.g., a data protection impact analysis) been conducted?**

  The dataset does not have individual-specific information.

### A.4 Preprocessing/cleaning/labeling

- **Was any preprocessing/cleaning/labeling of the data done (e.g., discretization or bucketing, tokenization, part-of-speech tagging, SIFT feature extraction, removal of instances, processing of missing values)?**

  N/A.

- **Was the "raw" data saved in addition to the preprocess/cleaned/labeled data (e.g., to support unanticipated future uses)?**

  N/A.

- **Is the software that was used to preprocess/clean/label the data available?**

  Preprocessing, cleaning, and labeling are done via Excel, Google Sheets, and Python.

### A.5 Uses

- **Has the dataset been used for any tasks already?**

  No.

- **Is there a repository that links to any or all papers or systems that use the dataset?**

  No.

- **What (other) tasks could the dataset be used for?**

  Our dataset is designed to promote research in question answering systems related to structured electronic health records (EHRs), chest X-ray (CXR) images, or a combination of both.

- **Is there anything about the composition of the dataset or the way it was collected and preprocessed/cleaned/labeled that might impact future uses?**

  N/A.

- **Are there tasks for which the dataset should not be used?**

  N/A.

## A.6 Distribution

- **Will the dataset be distributed to third parties outside of the entity (e.g., company, institution, organization) on behalf of which the dataset was created?**
  No.

- **How will the dataset be distributed?**
  The datasets will be released at `https://github.com/baeseongsu/ehrxqa` and `https://github.com/baeseongsu/mimic-cxr-vqa` upon publication.

- **Will the dataset be distributed under a copyright or other intellectual property (IP) license, and/or under applicable terms of use (ToU)?**
  The dataset is released under MIT License.

- **Have any third parties imposed IP-based or other restrictions on the data associated with the instances?**
  No.

- **Do any export controls or other regulatory restrictions apply to the dataset or to individual instances?**
  No.

## A.7 Maintenance

- **Who will be supporting/hosting/maintaining the dataset?**
  The authors of this paper.

- **How can the owner/curator/manager of the dataset be contacted(e.g., email address)?**
  Contact the first authors (`seongsu@kaist.ac.kr` & `kyungdaeun@kaist.ac.kr`).

- **Is there an erratum?**
  No.

- **Will the dataset be updated (e.g., to correct labeling erros, add new instances, delete instances)?**
  If any corrections are required, our plan is to upload an updated version of the dataset with comprehensive explanations for the changes. Furthermore, as we broaden our QA scope, we will consistently update the dataset with new QA templates/instances.

- **If the dataset relates to people, are there applicable limits on the retention of the data associated with the instances (e.g., were the individuals in question told that their data would be retained for a fixed period of time and then deleted)?**
  N/A

- **Will older versions of the dataset continue to be supported/hosted/maintained?**
  Primarily, we plan to maintain only the most recent version of the dataset. However, under certain circumstances, such as significant updates to our dataset or the need for validation of previous research work using older versions, we will exceptionally preserve previous versions of the dataset for up to one year.

- **If others want to extend/augment/build on/contribute to the dataset, is there a mechanism for them to do so?**
  Contact the authors in this paper.

# B Preliminary

## B.1 Uni-modal data resources

For our research, we utilize the dataset under the PhysioNet license, ensuring compliance with the required credentials and permissions. The following are the data resources we utilize:

- MIMIC-IV: Available at https://physionet.org/content/mimiciv/2.2/
- MIMIC-CXR: Available at https://physionet.org/content/mimic-cxr-jpg/2.0.0/
- Chest ImaGenome: Available at https://physionet.org/content/chest-imagenome/1.0.0/

## B.2 Uni-modal EHR QA datasets

### B.2.1 Table-based EHR QA

☑ **Template construction for MIMIC-IV** Given the structural similarities between the MIMIC-IV and MIMIC-III databases, we successfully adapted the original MIMIC-III question templates from the EHRSQL dataset for use with MIMIC-IV. Our methodology involved a comprehensive analysis of the question templates associated with both the MIMIC-III and MIMIC-IV database schemas to identify similarities and discrepancies. Through this comparative study, we were able to pinpoint the discrepancies between the question templates designed for MIMIC-III and those compatible with MIMIC-IV. While a substantial portion (*i.e.*, 165 templates) of the question templates from MIMIC-III could be seamlessly adapted to MIMIC-IV, we identified several question templates that were unique to each database, as presented in Table B1. For instance, MIMIC-IV provides information about microbiology test names, a feature absent in MIMIC-III. Taking these differences into account, we have assembled a collection of 174 question templates specifically designed for MIMIC-IV.

Table B1: We present a comparative analysis of template suitability between MIMIC-III and MIMIC-IV. This includes question templates from EHRSQL, indicating their applicability to MIMIC-III and/or MIMIC-IV. Checkmarks (✓) and crosses (✗) are used to denote compatibility and incompatibility, respectively.

| No. | Question Template | MIMIC-III | MIMIC-IV |
|---|---|---|---|
| 21 | How many [unit_count] have passed since the [time_filter_exact1] time patient {patient_id} was transferred to ward ward_id on the current hospital visit? | ✓ | ✗ |
| 22 | How many [unit_count] have passed since the [time_filter_exact1] time patient {patient_id} received a procedure on the current hospital visit? | ✗ | ✗ |
| 23 | How many [unit_count] have passed since the [time_filter_exact1] time patient {patient_id} received a procedure_name procedure on the current hospital visit? | ✗ | ✗ |
| 29 | What was the [time_filter_exact1] ward of patient {patient_id} [time_filter_global1]? | ✓ | ✗ |
| 46 | What was the name of the allergy that patient {patient_id} had [time_filter_global1]? | ✗ | ✗ |
| 47 | What was the name of the substance that patient {patient_id} was allergic to [time_filter_global1]? | ✗ | ✗ |
| 49 | What was the organism name found in the [time_filter_exact1] {test_name} test of patient {patient_id} [time_filter_global1]? | ✗ | ✓ |
| 51 | What was the name of the microbiology test that patient {patient_id} [time_filter_exact1] received [time_filter_global1]? | ✗ | ✓ |
| 78 | When was patient patient_id's [time_filter_exact1] {test_name} test [time_filter_global1]? | ✗ | ✓ |
| 85 | Has_verb patient patient_id received a {procedure_name} procedure in other than the current hospital [time_filter_global1]? | ✗ | ✗ |
| 98 | Has_verb patient {patient_id} had any allergy [time_filter_global1]? | ✗ | ✗ |
| 101 | Has_verb patient {patient_id} had any {test_name} test result [time_filter_global1]? | ✗ | ✓ |
| 103 | Has_verb there been any organism found in the [time_filter_exact1] {test_name} test of patient {patient_id} [time_filter_global1]? | ✗ | ✓ |
| 137 | Count the number of patients who stayed in ward {ward_id} [time_filter_global1]. | ✓ | ✗ |
| 153 | Count the number of patients who received a {test_name} test [time_filter_global1]. | ✗ | ✓ |
| 180 | What are_verb the top [n_rank] frequent microbiology tests that patients had [time_filter_within] after having received a {procedure_name} procedure [time_filter_global1]? | ✗ | ✓ |

Continued on next page

Table B1: We present a comparative analysis of template suitability between MIMIC-III and MIM-IC-IV. This includes question templates from EHRSQL, indicating their applicability to MIMIC-III and/or MIMIC-IV. Checkmarks (✓) and crosses (✗) are used to denote compatibility and incompatibility, respectively. (Continued)

| No. | Question Template | MIMIC-III | MIMIC-IV |
|---|---|---|---|
| 176 | What are_verb the top [n_rank] frequent microbiology tests [time_filter_global1]? | ✗ | ✓ |
| 178 | What are_verb the top [n_rank] frequent microbiology tests that patients had [time_filter_within] after having been diagnosed with {diagnosis_name} [time_filter_global1]? | ✗ | ✓ |
| 180 | What are_verb the top [n_rank] frequent microbiology tests that patients had [time_filter_within] after having received a {procedure_name} procedure [time_filter_global1]? | ✗ | ✓ |

☑ **SQL annotations**   Following a similar approach to the template construction, we utilize the EHRSQL SQL templates and adapt them for MIMIC-IV. Due to the structural similarities between MIMIC-III and MIMIC-IV, we can seamlessly map the schema from MIMIC-III to MIMIC-IV without altering the SQL query logic significantly. This enables the efficient conversion of SQL queries from MIMIC-III to MIMIC-IV, thus reducing the time and effort required to adapt the queries for the new database. While most of the schema information (*i.e.*, table and column names) remains the same, there are minor modifications introduced in MIMIC-IV compared to MIMIC-III, as presented in Table B2. Therefore, we annotate the corresponding SQL queries for 174 question templates.

Table B2: Column-wise schema mapping from MIMIC-III to MIMIC-IV

| MIMIC-III schema | MIMIC-IV schema |
|---|---|
| d_icd_diagnoses.icd9_code | d_icd_diagnoses.icd_code |
| d_icd_diagnoses.short_title | d_icd_diagnoses.long_title |
| d_icd_procedures.icd9_code | d_icd_procedures.icd_code |
| d_icd_procedures.short_title | d_icd_procedures.long_title |
| diagnoses_icd.icd9_code | diagnoses_icd.icd_code |
| procedures_icd.icd9_code | procedures_icd.icd_code |
| prescriptions.startdate | prescriptions.starttime |
| prescriptions.enddate | prescriptions.stoptime |
| chartevents.icustay_id | chartevents.stay_id |
| inputevents_cv.icustay_id | inputevents.stay_id |
| inputevents_cv.charttime | inputevents.starttime |
| outputevents.icustay_id | outputevents.stay_id |
| icustays.icustay_id | icustays.stay_id |
| transfers.icustay_id | transfers.transfer_id |

☑ **Other implementation details**   We use the fundamental template grammar from EHRSQL, including key components like the operation value slot, condition value slot, and time filter slot. Each slot has an associated natural language expression and a corresponding SQL pattern. For a complete list of time templates, operation values, and condition values, please refer to the comprehensive listing detailed in the original EHRSQL paper.

### B.2.2 Image-based EHR QA: MIMIC-CXR-VQA

☑ **Preprocessing** Our dataset is preprocessed following these steps:

1. **Image selection**

   (a) We filter out images captured from frontal view positions (*i.e.*, PA, AP).

   (b) Per study, we select one representative CXR image based on the earliest study datetime.

   (c) When multiple images in a study share the same study datetime, we choose the image whose dicom ID is alphabetically first.

2. **Outlier removal**

   (a) We cap the maximum number of consecutive studies per patient at 20.

   (b) We eliminate images missing bounding boxes for any anatomical locations.

   (c) We discard images with widths exceeding three standard deviations from the mean for each anatomical location.

   (d) To maintain uniformity between the gold and silver datasets' object and attribute pools, we eliminate six attributes (*i.e.*, "*aortic graft/repair*", "*artifact*", "*bronchiectasis*", "*diaphragmatic eventration (benign)*", "*pigtail catheter*", "*skin fold*") and one object (*i.e.*, "*left arm*") that are exclusively present in the silver dataset.

   (e) We also exclude the object "*right arm*" due to its association with the object "*left arm*", which is not present in the gold dataset.

3. **Label refinement**

   (a) To enhance the precision of label assignments, we employ three types of CXR ontology used in Chest ImaGenome: (i) parent-to-child (object) relationships; (ii) parent-to-child (attribute) relationships; (iii) possible relationships (object-attribute).

   (b) For parent-to-child (object) relationships, we propagate the presence of attributes from a child object to its parent object. For instance, if the left middle lung zone (a child object) is labeled with pneumonia (an attribute), then the left lung (its parent object) must also be labeled with pneumonia.

   (c) For parent-to-child (attribute) relationships, we propagate the presence of a child attribute to its parent attribute. For example, if lung cancer (a child attribute) is present in the left lung (an object), then lung opacity (a parent attribute) must also be present in the same object.

   (d) For possible relationships (object-attribute), we exclude any relationships between an object and its associated attributes that are not allowed by the ontology. For instance, according to this ontology, the object 'left lung' cannot be associated with the attribute 'clavicle fracture'.

4. **Dataset split**

   (a) From the *silver* dataset in Chest ImaGenome, which is machine-generated, we divide it into a 95:5 ratio, with 95% serving as the training image pool and 5% as the validation image pool. These image pools consist of 164,398 training images and 8,653 validation images.

   (b) The *gold* dataset, which is human-labeled, is used as the test image pool, consisting of 500 test images.

   (c) During the splitting process, we also balance the distribution between abnormal images (*i.e.*, studies with at least one attribute present) and normal images (*i.e.*, studies without any attributes).

   (d) Note that each image in image pools represents 563 relationships between 36 anatomical locations (*i.e.*, objects) and their associated attributes (a total of 68), indicating the presence or absence of an attribute for an object.

☑ **Question template construction -** *argument*  In our template, we designate five primary arguments, represented as ${...}: ${object}, ${attribute}, ${category}, ${viewpos}, and ${gender}. When an argument is required to appear more than once in a question template, each time with a unique value, we append an index to it, like ${object_1} or ${object_2}. Each of these arguments can be replaced by a specific value, as will be displayed in the following Table B3.

Table B3: Mapping of Arguments to their Potential Values

| Argument | Values |
|---|---|
| ${object} | abdomen, aortic arch, cardiac silhouette, carina, cavoatrial junction, left apical zone, left breast, left chest wall, left clavicle, left costophrenic angle, left hemidiaphragm, left hilar structures, left lower lung zone, left lung, left mid lung zone, left shoulder, left upper lung zone, mediastinum, neck, right apical zone, right atrium, right breast, right chest wall, right clavicle, right costophrenic angle, right hemidiaphragm, right hilar structures, right lower lung zone, right lung, right mid lung zone, right shoulder, right upper lung zone, spine, svc, trachea, upper mediastinum |
| ${attribute} | airspace opacity, alveolar hemorrhage, aspiration, atelectasis, bone lesion, breast-/nipple shadows, cabg grafts, calcified nodule, cardiac pacer and wires, chest port, chest tube, clavicle fracture, consolidation, copd/emphysema, costophrenic angle blunting, cyst/bullae, elevated hemidiaphragm, endotracheal tube, enlarged cardiac silhouette, enlarged hilum, enteric tube, fluid overload/heart failure, goiter, granulomatous disease, hernia, hydropneumothorax, hyperaeration, ij line, increased reticular markings/ild pattern, infiltration, interstitial lung disease, intra-aortic balloon pump, linear/patchy atelectasis, lobar/segmental collapse, low lung volumes, lung cancer, lung lesion, lung opacity, mass/nodule (not otherwise specified), mediastinal displacement, mediastinal drain, mediastinal widening, multiple masses/nodules, pericardial effusion, picc, pleural effusion, pleural/parenchymal scarring, pneumomediastinum, pneumonia, pneumothorax, prosthetic valve, pulmonary edema/hazy opacity, rib fracture, rotated, scoliosis, shoulder osteoarthritis, spinal degenerative changes, spinal fracture, sub-diaphragmatic air, subclavian line, subcutaneous air, superior mediastinal mass/enlargement, swan-ganz catheter, tortuous aorta, tracheostomy tube, vascular calcification, vascular congestion, vascular redistribution |
| ${category} | anatomicalfinding, device, disease, technicalassessment, tubesandlines |
| ${viewpos} | AP, PA |
| ${gender} | male, female |

☑ **Question template construction -** *template component*  We define each question template in our structure with three major components:

- **Filter condition**: This defines the question's domain or subject area. For instance, in the template "*Is there ${attribute} in the ${object}?*", "*${object}*" serves as the filter condition, focusing the question on a particular anatomical location. Filter conditions can include multiple arguments to create more complex queries using unions, intersections, or differences.

- **Target pattern**: This denotes the particular detail within the filter condition's scope that the question seeks to explore. In the example template, "*Is there ${attribute} in the ${object}?*", "*${attribute}*" forms the target pattern. When this is combined with the semantic type, it yields a question with a completed intent.

- **Semantic type**: This labels the question based on the nature of the expected response. There are three primary semantic types: '*verify*' for yes/no questions, '*query*' for answers in the form of a list or set, and '*choose*' for questions that involve selection from provided options.

☑ **Question template construction - *content type*** Following the construction of our templates, we classify our 48 question templates into seven distinct **content types** (See Table B4): *anatomy*, *attribute*, *presence*, *abnormality*, *plane*, *gender*, and *size*. Although these categories are often referred to as "question type" in other medical VQA datasets, we choose to refer to them as "content type" to provide a more precise characterization. Each content type is described in detail as follows:

- *anatomy*: We include all question templates related to asking about anatomical locations in the target pattern, but exclude verification questions from this content type.

- *attribute*: We include all question templates related to asking about attributes or categories in the target pattern, but exclude verification questions from this content type.

- *presence*: We include all verify questions that ask about the presence of attributes or categories given the entire image or specific anatomical locations.

- *abnormality*: We include all question templates that are related to abnormality (defining the concept of "*abnormality*" as a superset of four categories) in their questions.

- *plane*: We include the determination of the radiography's view position, following the VQA-RAD's QA scope.

- *gender*: We include the identification of gender from the images, following the VQA-RAD's QA scope.

- *size*: We include two clinically significant measurements: cardiothoracic ratio (*i.e.*, CTR) and mediastinal-thoracic ratio (*i.e.*, MTR). The CTR measures the maximal horizontal cardiac diameter against the maximal horizontal thoracic diameter (inner edge of ribs/edge of pleura). Conversely, the MTR calculates the ratio of the maximum mediastinal width to the maximum thoracic width. We derive these ratios using three measurements: the cardiac silhouette's width, the upper mediastinum's width, and the thorax width. The thorax width is defined by the largest x-axis value of the left lung and the smallest x-axis value of the right lung, considering the original reverse orientation of an X-ray. We have established normal measurement thresholds, aligning with the conventional parameters in radiology (CTR: 1/2, MTR: 1/3).

Table B4: Content type of VQA question templates on a chest X-ray image.

| Content type | Sample question |
|---|---|
| anatomy | What are all anatomical locations where both infiltration and interstitial lung diseases can be found? |
| attribute | List all detected anatomical findings. |
| presence | Does the cardiac silhouette show any evidence of diseases or devices? |
| abnormality | Are there signs of abnormalities in both the left lung and the right lung? |
| plane | Is this X-ray image in the AP or PA view? |
| gender | Please specify the patient's gender. |
| size | Is the cardiac silhouette's width larger than half of the total thorax width? |

☑ **VQA dataset generation - *dataset balancing*** To build an unbiased VQA dataset, we designed the balancing rules based on the following considerations:

- Balancing Answers: To avoid language biases within the VQA dataset, we maximized the answer entropy during the sampling. This approach ensures diverse and well-distributed answers to each question, promoting comprehensive image understanding.

- Balancing Questions per Image: We considered the number of questions per image for sampling a variety of images. We limited each question template to one use per CXR image. Therefore, an image can have a minimum of 0 questions and a maximum of 48 (*i.e.*, the total number of our templates). We globally defined an image counter to increase the probability of less frequently sampled images being selected, thereby promoting greater diversity in our image set.

- Balancing Sampled Questions per Template: Lastly, we ensured a balanced number of sampled questions per template to maintain uniformity. It ensures that no particular template is over or under-sampled, leading to a fair and diverse question dataset.

☑ **Dataset collection - *paraphrasing*** To generate paraphrases, we leveraged the OpenAI UI, applying Figure B1 to create 30 paraphrases per template using the GPT-4 model (version May 24, 2023) as a base. Following this, human reviewers pruned any paraphrases that strayed from the initial template's meaning. For enhanced diversity in the training, validation, and test sets, we performed k-means clustering ($k = 4$) on the paraphrases of each template, based on their edit distance. This process grouped alike paraphrases, which we then distributed in a 3-to-1 ratio for the train and validation/test sets respectively. To conclude, we randomly implemented these paraphrases into the datasets, thereby ensuring a broad spectrum of linguistic variations. On an average, we had 16.5 paraphrases representing each template.

---

**Prompt Template for Paraphrasing: MIMIC-CXR-VQA**

You are an AI paraphraser for the medical domain (radiology).
Write `{{num_of_paraphrase}}` paraphrases for the given question without changing its original meaning. The paraphrases must adhere to the following conditions.

Conditions:

- The paraphrased question should be similar to real-world questions asked by a medical doctor when given a chest x-ray image.
- Keep the paraphrased question concise and straightforward.
- The answer to the paraphrased question should be identical to the answer to the original question.
- Maintain the placeholders in the format of ${placeholder} (e.g., ${object}, ${attribute}, ${category}, ${attribute_1}, ${object_1}).
- Ensure that the paraphrased question maintains these placeholders.

```
{% if content_type in ["anatomy", "attribute", "condition",
"abnormality"] %}
```

- The object-related placeholder will be replaced with the actual anatomical locations found in the image, such as 'left lung' or 'cardiac silhouette'.
- The attribute-related placeholder will be replaced with specific abnormalities that can be found in chest X-ray images, such as 'lung opacity' or 'lung cancer'.
- The category-related placeholder will be replaced with the corresponding category of the attribute, such as 'anatomical finding', 'disease', or 'tubes/lines'.
- Abnormality is a superset of four categories: anatomical finding, disease, device, and tubes/lines.
```
  {% if semantic_type == "verify" %}
```
  - Formulate the paraphrased question so it can be answered with a "yes" or "no" response.
```
  {% elif semantic_type == "query" %}
```
  - Formulate the paraphrased questions to elicit multiple possible answers.
```
  {% elif semantic_type == "choose" %}
```
  - The paraphrased questions should include 'which' (and ',') and be answerable by selecting one of two placeholders.
```
  {% else %}
  {% endif %}
{% else %}
```

- The viewpos-related placeholder will be replaced with the actual view position of the chest x-ray image, such as 'PA' or 'AP'.
- The gender-related placeholder will be replaced with the gender of the patient in the chest x-ray image, such as 'M' or 'F'.
- We define the cardiothoracic ratio (CTR) as the ratio of the width of the heart to the width of the thorax.
- We define the mediastinal thoracic ratio (MTR) as the ratio of the width of the mediastinum to the width of the thorax.
```
{% endif %}
```
Question: `{{question_template}}`
Paraphrased questions:

---

Figure B1: Prompt Template for Paraphrasing Question Templates for MIMIC-CXR-VQA. Elements enclosed within double braces {{}} are substituted with values specific to each template.

☑ **Dataset statistics of MIMIC-CXR-VQA** [Table B5,](#) [Table B6,](#) and [Table B7](#) present comprehensive statistics of the MIMIC-CXR-VQA dataset, detailing its overall, content type, and semantic type distributions respectively.

Table B5: Overall statistics of MIMIC-CXR-VQA.

|  | Training | Validation | Test |
|---|---|---|---|
| Images | 133,687 | 8,610 | 500 |
| Questions | 132,387 | 31,148 | 7,565 |
| Answers | 6,628 | 2,508 | 700 |
| Samples | 290,031 | 73,567 | 13,793 |

Table B6: Statistics of MIMIC-CXR-VQA by content type.

| Content Type | Training | Validation | Test |
|---|---|---|---|
| presence | 109,455 (37.7%) | 26,153 (35.5%) | 4,566 (33.1%) |
| anatomy | 37,952 (13.1%) | 10,210 (13.9%) | 1,963 (14.2%) |
| attribute | 49,948 (17.2%) | 13,111 (17.8%) | 2,578 (18.7%) |
| abnormality | 60,692 (20.9%) | 16,109 (21.9%) | 3,199 (23.2%) |
| size | 16,000 (5.5%) | 4,000 (5.4%) | 705 (5.1%) |
| plane | 7,992 (2.8%) | 1,992 (2.7%) | 386 (2.8%) |
| gender | 7,992 (2.8%) | 1,992 (2.7%) | 396 (2.9%) |

Table B7: Statistics of MIMIC-CXR-VQA by semantic type.

| Semantic Type | Training | Validation | Test |
|---|---|---|---|
| verify | 162,689 (56.1%) | 39,336 (53.5%) | 6,945 (50.4%) |
| choose | 28,560 (9.8%) | 7,806 (10.6%) | 1,523 (11.0%) |
| query | 98,782 (34.1%) | 26,425 (35.9%) | 5,325 (38.6%) |

☑ **Comparison with other medical VQA datasets** [Table B8](#) provides a comparison of MIMIC-CXR-VQA to other medical VQA datasets. Compared to other VQA datasets, MIMIC-CXR-VQA presents broader templates and covers a wider range of question types. While PathVQA, SLAKE, and P-VQA also have diverse question templates, they primarily focus on pathological questions and the utilization of medical knowledge graphs, which differs from our focus. We emphasize questions that can be answered by solely looking at the patient's X-ray image. When considering the number of templates, other datasets often categorize templates with minor linguistic differences as distinct, even if they express the same content and semantics (*e.g.*, "Is the POS scan normal?" and "Is the POS normal?"). We choose to unify these variations, denoting the count in parentheses in the "# Templates" column of [Table B8,](#) thereby recognizing them as identical templates. This approach highlights that our dataset includes a significantly larger number of unique templates.[15] Furthermore, our dataset's templates are more complex than others, incorporating compositional templates developed through set/logical operations, which is not commonly observed in other datasets.

Table B8: Statistics for MIMIC-CXR-VQA and comparisons with existing datasets.

| Dataset | # Images | # QA pairs | Source of images | QA Creation | # Question types | # Templates | Compositional | Publicly accessible |
|---|---|---|---|---|---|---|---|---|
| VQA-RAD | 315 | 3,515 | MedPix database | natural | 11 | ∗ | | ✓ |
| VQA-Med-2019 | 4,200 | 15,292 | MedPix database | synthetic | 4 | ∗ | | ✓ |
| PathVQA | 4,998 | 32,799 | Pathology textbook | synthetic | 7 | ∗ | | ✓ |
| VQA-Med-2020 | 5,000 | 5,000 | MedPix database Medical Decathlon | synthetic | 1 | 18 (8) | | ✓ |
| SLAKE | 642 | 14,000 | NIH Chest X-ray CHAOS | natural | 10 | ∗ | | ✓ |
| VQA-Med-2021 | 5,000 | 5,000 | MedPix database | synthetic | 1 | 6 (4) | | ✓ |
| RadVisDial | 91,060 | 455,300 | MIMIC-CXR | synthetic, natural | 1 | 1 (1) | | |
| OVQA | 2,001 | 19,020 | EMRs | synthetic | 6 | 72 (19) | | ✓ |
| Mimic-VQ | 134,400 | 297,723 | MIMIC-CXR | synthetic | 6 | 15 (13) | | |
| P-VQA | 2,169 | 24,800 | hospitals | synthetic | 13 | ∗ | | ✓ |
| **MIMIC-CXR-VQA** | 142,797 | 377,391 | MIMIC-CXR | synthetic, paraphrased | 7 | 794 (48) | ✓ | ✓ |

[15]Note that the asterisk (∗) in the "# Templates" column indicates that the templates were either manually created by physicians, derived from natural questions, or undocumented, making it challenging to represent the number of templates accurately.

## ☑ Full list of VQA question template in MIMIC-CXR-VQA

Table B9: Full list of 48 VQA question templates in MIMIC-CXR-VQA

| Index | Content Type | Question Template |
|-------|--------------|-------------------|
| 1 | presence | Are there any ${category} in the ${object}? |
| 2 | presence | Is there ${attribute} in the ${object}? |
| 3 | abnormality | Is the ${object} abnormal? |
| 4 | presence | Are there any ${category_1} or ${category_2} in the ${object}? |
| 5 | presence | Are there both ${attribute_1} and ${attribute_2} in the ${object}? |
| 6 | presence | Is there either ${attribute_1} or ${attribute_2} in the ${object}? |
| 7 | attribute | List all ${category} in the ${object}. |
| 8 | abnormality | List all abnormalities in the ${object}. |
| 9 | attribute | List all ${category_1} and ${category_2} in the ${object}. |
| 10 | attribute | Which ${category} is related to the ${object}, ${attribute_1} or ${attribute_2}? |
| 11 | abnormality | Are there any abnormalities in either the ${object_1} or the ${object_2}? |
| 12 | abnormality | Are there any abnormalities in both the ${object_1} and the ${object_2}? |
| 13 | attribute | List all ${category} in either the ${object_1} or the ${object_2}. |
| 14 | attribute | List all common ${category} in both the ${object_1} and the ${object_2}. |
| 15 | attribute | List all ${category} only in the ${object_1} but not in the ${object_2}. |
| 16 | abnormality | List all abnormalities in either the ${object_1} or the ${object_2}. |
| 17 | abnormality | List all common abnormalities in both the ${object_1} and the ${object_2}. |
| 18 | abnormality | List all abnormalities only in the ${object_1} but not in the ${object_2}. |
| 19 | presence | Are there any ${category}? |
| 20 | abnormality | Are there any abnormalities? |
| 21 | presence | Are there any ${category_1} or ${category_2}? |
| 22 | presence | Is there ${attribute}? |
| 23 | presence | Are there both ${attribute_1} and ${attribute_2}? |
| 24 | presence | Is there either ${attribute_1} or ${attribute_2}? |
| 25 | attribute | List all ${category}. |
| 26 | attribute | List all ${category_1} and ${category_2}. |
| 27 | abnormality | List all abnormalities. |
| 28 | attribute | Which ${category} is related, ${attribute_1} or ${attribute_2}? |
| 29 | presence | Are both the ${object_1} and the ${object_2} related to ${attribute}? |
| 30 | presence | Is either the ${object_1} or the ${object_2} related to ${attribute}? |
| 31 | anatomy | List all anatomical locations related to ${attribute}. |
| 32 | anatomy | Which anatomical location is related to ${attribute}, the ${object_1} or the ${object_2}? |
| 33 | abnormality | Which anatomical location is abnormal, the ${object_1} or the ${object_2}? |
| 34 | anatomy | List all anatomical locations related to either ${attribute_1} or ${attribute_2}. |
| 35 | anatomy | List all common anatomical locations related to both ${attribute_1} and ${attribute_2}. |
| 36 | anatomy | List all anatomical locations related to ${attribute_1} but not ${attribute_2}. |
| 37 | presence | Are there any ${category} related to the ${object_1} and the ${object_2}? |
| 38 | presence | Are there any ${category} related to the ${object_1} or the ${object_2}? |
| 39 | anatomy | List all anatomical locations related to any ${category}. |
| 40 | anatomy | List all anatomical locations related to any ${category_1} or ${category_2}. |
| 41 | plane | Is this an ${viewpos} view? |
| 42 | plane | Which view is in this image, AP or PA? |
| 43 | plane | What is the view of this image? |
| 44 | gender | Is this patient ${gender}? |
| 45 | gender | What is the gender of this patient, male or female? |
| 46 | gender | What is the gender of this patient? |
| 47 | size | Is the width of the cardiac silhouette wider than 1/2 of the thorax width? |
| 48 | size | Is the width of the upper mediastinum wider than 1/3 of the thorax width? |

## C EHRXQA

### C.1 Database construction

#### C.1.1 Database pre-processing

- We create a "*dod*" column to the PATIENTS table and assign it the date of birth calculated as follows: `dob = anchor_year - anchor_age`. The month and day of dob are randomly sampled.
- We create an "*age*" column in the ADMISSIONS table. To calculate the age at the time of admission for each subject, we subtract their anchor year from their admission year and then add their anchor age. This can be represented as: `age = (admission_year - anchor_year) + anchor_age`.
- We includes patients aged between 11 and 89.
- We manually time-shift each patient's first study time to a random time point between 2100 and 2105, while preserving the same intervals between all records.
- We limit the number of current patients to approximately 10% of the total patient population.
- We sample 800 patients for the silver database and 400 for the gold database.
- If a particular type of value has multiple associated units of measurement, we retain only the value with the most common unit and discard the others from the database.
- All records are converted to lowercase.

#### C.1.2 Overview of EHR database

The entire database schema is illustrated in Figure C2.

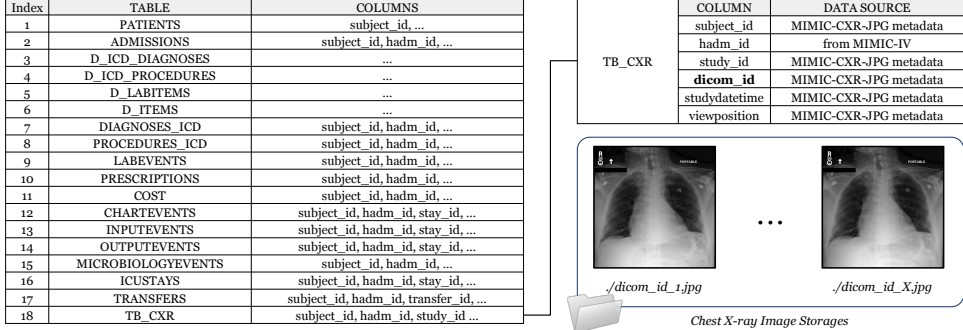

| Index | TABLE | COLUMNS |
|---|---|---|
| 1 | PATIENTS | subject_id, ... |
| 2 | ADMISSIONS | subject_id, hadm_id, ... |
| 3 | D_ICD_DIAGNOSES | ... |
| 4 | D_ICD_PROCEDURES | ... |
| 5 | D_LABITEMS | ... |
| 6 | D_ITEMS | ... |
| 7 | DIAGNOSES_ICD | subject_id, hadm_id, ... |
| 8 | PROCEDURES_ICD | subject_id, hadm_id, ... |
| 9 | LABEVENTS | subject_id, hadm_id, ... |
| 10 | PRESCRIPTIONS | subject_id, hadm_id, ... |
| 11 | COST | subject_id, hadm_id, ... |
| 12 | CHARTEVENTS | subject_id, hadm_id, stay_id, ... |
| 13 | INPUTEVENTS | subject_id, hadm_id, stay_id, ... |
| 14 | OUTPUTEVENTS | subject_id, hadm_id, stay_id, ... |
| 15 | MICROBIOLOGYEVENTS | subject_id, hadm_id, ... |
| 16 | ICUSTAYS | subject_id, hadm_id, stay_id, ... |
| 17 | TRANSFERS | subject_id, hadm_id, transfer_id, ... |
| 18 | TB_CXR | subject_id, hadm_id, study_id ... |

| TB_CXR | COLUMN | DATA SOURCE |
|---|---|---|
| | subject_id | MIMIC-CXR-JPG metadata |
| | hadm_id | from MIMIC-IV |
| | study_id | MIMIC-CXR-JPG metadata |
| | **dicom_id** | MIMIC-CXR-JPG metadata |
| | studydatetime | MIMIC-CXR-JPG metadata |
| | viewposition | MIMIC-CXR-JPG metadata |

./dicom_id_1.jpg  ...  ./dicom_id_X.jpg

*Chest X-ray Image Storages*

Figure C2: Overview of our EHR database schema, which comprises 18 tables: 17 from MIMIC-IV and one from MIMIC-CXR. The TB_CXR table includes DICOM image identifiers, enabling the loading of corresponding images directly from CXR image storage.

## C.2 Question template construction

### C.2.1 Detail of template construction strategy

☑ *Image*-related question    In the EHRXQA dataset, we present a unique scenario that diverges from the traditional Visual Question Answering (VQA) framework. In this scenario, instead of providing just an image and a question, our question templates also involve retrieving relevant images from a database to answer the question. However, specifying a particular study by directly stating its unique study ID can be complex and inconvenient for users, especially when questions refer to one or two images. To address this issue, we propose using practical, conversational expressions such as "*last study of patient patient_id in 2023*" or "*compared to the previous study*". This approach allows users to intuitively identify chest X-ray (CXR) studies using a variety of practical natural language expressions, as demonstrated in Table C10.

Regarding templates for 2-image question templates, our aim is to replicate clinical scenarios where we compare two separate or consecutive patient studies conducted during a hospital stay. In these situations, determining the severity of a particular disease can be subjective across different physicians, which can lead to higher costs for labeling. To address this issue, we have established four comparison labels: "*still present*", "*still absent*", "*newly detected*" and "*resolved*". These labels are automatically assigned based on changes in the presence of an attribute in an object. For example, if an initial study shows no signs of pneumonia in the left lung, but a subsequent study reveals the presence of pneumonia, the comparison label would be assigned as "*newly detected*".

Table C10: Example expressions for indicating CXR Studies in EHRXQA Dataset

| # of images | reference target study | reference to compared study |
|---|---|---|
| 1-image | study id {study_id} | - |
| | [time_filter_exact1] study of patient {patient_id} [time_filter_global1] | - |
| 2-image | study id {study_id1} | study id {study_id2} |
| | study id {study_id1} | previous study |
| | [time_filter_exact1] study of patient {patient_id} [time_filter_global1] | [time_filter_exact2] study of patient {patient_id} [time_filter_global2] |
| | [time_filter_exact1] study of patient {patient_id} [time_filter_global1] | previous study |

☑ *Image+Table*-related question    In the *Image+Table* modality, our focus is on three primary table events found in structured electronic health records (EHRs) along with CXR events: diagnosis, prescriptions, and procedures. It's important to note that CXR events are considered at the same admission level as these three medical events, indicating their comparable temporal hierarchy. By incorporating both structured medical events and CXR events, we explore three primary temporal scenarios: co-occurring events, where the events happen simultaneously; CXR events that occur after table events; and table events that follow the CXR events. Furthermore, when constructing our question templates in the *Image+Table* modality, we take into account demographic information such as age and gender.

### C.2.2 Full List of Question Template in EHRXQA

We provide a comprehensive collection of question templates for your reference: 168 image-related templates can be found in Table C11, 174 table-related templates are detailed in Table C12, and a further 75 templates are presented in Table C13.

Table C11: Full list of 168 *Image*-related question templates in EHRXQA

| Patient scope | Modality scope | | Question Template |
|---|---|---|---|
| single | Image | 1-image | Given the [time_filter_exact1] study of patient {patient_id} [time_filter_global1], are there any ${category} in the ${object}? |
| single | Image | 1-image | Given the [time_filter_exact1] study of patient {patient_id} [time_filter_global1], is there ${attribute} in the ${object}? |
| single | Image | 1-image | Given the [time_filter_exact1] study of patient {patient_id} [time_filter_global1], is the ${object} abnormal? |
| single | Image | 1-image | Given the [time_filter_exact1] study of patient {patient_id} [time_filter_global1], are there any ${category_1} or ${category_2} in the ${object}? |
| single | Image | 1-image | Given the [time_filter_exact1] study of patient {patient_id} [time_filter_global1], are there both ${attribute_1} and ${attribute_2} in the ${object}? |
| single | Image | 1-image | Given the [time_filter_exact1] study of patient {patient_id} [time_filter_global1], is there either ${attribute_1} or ${attribute_2} in the ${object}? |
| single | Image | 1-image | Given the [time_filter_exact1] study of patient {patient_id} [time_filter_global1], list all ${category} in the ${object}. |
| single | Image | 1-image | Given the [time_filter_exact1] study of patient {patient_id} [time_filter_global1], list all abnormality in the ${object}. |
| single | Image | 1-image | Given the [time_filter_exact1] study of patient {patient_id} [time_filter_global1], list all ${category_1} and ${category_2} in the ${object}. |
| single | Image | 1-image | Given the [time_filter_exact1] study of patient {patient_id} [time_filter_global1], which ${category} is related to the ${object}, ${attribute_1} or ${attribute_2}? |
| single | Image | 1-image | Given the [time_filter_exact1] study of patient {patient_id} [time_filter_global1], are there any abnormality in either the ${object_1} or the ${object_2}? |
| single | Image | 1-image | Given the [time_filter_exact1] study of patient {patient_id} [time_filter_global1], are there any abnormality in both the ${object_1} and the ${object_2}? |
| single | Image | 1-image | Given the [time_filter_exact1] study of patient {patient_id} [time_filter_global1], list all ${category} in either the ${object_1} or the ${object_2}. |
| single | Image | 1-image | Given the [time_filter_exact1] study of patient {patient_id} [time_filter_global1], list all common ${category} in both the ${object_1} and the ${object_2}. |
| single | Image | 1-image | Given the [time_filter_exact1] study of patient {patient_id} [time_filter_global1], list all ${category} only in the ${object_1} but not in the ${object_2}. |
| single | Image | 1-image | Given the [time_filter_exact1] study of patient {patient_id} [time_filter_global1], list all abnormality in either the ${object_1} or the ${object_2}. |
| single | Image | 1-image | Given the [time_filter_exact1] study of patient {patient_id} [time_filter_global1], list all common abnormality in both the ${object_1} and the ${object_2}. |
| single | Image | 1-image | Given the [time_filter_exact1] study of patient {patient_id} [time_filter_global1], list all abnormality only in the ${object_1} but not in the ${object_2}. |
| single | Image | 1-image | Given the [time_filter_exact1] study of patient {patient_id} [time_filter_global1], are there any ${category}? |
| single | Image | 1-image | Given the [time_filter_exact1] study of patient {patient_id} [time_filter_global1], are there any abnormality? |
| single | Image | 1-image | Given the [time_filter_exact1] study of patient {patient_id} [time_filter_global1], are there any ${category_1} or ${category_2}? |
| single | Image | 1-image | Given the [time_filter_exact1] study of patient {patient_id} [time_filter_global1], is there ${attribute}? |
| single | Image | 1-image | Given the [time_filter_exact1] study of patient {patient_id} [time_filter_global1], are there both ${attribute_1} and ${attribute_2}? |
| single | Image | 1-image | Given the [time_filter_exact1] study of patient {patient_id} [time_filter_global1], is there either ${attribute_1} or ${attribute_2}? |
| single | Image | 1-image | Given the [time_filter_exact1] study of patient {patient_id} [time_filter_global1], list all ${category}. |
| single | Image | 1-image | Given the [time_filter_exact1] study of patient {patient_id} [time_filter_global1], list all ${category_1} and ${category_2}. |
| single | Image | 1-image | Given the [time_filter_exact1] study of patient {patient_id} [time_filter_global1], list all abnormality. |
| single | Image | 1-image | Given the [time_filter_exact1] study of patient {patient_id} [time_filter_global1], which ${category} is related, ${attribute_1} or ${attribute_2}? |
| single | Image | 1-image | Given the [time_filter_exact1] study of patient {patient_id} [time_filter_global1], are both the ${object_1} and the ${object_2} related to ${attribute}? |
| single | Image | 1-image | Given the [time_filter_exact1] study of patient {patient_id} [time_filter_global1], is either the ${object_1} or the ${object_2} related to ${attribute}? |
| single | Image | 1-image | Given the [time_filter_exact1] study of patient {patient_id} [time_filter_global1], list all anatomical locations related to ${attribute}. |
| single | Image | 1-image | Given the [time_filter_exact1] study of patient {patient_id} [time_filter_global1], which anatomical location is related to ${attribute}, the ${object_1} or the ${object_2}? |
| single | Image | 1-image | Given the [time_filter_exact1] study of patient {patient_id} [time_filter_global1], which anatomical location is abnormal, the ${object_1} or the ${object_2}? |
| single | Image | 1-image | Given the [time_filter_exact1] study of patient {patient_id} [time_filter_global1], list all anatomical locations related to either ${attribute_1} or ${attribute_2}. |
| single | Image | 1-image | Given the [time_filter_exact1] study of patient {patient_id} [time_filter_global1], list all common anatomical locations related to both ${attribute_1} and ${attribute_2}. |
| single | Image | 1-image | Given the [time_filter_exact1] study of patient {patient_id} [time_filter_global1], list all anatomical locations related to ${attribute_1} but not ${attribute_2}. |

Continued on next page

Table C11: Full list of 168 *Image*-related question templates in EHRXQA (Continued)

| Patient scope | Modality scope | | Question Template |
|---|---|---|---|
| single | Image | 1-image | Given the [time_filter_exact1] study of patient {patient_id} [time_filter_global1], are there any ${category} related to the ${object_1} and the ${object_2}? |
| single | Image | 1-image | Given the [time_filter_exact1] study of patient {patient_id} [time_filter_global1], are there any ${category} related to the ${object_1} or the ${object_2}? |
| single | Image | 1-image | Given the [time_filter_exact1] study of patient {patient_id} [time_filter_global1], list all anatomical locations related to any ${category}. |
| single | Image | 1-image | Given the [time_filter_exact1] study of patient {patient_id} [time_filter_global1], list all anatomical locations related to any ${category_1} or ${category_2}. |
| single | Image | 1-image | Given the [time_filter_exact1] study of patient {patient_id} [time_filter_global1], is the width of the cardiac silhouette wider than 1/2 of the thorax width? |
| single | Image | 1-image | Given the [time_filter_exact1] study of patient {patient_id} [time_filter_global1], is the width of the upper mediastinum wider than 1/3 of the thorax width? |
| single | Image | 1-image | Given the study {study_id}, are there any ${category} in the ${object}? |
| single | Image | 1-image | Given the study {study_id}, is there ${attribute} in the ${object}? |
| single | Image | 1-image | Given the study {study_id}, is the ${object} abnormal? |
| single | Image | 1-image | Given the study {study_id}, are there any ${category_1} or ${category_2} in the ${object}? |
| single | Image | 1-image | Given the study {study_id}, are there both ${attribute_1} and ${attribute_2} in the ${object}? |
| single | Image | 1-image | Given the study {study_id}, is there either ${attribute_1} or ${attribute_2} in the ${object}? |
| single | Image | 1-image | Given the study {study_id}, list all ${category} in the ${object}. |
| single | Image | 1-image | Given the study {study_id}, list all abnormality in the ${object}. |
| single | Image | 1-image | Given the study {study_id}, list all ${category_1} and ${category_2} in the ${object}. |
| single | Image | 1-image | Given the study {study_id}, which ${category} is related to the ${object}, ${attribute_1} or ${attribute_2}? |
| single | Image | 1-image | Given the study {study_id}, are there any abnormality in either the ${object_1} or the ${object_2}? |
| single | Image | 1-image | Given the study {study_id}, are there any abnormality in both the ${object_1} and the ${object_2}? |
| single | Image | 1-image | Given the study {study_id}, list all ${category} in either the ${object_1} or the ${object_2}. |
| single | Image | 1-image | Given the study {study_id}, list all common ${category} in both the ${object_1} and the ${object_2}. |
| single | Image | 1-image | Given the study {study_id}, list all ${category} only in the ${object_1} but not in the ${object_2}. |
| single | Image | 1-image | Given the study {study_id}, list all abnormality in either the ${object_1} or the ${object_2}. |
| single | Image | 1-image | Given the study {study_id}, list all common abnormality in both the ${object_1} and the ${object_2}. |
| single | Image | 1-image | Given the study {study_id}, list all abnormality only in the ${object_1} but not in the ${object_2}. |
| single | Image | 1-image | Given the study {study_id}, are there any ${category}? |
| single | Image | 1-image | Given the study {study_id}, are there any abnormality? |
| single | Image | 1-image | Given the study {study_id}, are there any ${category_1} or ${category_2}? |
| single | Image | 1-image | Given the study {study_id}, is there ${attribute}? |
| single | Image | 1-image | Given the study {study_id}, are there both ${attribute_1} and ${attribute_2}? |
| single | Image | 1-image | Given the study {study_id}, is there either ${attribute_1} or ${attribute_2}? |
| single | Image | 1-image | Given the study {study_id}, list all ${category}. |
| single | Image | 1-image | Given the study {study_id}, list all ${category_1} and ${category_2}. |
| single | Image | 1-image | Given the study {study_id}, list all abnormality. |
| single | Image | 1-image | Given the study {study_id}, which ${category} is related, ${attribute_1} or ${attribute_2}? |
| single | Image | 1-image | Given the study {study_id}, are both the ${object_1} and the ${object_2} related to ${attribute}? |
| single | Image | 1-image | Given the study {study_id}, is either the ${object_1} or the ${object_2} related to ${attribute}? |
| single | Image | 1-image | Given the study {study_id}, list all anatomical locations related to ${attribute}. |
| single | Image | 1-image | Given the study {study_id}, which anatomical location is related to ${attribute}, the ${object_1} or the ${object_2}? |
| single | Image | 1-image | Given the study {study_id}, which anatomical location is abnormal, the ${object_1} or the ${object_2}? |
| single | Image | 1-image | Given the study {study_id}, list all anatomical locations related to either ${attribute_1} or ${attribute_2}. |
| single | Image | 1-image | Given the study {study_id}, list all common anatomical locations related to both ${attribute_1} and ${attribute_2}. |
| single | Image | 1-image | Given the study {study_id}, list all anatomical locations related to ${attribute_1} but not ${attribute_2}. |
| single | Image | 1-image | Given the study {study_id}, are there any ${category} related to the ${object_1} and the ${object_2}? |
| single | Image | 1-image | Given the study {study_id}, are there any ${category} related to the ${object_1} or the ${object_2}? |
| single | Image | 1-image | Given the study {study_id}, list all anatomical locations related to any ${category}. |
| single | Image | 1-image | Given the study {study_id}, list all anatomical locations related to any ${category_1} or ${category_2}. |
| single | Image | 1-image | Given the study {study_id}, is the width of the cardiac silhouette wider than 1/2 of the thorax width? |

Continued on next page

Table C11: Full list of 168 *Image*-related question templates in EHRXQA (Continued)

| Patient scope | Modality scope | | Question Template |
|---|---|---|---|
| single | Image | 1-image | Given the study {study_id}, is the width of the upper mediastinum wider than 1/3 of the thorax width? |
| single | Image | 2-image | Given the [time_filter_exact1] study of patient {patient_id} [time_filter_global1], are there any ${category} that are ${comparison} in the ${object} compared to the [time_filter_exact2] study of patient {patient_id} [time_filter_global2]? |
| single | Image | 2-image | Given the [time_filter_exact1] study of patient {patient_id} [time_filter_global1], is ${attribute} ${comparison} in the ${object} compared to the [time_filter_exact2] study of patient {patient_id} [time_filter_global2]? |
| single | Image | 2-image | Given the [time_filter_exact1] study of patient {patient_id} [time_filter_global1], are there any abnormality that are ${comparison} in the ${object} compared to the [time_filter_exact2] study of patient {patient_id} [time_filter_global2]? |
| single | Image | 2-image | Given the [time_filter_exact1] study of patient {patient_id} [time_filter_global1], are there any ${category} that are ${comparison} compared to the [time_filter_exact2] study of patient {patient_id} [time_filter_global2]? |
| single | Image | 2-image | Given the [time_filter_exact1] study of patient {patient_id} [time_filter_global1], are there any abnormality that are ${comparison} compared to the [time_filter_exact2] study of patient {patient_id} [time_filter_global2]? |
| single | Image | 2-image | Given the [time_filter_exact1] study of patient {patient_id} [time_filter_global1], is ${attribute} ${comparison} compared to the [time_filter_exact2] study of patient {patient_id} [time_filter_global2]? |
| single | Image | 2-image | Given the [time_filter_exact1] study of patient {patient_id} [time_filter_global1], list all ${category} that are ${comparison} in the ${object} compared to the [time_filter_exact2] study of patient {patient_id} [time_filter_global2]? |
| single | Image | 2-image | Given the [time_filter_exact1] study of patient {patient_id} [time_filter_global1], list all abnormality that are ${comparison} in the ${object} compared to the [time_filter_exact2] study of patient {patient_id} [time_filter_global2]? |
| single | Image | 2-image | Given the [time_filter_exact1] study of patient {patient_id} [time_filter_global1], list all ${category} that are ${comparison} compared to the [time_filter_exact2] study of patient {patient_id} [time_filter_global2]? |
| single | Image | 2-image | Given the [time_filter_exact1] study of patient {patient_id} [time_filter_global1], list all abnormality that are ${comparison} compared to the [time_filter_exact2] study of patient {patient_id} [time_filter_global2]? |
| single | Image | 2-image | Given the [time_filter_exact1] study of patient {patient_id} [time_filter_global1], list all anatomical locations related to ${attribute} that are ${comparison} compared to the [time_filter_exact2] study of patient {patient_id} [time_filter_global2]? |
| single | Image | 2-image | Given the [time_filter_exact1] study of patient {patient_id} [time_filter_global1], list all anatomical locations related to any ${category} that are ${comparison} compared to the [time_filter_exact2] study of patient {patient_id} [time_filter_global2]? |
| single | Image | 2-image | Given the [time_filter_exact1] study of patient {patient_id} [time_filter_global1], are there any ${category} that are ${comparison} in the ${object} compared to the previous study? |
| single | Image | 2-image | Given the [time_filter_exact1] study of patient {patient_id} [time_filter_global1], is ${attribute} ${comparison} in the ${object} compared to the previous study? |
| single | Image | 2-image | Given the [time_filter_exact1] study of patient {patient_id} [time_filter_global1], are there any abnormality that are ${comparison} in the ${object} compared to the previous study? |
| single | Image | 2-image | Given the [time_filter_exact1] study of patient {patient_id} [time_filter_global1], are there any ${category} that are ${comparison} compared to the previous study? |
| single | Image | 2-image | Given the [time_filter_exact1] study of patient {patient_id} [time_filter_global1], are there any abnormality that are ${comparison} compared to the previous study? |
| single | Image | 2-image | Given the [time_filter_exact1] study of patient {patient_id} [time_filter_global1], is ${attribute} ${comparison} compared to the previous study? |
| single | Image | 2-image | Given the [time_filter_exact1] study of patient {patient_id} [time_filter_global1], list all ${category} that are ${comparison} in the ${object} compared to the previous study? |
| single | Image | 2-image | Given the [time_filter_exact1] study of patient {patient_id} [time_filter_global1], list all abnormality that are ${comparison} in the ${object} compared to the previous study? |
| single | Image | 2-image | Given the [time_filter_exact1] study of patient {patient_id} [time_filter_global1], list all ${category} that are ${comparison} compared to the previous study? |
| single | Image | 2-image | Given the [time_filter_exact1] study of patient {patient_id} [time_filter_global1], list all abnormality that are ${comparison} compared to the previous study? |
| single | Image | 2-image | Given the [time_filter_exact1] study of patient {patient_id} [time_filter_global1], list all anatomical locations related to ${attribute} that are ${comparison} compared to the previous study? |
| single | Image | 2-image | Given the [time_filter_exact1] study of patient {patient_id} [time_filter_global1], list all anatomical locations related to any ${category} that are ${comparison} compared to the previous study? |
| single | Image | 2-image | Given the {study_id1} study, are there any ${category} that are ${comparison} in the ${object} compared to the {study_id2} study? |
| single | Image | 2-image | Given the {study_id1} study, is ${attribute} ${comparison} in the ${object} compared to the {study_id2} study? |
| single | Image | 2-image | Given the {study_id1} study, are there any abnormality that are ${comparison} in the ${object} compared to the {study_id2} study? |
| single | Image | 2-image | Given the {study_id1} study, are there any ${category} that are ${comparison} compared to the {study_id2} study? |
| single | Image | 2-image | Given the {study_id1} study, are there any abnormality that are ${comparison} compared to the {study_id2} study? |
| single | Image | 2-image | Given the {study_id1} study, is ${attribute} ${comparison} compared to the {study_id2} study? |
| single | Image | 2-image | Given the {study_id1} study, list all ${category} that are ${comparison} in the ${object} compared to the {study_id2} study? |
| single | Image | 2-image | Given the {study_id1} study, list all abnormality that are ${comparison} in the ${object} compared to the {study_id2} study? |
| single | Image | 2-image | Given the {study_id1} study, list all ${category} that are ${comparison} compared to the {study_id2} study? |
| single | Image | 2-image | Given the {study_id1} study, list all abnormality that are ${comparison} compared to the {study_id2} study? |

Continued on next page

Table C11: Full list of 168 *Image*-related question templates in EHRXQA (Continued)

| Patient scope | Modality scope | | Question Template |
|---|---|---|---|
| single | Image | 2-image | Given the {study_id1} study, list all anatomical locations related to ${attribute} that are ${comparison} compared to the {study_id2} study? |
| single | Image | 2-image | Given the {study_id1} study, list all anatomical locations related to any ${category} that are ${comparison} compared to the {study_id2} study? |
| single | Image | 2-image | Given the {study_id1} study, are there any ${category} that are ${comparison} in the ${object} compared to the previous study? |
| single | Image | 2-image | Given the {study_id1} study, is ${attribute} ${comparison} in the ${object} compared to the previous study? |
| single | Image | 2-image | Given the {study_id1} study, are there any abnormality that are ${comparison} in the ${object} compared to the previous study? |
| single | Image | 2-image | Given the {study_id1} study, are there any ${category} that are ${comparison} compared to the previous study? |
| single | Image | 2-image | Given the {study_id1} study, are there any abnormality that are ${comparison} compared to the previous study? |
| single | Image | 2-image | Given the {study_id1} study, is ${attribute} ${comparison} compared to the previous study? |
| single | Image | 2-image | Given the {study_id1} study, list all ${category} that are ${comparison} in the ${object} compared to the previous study? |
| single | Image | 2-image | Given the {study_id1} study, list all abnormality that are ${comparison} in the ${object} compared to the previous study? |
| single | Image | 2-image | Given the {study_id1} study, list all ${category} that are ${comparison} compared to the previous study? |
| single | Image | 2-image | Given the {study_id1} study, list all abnormality that are ${comparison} compared to the previous study? |
| single | Image | 2-image | Given the {study_id1} study, list all anatomical locations related to ${attribute} that are ${comparison} compared to the previous study? |
| single | Image | 2-image | Given the {study_id1} study, list all anatomical locations related to any ${category} that are ${comparison} compared to the previous study? |
| single | Image | N-image | How many [unit_count] have passed since the [time_filter_exact1] time patient {patient_id} had a chest X-ray study indicating ${attribute} in the ${object} [time_filter_global1]? |
| single | Image | N-image | How many [unit_count] have passed since the [time_filter_exact1] time patient {patient_id} had a chest X-ray study indicating any ${category} in the ${object} [time_filter_global1]? |
| single | Image | N-image | How many [unit_count] have passed since the [time_filter_exact1] time patient {patient_id} had a chest X-ray study indicating any abnormality in the ${object} [time_filter_global1]? |
| single | Image | N-image | How many [unit_count] have passed since the [time_filter_exact1] time patient {patient_id} had a chest X-ray study indicating ${attribute} [time_filter_global1]? |
| single | Image | N-image | How many [unit_count] have passed since the [time_filter_exact1] time patient {patient_id} had a chest X-ray study indicating any ${category} [time_filter_global1]? |
| single | Image | N-image | How many [unit_count] have passed since the [time_filter_exact1] time patient {patient_id} had a chest X-ray study indicating any abnormality [time_filter_global1]? |
| single | Image | N-image | When was the [time_filter_exact1] time that patient {patient_id} had a chest X-ray study indicating ${attribute} in the ${object} [time_filter_global1]? |
| single | Image | N-image | When was the [time_filter_exact1] time that patient {patient_id} had a chest X-ray study indicating any ${category} in the ${object} [time_filter_global1]? |
| single | Image | N-image | When was the [time_filter_exact1] time that patient {patient_id} had a chest X-ray study indicating any abnormality in the ${object} [time_filter_global1]? |
| single | Image | N-image | When was the [time_filter_exact1] time that patient {patient_id} had a chest X-ray study indicating ${attribute} [time_filter_global1]? |
| single | Image | N-image | When was the [time_filter_exact1] time that patient {patient_id} had a chest X-ray study indicating any ${category} [time_filter_global1]? |
| single | Image | N-image | When was the [time_filter_exact1] time that patient {patient_id} had a chest X-ray study indicating any abnormality [time_filter_global1]? |
| single | Image | N-image | Has {patient_id} had any chest X-ray study indicating ${attribute} in the ${object} [time_filter_global1]? |
| single | Image | N-image | Has {patient_id} had any chest X-ray study indicating any ${category} in the ${object} [time_filter_global1]? |
| single | Image | N-image | Has {patient_id} had any chest X-ray study indicating any abnormality in the ${object} [time_filter_global1]? |
| single | Image | N-image | Has {patient_id} had any chest X-ray study indicating ${attribute} [time_filter_global1]? |
| single | Image | N-image | Has {patient_id} had any chest X-ray study indicating any ${category} [time_filter_global1]? |
| single | Image | N-image | Has {patient_id} had any chest X-ray study indicating any abnormality [time_filter_global1]? |
| single | Image | N-image | Count the number of times that patient {patient_id} had chest X-ray studies indicating ${attribute} in the ${object} [time_filter_global1]. |
| single | Image | N-image | Count the number of times that patient {patient_id} had chest X-ray studies indicating any ${category} in the ${object} [time_filter_global1]. |
| single | Image | N-image | Count the number of times that patient {patient_id} had chest X-ray studies indicating any abnormality in the ${object} [time_filter_global1]. |
| single | Image | N-image | Count the number of times that patient {patient_id} had chest X-ray studies indicating ${attribute} [time_filter_global1]. |
| single | Image | N-image | Count the number of times that patient {patient_id} had chest X-ray studies indicating any ${category} [time_filter_global1]. |
| single | Image | N-image | Count the number of times that patient {patient_id} had chest X-ray studies indicating any abnormality [time_filter_global1]. |

Continued on next page

Table C11: Full list of 168 *Image*-related question templates in EHRXQA (Continued)

| Patient scope | Modality scope | | Question Template |
|---|---|---|---|
| group | Image | N-image | Count the number of patients who had any chest X-ray study indicating ${attribute} in the ${object} [time_filter_global1]. |
| group | Image | N-image | Count the number of patients who had any chest X-ray study indicating any ${category} in the ${object} [time_filter_global1]. |
| group | Image | N-image | Count the number of patients who had any chest X-ray study indicating any abnormality in the ${object} [time_filter_global1]. |
| group | Image | N-image | Count the number of patients who had any chest X-ray study indicating ${attribute} [time_filter_global1]. |
| group | Image | N-image | Count the number of patients who had any chest X-ray study indicating any ${category} [time_filter_global1]. |
| group | Image | N-image | Count the number of patients who had any chest X-ray study indicating any abnormality [time_filter_global1]. |
| group | Image | N-image | List the IDs of patients who had any chest X-ray study indicating ${attribute} in the ${object} [time_filter_global1]. |
| group | Image | N-image | List the IDs of patients who had any chest X-ray study indicating any ${category} in the ${object} [time_filter_global1]. |
| group | Image | N-image | List the IDs of patients who had any chest X-ray study indicating any abnormality in the ${object} [time_filter_global1]. |
| group | Image | N-image | List the IDs of patients who had any chest X-ray study indicating ${attribute} [time_filter_global1]. |
| group | Image | N-image | List the IDs of patients who had any chest X-ray study indicating any ${category} [time_filter_global1]. |
| group | Image | N-image | List the IDs of patients who had any chest X-ray study indicating any abnormality [time_filter_global1]. |

Table C12: Full list of 174 *Table*-related question templates in EHRXQA

| Patient scope | Modality scope | Question Template |
|---|---|---|
| none | Table | What is the intake method of {drug_name}? |
| none | Table | What is the cost of a procedure named {procedure_name}? |
| none | Table | What is the cost of a {lab_name} lab test? |
| none | Table | What is the cost of a drug named {drug_name}? |
| none | Table | What is the cost of diagnosing {diagnosis_name}? |
| none | Table | What does {abbreviation} stand for? |
| single | Table | What is the gender of patient {patient_id}? |
| single | Table | What is the date of birth of patient {patient_id}? |
| single | Table | What was the [time_filter_exact1] length of hospital stay of patient {patient_id}? |
| single | Table | What is the change in the weight of patient {patient_id} from the [time_filter_exact2] value measured [time_filter_global2] compared to the [time_filter_exact1] value measured [time_filter_global1]? |
| single | Table | What is the change in the value of {lab_name} of patient {patient_id} from the [time_filter_exact2] value measured [time_filter_global2] compared to the [time_filter_exact1] value measured [time_filter_global1]? |
| single | Table | What is the change in the {vital_name} of patient {patient_id} from the [time_filter_exact2] value measured [time_filter_global2] compared to the [time_filter_exact1] value measured [time_filter_global1]? |
| single | Table | Is the value of {lab_name} of patient {patient_id} [time_filter_exact2] measured [time_filter_global2] [comparison] than the [time_filter_exact1] value measured [time_filter_global1]? |
| single | Table | Is the {vital_name} of patient {patient_id} [time_filter_exact2] measured [time_filter_global2] [comparison] than the [time_filter_exact1] value measured [time_filter_global1]? |
| single | Table | What is_verb the age of patient {patient_id} [time_filter_global1]? |
| single | Table | What is_verb the name of insurance of patient {patient_id} [time_filter_global1]? |
| single | Table | What is_verb the marital status of patient {patient_id} [time_filter_global1]? |
| single | Table | What percentile is the value of {lab_value} in a {lab_name} lab test among patients of the same age as patient {patient_id} [time_filter_global1]? |
| single | Table | How many [unit_count] have passed since patient {patient_id} was admitted to the hospital currently? |
| single | Table | How many [unit_count] have passed since patient {patient_id} was admitted to the ICU currently? |
| single | Table | How many [unit_count] have passed since the [time_filter_exact1] time patient {patient_id} was transferred to careunit {careunit} on the current hospital visit? |
| single | Table | How many [unit_count] have passed since the [time_filter_exact1] time patient {patient_id} was diagnosed with {diagnosis_name} on the current hospital visit? |
| single | Table | How many [unit_count] have passed since the [time_filter_exact1] time patient {patient_id} was prescribed {drug_name} on the current hospital visit? |
| single | Table | How many [unit_count] have passed since the [time_filter_exact1] time patient {patient_id} received a {lab_name} lab test on the current hospital visit? |
| single | Table | How many [unit_count] have passed since the [time_filter_exact1] time patient {patient_id} had a {intake_name} intake on the current ICU visit? |
| single | Table | What was the [time_filter_exact1] hospital admission type of patient {patient_id} [time_filter_global1]? |
| single | Table | What was the [time_filter_exact1] careunit of patient {patient_id} [time_filter_global1]? |

Table C12: Full list of 174 *Table*-related question templates in EHRXQA (Continued)

| Patient scope | Modality scope | Question Template |
|---|---|---|
| single | Table | What was the [time_filter_exact1] measured height of patient {patient_id} [time_filter_global1]? |
| single | Table | What was the [time_filter_exact1] measured weight of patient {patient_id} [time_filter_global1]? |
| single | Table | What was the name of the diagnosis that patient {patient_id} [time_filter_exact1] received [time_filter_global1]? |
| single | Table | What was the name of the procedure that patient {patient_id} [time_filter_exact1] received [time_filter_global1]? |
| single | Table | What was the name of the drug that patient {patient_id} was [time_filter_exact1] prescribed via {drug_route} route [time_filter_global1]? |
| single | Table | What was the name of the drug that patient {patient_id} was [time_filter_exact1] prescribed [time_filter_global1]? |
| single | Table | What was the name of the drug that patient {patient_id} was prescribed [time_filter_within] after having been diagnosed with {diagnosis_name} [time_filter_global1]? |
| single | Table | What was the name of the drug that patient {patient_id} was prescribed [time_filter_within] after having received a {procedure_name} procedure [time_filter_global1]? |
| single | Table | What was the dose of {drug_name} that patient {patient_id} was [time_filter_exact1] prescribed [time_filter_global1]? |
| single | Table | What was the total amount of dose of {drug_name} that patient {patient_id} were prescribed [time_filter_global1]? |
| single | Table | What was the name of the drug that patient {patient_id} were prescribed [n_times] [time_filter_global1]? |
| single | Table | What is the new prescription of patient {patient_id} [time_filter_global2] compared to the prescription [time_filter_global1]? |
| single | Table | What was the [time_filter_exact1] measured value of a {lab_name} lab test of patient {patient_id} [time_filter_global1]? |
| single | Table | What was the name of the lab test that patient {patient_id} [time_filter_exact1] received [time_filter_global1]? |
| single | Table | What was the [agg_function] {lab_name} value of patient {patient_id} [time_filter_global1]? |
| single | Table | What was the organism name found in the [time_filter_exact1] {culture_name} microbiology test of patient {patient_id} [time_filter_global1]? |
| single | Table | What was the organism name found in the [time_filter_exact1] {test_name} test of patient {patient_id} [time_filter_global1]? |
| single | Table | What was the name of the specimen that patient {patient_id} was [time_filter_exact1] tested [time_filter_global1]? |
| single | Table | What was the name of the microbiology test that patient {patient_id} [time_filter_exact1] received [time_filter_global1]? |
| single | Table | What was the name of the intake that patient {patient_id} [time_filter_exact1] had [time_filter_global1]? |
| single | Table | What was the total volume of {intake_name} intake that patient {patient_id} received [time_filter_global1]? |
| single | Table | What was the total volume of intake that patient {patient_id} received [time_filter_global1]? |
| single | Table | What was the name of the output that patient {patient_id} [time_filter_exact1] had [time_filter_global1]? |
| single | Table | What was the total volume of {output_name} output that patient {patient_id} had [time_filter_global1]? |
| single | Table | What was the total volume of output that patient {patient_id} had [time_filter_global1]? |
| single | Table | What is the difference between the total volume of intake and output of patient {patient_id} [time_filter_global1]? |
| single | Table | What was the [time_filter_exact1] measured {vital_name} of patient {patient_id} [time_filter_global1]? |
| single | Table | What was the [agg_function] {vital_name} of patient {patient_id} [time_filter_global1]? |
| single | Table | What is_verb the total hospital cost of patient {patient_id} [time_filter_global1]? |
| single | Table | When was the [time_filter_exact1] hospital admission time of patient {patient_id} [time_filter_global1]? |
| single | Table | When was the [time_filter_exact1] hospital admission time that patient {patient_id} was admitted via {admission_route} [time_filter_global1]? |
| single | Table | When was the [time_filter_exact1] hospital discharge time of patient {patient_id} [time_filter_global1]? |
| single | Table | What was the [time_filter_exact1] length of ICU stay of patient {patient_id}? |
| single | Table | When was the [time_filter_exact1] time that patient {patient_id} was diagnosed with {diagnosis_name} [time_filter_global1]? |
| single | Table | When was the [time_filter_exact1] procedure time of patient {patient_id} [time_filter_global1]? |
| single | Table | When was the [time_filter_exact1] time that patient {patient_id} received a {procedure_name} procedure [time_filter_global1]? |
| single | Table | When was the [time_filter_exact1] prescription time of patient {patient_id} [time_filter_global1]? |
| single | Table | When was the [time_filter_exact1] time that patient {patient_id} was prescribed {drug_name} [time_filter_global1]? |
| single | Table | When was the [time_filter_exact1] time that patient {patient_id} was prescribed {drug_name1} and {drug_name2} [time_filter_within] [time_filter_global1]? |
| single | Table | When was the [time_filter_exact1] time that patient {patient_id} was prescribed a medication via {drug_route} route [time_filter_global1]? |
| single | Table | When was the [time_filter_exact1] lab test of patient {patient_id} [time_filter_global1]? |
| single | Table | When was the [time_filter_exact1] time that patient {patient_id} received a {lab_name} lab test [time_filter_global1]? |
| single | Table | When was the [time_filter_exact1] time that patient {patient_id} had the [sort] value of {lab_name} [time_filter_global1]? |
| single | Table | When was the [time_filter_exact1] microbiology test of patient {patient_id} [time_filter_global1]? |
| single | Table | When was patient {patient_id}'s [time_filter_exact1] {culture_name} microbiology test [time_filter_global1]? |
| single | Table | When was patient {patient_id}'s [time_filter_exact1] {test_name} test [time_filter_global1]? |

Continued on next page

Table C12: Full list of 174 *Table*-related question templates in EHRXQA (Continued)

| Patient scope | Modality scope | Question Template |
|---|---|---|
| single | Table | When was the [time_filter_exact1] time that patient {patient_id} had a {intake_name} intake [time_filter_global1]? |
| single | Table | When was the [time_filter_exact1] intake time of patient {patient_id} [time_filter_global1]? |
| single | Table | When was the [time_filter_exact1] time that patient {patient_id} had a {output_name} output [time_filter_global1]? |
| single | Table | When was the [time_filter_exact1] time that patient {patient_id} had a {vital_name} measured [time_filter_global1]? |
| single | Table | When was the [time_filter_exact1] time that the {vital_name} of patient {patient_id} was [comparison] than {vital_value} [time_filter_global1]? |
| single | Table | When was the [time_filter_exact1] time that patient {patient_id} had the [sort] {vital_name} [time_filter_global1]? |
| single | Table | Has_verb patient {patient_id} been admitted to the hospital [time_filter_global1]? |
| single | Table | Has_verb patient {patient_id} been to an emergency room [time_filter_global1]? |
| single | Table | Has_verb patient {patient_id} received any procedure [time_filter_global1]? |
| single | Table | Has_verb patient {patient_id} received a {procedure_name} procedure [time_filter_global1]? |
| single | Table | What was the name of the procedure that patient {patient_id} received [n_times] [time_filter_global1]? |
| single | Table | Has_verb patient {patient_id} received any diagnosis [time_filter_global1]? |
| single | Table | Has_verb patient {patient_id} been diagnosed with {diagnosis_name} [time_filter_global1]? |
| single | Table | Has_verb patient {patient_id} been prescribed {drug_name1}, {drug_name2}, or {drug_name3} [time_filter_global1]? |
| single | Table | Has_verb patient {patient_id} been prescribed any medication [time_filter_global1]? |
| single | Table | Has_verb patient {patient_id} been prescribed {drug_name} [time_filter_global1]? |
| single | Table | Has_verb patient {patient_id} received any lab test [time_filter_global1]? |
| single | Table | Has_verb patient {patient_id} received a {lab_name} lab test [time_filter_global1]? |
| single | Table | Has_verb patient {patient_id} had any microbiology test result [time_filter_global1]? |
| single | Table | Has_verb patient {patient_id} had any {culture_name} microbiology test result [time_filter_global1]? |
| single | Table | Has_verb patient {patient_id} had any {test_name} test result [time_filter_global1]? |
| single | Table | Has_verb there been any organism found in the [time_filter_exact1] {culture_name} microbiology test of patient {patient_id} [time_filter_global1]? |
| single | Table | Has_verb there been any organism found in the [time_filter_exact1] {test_name} test of patient {patient_id} [time_filter_global1]? |
| single | Table | Has_verb patient {patient_id} had any {intake_name} intake [time_filter_global1]? |
| single | Table | Has_verb patient {patient_id} had any {output_name} output [time_filter_global1]? |
| single | Table | Has_verb the {vital_name} of patient {patient_id} been ever [comparison] than {vital_value} [time_filter_global1]? |
| single | Table | Has_verb the {vital_name} of patient {patient_id} been normal [time_filter_global1]? |
| single | Table | List the hospital admission time of patient {patient_id} [time_filter_global1]. |
| single | Table | List the [unit_average] [agg_function] {lab_name} lab value of patient {patient_id} [time_filter_global1]. |
| single | Table | List the [unit_average] [agg_function] weight of patient {patient_id} [time_filter_global1]. |
| single | Table | List the [unit_average] [agg_function] volume of {intake_name} intake that patient {patient_id} received [time_filter_global1]. |
| single | Table | List the [unit_average] [agg_function] volume of {output_name} output that patient {patient_id} had [time_filter_global1]. |
| single | Table | List the [unit_average] [agg_function] {vital_name} of patient {patient_id} [time_filter_global1]. |
| single | Table | Count the number of hospital visits of patient {patient_id} [time_filter_global1]. |
| single | Table | Count the number of ICU visits of patient {patient_id} [time_filter_global1]. |
| single | Table | Count the number of times that patient {patient_id} received a {procedure_name} procedure [time_filter_global1]. |
| single | Table | Count the number of drugs patient {patient_id} were prescribed [time_filter_global1]. |
| single | Table | Count the number of times that patient {patient_id} were prescribed {drug_name} [time_filter_global1]. |
| single | Table | Count the number of times that patient {patient_id} received a {lab_name} lab test [time_filter_global1]. |
| single | Table | Count the number of times that patient {patient_id} had a {intake_name} intake [time_filter_global1]. |
| single | Table | Count the number of times that patient {patient_id} had a {output_name} output [time_filter_global1]. |
| group | Table | Count the number of current patients. |
| group | Table | Count the number of current patients aged [age_group]. |
| group | Table | What is the [n_survival_period] survival rate of patients diagnosed with {diagnosis_name}? |
| group | Table | What is the [n_survival_period] survival rate of patients who were prescribed {drug_name} after having been diagnosed with {diagnosis_name}? |
| group | Table | What are the top [n_rank] diagnoses that have the highest [n_survival_period] mortality rate? |
| group | Table | What is_verb the [agg_function] total hospital cost that involves a procedure named {procedure_name} [time_filter_global1]? |
| group | Table | What is_verb the [agg_function] total hospital cost that involves a {lab_name} lab test [time_filter_global1]? |
| group | Table | What is_verb the [agg_function] total hospital cost that involves a drug named {drug_name} [time_filter_global1]? |

Continued on next page

Table C12: Full list of 174 *Table*-related question templates in EHRXQA (Continued)

| Patient scope | Modality scope | Question Template |
|---|---|---|
| group | Table | What is_verb the [agg_function] total hospital cost that involves a diagnosis named {diagnosis_name} [time_filter_global1]? |
| group | Table | List the IDs of patients diagnosed with {diagnosis_name} [time_filter_global1]. |
| group | Table | What is the [agg_function] [unit_average] number of patient records diagnosed with {diagnosis_name} [time_filter_global1]? |
| group | Table | Count the number of patients who were dead after having been diagnosed with {diagnosis_name} [time_filter_within] [time_filter_global1]. |
| group | Table | Count the number of patients who did not come back to the hospital [time_filter_within] after diagnosed with {diagnosis_name} [time_filter_global1]. |
| group | Table | Count the number of patients who were admitted to the hospital [time_filter_global1]. |
| group | Table | Count the number of patients who were discharged from the hospital [time_filter_global1]. |
| group | Table | Count the number of patients who stayed in careunit {careunit} [time_filter_global1]. |
| group | Table | Count the number of patients who were diagnosed with {diagnosis_name} [time_filter_within] after having received a {procedure_name} procedure [time_filter_global1]. |
| group | Table | Count the number of patients who were diagnosed with {diagnosis_name2} [time_filter_within] after having been diagnosed with {diagnosis_name1} [time_filter_global1]. |
| group | Table | Count the number of patients who were diagnosed with {diagnosis_name} [time_filter_global1]. |
| group | Table | Count the number of patients who received a {procedure_name} procedure [time_filter_global1]. |
| group | Table | Count the number of patients who received a {procedure_name} procedure [n_times] [time_filter_global1]. |
| group | Table | Count the number of patients who received a {procedure_name2} procedure [time_filter_within] after having received a {procedure_name1} procedure [time_filter_global1]. |
| group | Table | Count the number of patients who received a {procedure_name} procedure [time_filter_within] after having been diagnosed with {diagnosis_name} [time_filter_global1]. |
| group | Table | Count the number of {procedure_name} procedure cases [time_filter_global1]. |
| group | Table | Count the number of patients who were prescribed {drug_name} [time_filter_global1]. |
| group | Table | Count the number of {drug_name} prescription cases [time_filter_global1]. |
| group | Table | Count the number of patients who were prescribed {drug_name} [time_filter_within] after having received a {procedure_name} procedure [time_filter_global1]. |
| group | Table | Count the number of patients who were prescribed {drug_name} [time_filter_within] after having been diagnosed with {diagnosis_name} [time_filter_global1]. |
| group | Table | Count the number of patients who received a {lab_name} lab test [time_filter_global1]. |
| group | Table | Count the number of patients who received a {culture_name} microbiology test [time_filter_global1]. |
| group | Table | Count the number of patients who received a {test_name} test [time_filter_global1]. |
| group | Table | Count the number of patients who had a {intake_name} intake [time_filter_global1]. |
| group | Table | What are_verb the top [n_rank] frequent diagnoses [time_filter_global1]? |
| group | Table | What are_verb the top [n_rank] frequent diagnoses of patients aged [age_group] [time_filter_global1]? |
| group | Table | What are_verb the top [n_rank] frequent diagnoses that patients were diagnosed [time_filter_within] after having received a {procedure_name} procedure [time_filter_global1]? |
| group | Table | What are_verb the top [n_rank] frequent diagnoses that patients were diagnosed [time_filter_within] after having been diagnosed with {diagnosis_name} [time_filter_global1]? |
| group | Table | What are_verb the top [n_rank] frequent procedures [time_filter_global1]? |
| group | Table | What are_verb the top [n_rank] frequent procedures of patients aged [age_group] [time_filter_global1]? |
| group | Table | What are_verb the top [n_rank] frequent procedures that patients received [time_filter_within] after having received a {procedure_name} procedure [time_filter_global1]? |
| group | Table | What are_verb the top [n_rank] frequent procedures that patients received [time_filter_within] after having been diagnosed with {diagnosis_name} [time_filter_global1]? |
| group | Table | What are_verb the top [n_rank] frequently prescribed drugs [time_filter_global1]? |
| group | Table | What are_verb the top [n_rank] frequently prescribed drugs of patients aged [age_group] [time_filter_global1]? |
| group | Table | What are_verb the top [n_rank] frequent prescribed drugs for patients who were also prescribed {drug_name} at the same time [time_filter_global1]? |
| group | Table | What are_verb the top [n_rank] frequent drugs that patients were prescribed [time_filter_within] after having been prescribed with {drug_name} [time_filter_global1]? |
| group | Table | What are_verb the top [n_rank] frequently prescribed drugs that patients were prescribed [time_filter_within] after having received a {procedure_name} procedure [time_filter_global1]? |
| group | Table | What are_verb the top [n_rank] frequently prescribed drugs that patients were prescribed [time_filter_within] after having been diagnosed with {diagnosis_name} [time_filter_global1]? |
| group | Table | What are_verb the top [n_rank] frequently prescribed drugs that patients aged [age_group] were prescribed [time_filter_within] after having been diagnosed with {diagnosis_name} [time_filter_global1]? |
| group | Table | What are_verb the top [n_rank] frequently prescribed drugs that {gender} patients aged [age_group] were prescribed [time_filter_within] after having been diagnosed with {diagnosis_name} [time_filter_global1]? |

Table C12: Full list of 174 *Table*-related question templates in EHRXQA (Continued)

| Patient scope | Modality scope | Question Template |
|---|---|---|
| group | Table | What are_verb the top [n_rank] frequent lab tests [time_filter_global1]? |
| group | Table | What are_verb the top [n_rank] frequent lab tests of patients aged [age_group] [time_filter_global1]? |
| group | Table | What are_verb the top [n_rank] frequent lab tests that patients had [time_filter_within] after having been diagnosed with {diagnosis_name} [time_filter_global1]? |
| group | Table | What are_verb the top [n_rank] frequent lab tests that patients had [time_filter_within] after having received a {procedure_name} procedure [time_filter_global1]? |
| group | Table | What are_verb the top [n_rank] frequent specimens tested [time_filter_global1]? |
| group | Table | What are_verb the top [n_rank] frequent microbiology tests [time_filter_global1]? |
| group | Table | What are_verb the top [n_rank] frequent specimens that patients were tested [time_filter_within] after having been diagnosed with {diagnosis_name} [time_filter_global1]? |
| group | Table | What are_verb the top [n_rank] frequent microbiology tests that patients had [time_filter_within] after having been diagnosed with {diagnosis_name} [time_filter_global1]? |
| group | Table | What are_verb the top [n_rank] frequent specimens that patients were tested [time_filter_within] after having received a {procedure_name} procedure [time_filter_global1]? |
| group | Table | What are_verb the top [n_rank] frequent microbiology tests that patients had [time_filter_within] after having received a {procedure_name} procedure [time_filter_global1]? |
| group | Table | What are_verb the top [n_rank] frequent intake events [time_filter_global1]? |
| group | Table | What are_verb the top [n_rank] frequent output events [time_filter_global1]? |

Table C13: Full list of 75 *Image+Table*-related question templates in EHRXQA

| Patient scope | Modality scope | Question Template |
|---|---|---|
| Single | Image + Table | Has_verb patient {patient_id} been diagnosed with {diagnosis_name} [time_filter_global1] and also had a chest X-ray study indicating ${attribute} in the ${object} within the same period? |
| Single | Image + Table | Has_verb patient {patient_id} been diagnosed with {diagnosis_name} [time_filter_global1] and also had a chest X-ray study indicating any ${category} in the ${object} within the same period? |
| Single | Image + Table | Has_verb patient {patient_id} been diagnosed with {diagnosis_name} [time_filter_global1] and also had a chest X-ray study indicating any abnormality in the ${object} within the same period? |
| Single | Image + Table | Has_verb patient {patient_id} been diagnosed with {diagnosis_name} [time_filter_global1] and also had a chest X-ray study indicating ${attribute} within the same period? |
| Single | Image + Table | Has_verb patient {patient_id} been diagnosed with {diagnosis_name} [time_filter_global1] and also had a chest X-ray study indicating any ${category} within the same period? |
| Single | Image + Table | Has_verb patient {patient_id} been diagnosed with {diagnosis_name} [time_filter_global1] and also had a chest X-ray study indicating any abnormality within the same period? |
| Single | Image + Table | Has_verb patient {patient_id} received a {procedure_name} procedure [time_filter_global1] and also had a chest X-ray study indicating ${attribute} in the ${object} within the same period? |
| Single | Image + Table | Has_verb patient {patient_id} received a {procedure_name} procedure [time_filter_global1] and also had a chest X-ray study indicating any ${category} in the ${object} within the same period? |
| Single | Image + Table | Has_verb patient {patient_id} received a {procedure_name} procedure [time_filter_global1] and also had a chest X-ray study indicating any abnormality in the ${object} within the same period? |
| Single | Image + Table | Has_verb patient {patient_id} received a {procedure_name} procedure [time_filter_global1] and also had a chest X-ray study indicating ${attribute} within the same period? |
| Single | Image + Table | Has_verb patient {patient_id} received a {procedure_name} procedure [time_filter_global1] and also had a chest X-ray study indicating any ${category} within the same period? |
| Single | Image + Table | Has_verb patient {patient_id} received a {procedure_name} procedure [time_filter_global1] and also had a chest X-ray study indicating any abnormality within the same period? |
| Single | Image + Table | Has_verb patient {patient_id} been prescribed with {drug_name} [time_filter_global1] and also had a chest X-ray study indicating ${attribute} in the ${object} within the same period? |
| Single | Image + Table | Has_verb patient {patient_id} been prescribed with {drug_name} [time_filter_global1] and also had a chest X-ray study indicating any ${category} in the ${object} within the same period? |
| Single | Image + Table | Has_verb patient {patient_id} been prescribed with {drug_name} [time_filter_global1] and also had a chest X-ray study indicating any abnormality in the ${object} within the same period? |
| Single | Image + Table | Has_verb patient {patient_id} been prescribed with {drug_name} [time_filter_global1] and also had a chest X-ray study indicating ${attribute} within the same period? |
| Single | Image + Table | Has_verb patient {patient_id} been prescribed with {drug_name} [time_filter_global1] and also had a chest X-ray study indicating any ${category} within the same period? |
| Single | Image + Table | Has_verb patient {patient_id} been prescribed with {drug_name} [time_filter_global1] and also had a chest X-ray study indicating any abnormality within the same period? |

Table C13: Full list of 75 *Image+Table*-related question templates in EHRXQA (Continued)

| Patient scope | Modality scope | Question Template |
|---|---|---|
| Single | Image + Table | Has_verb patient {patient_id} received any diagnosis [time_filter_global1] and also had a chest X-ray study indicating ${attribute} in the ${object} within the same period? |
| Single | Image + Table | Has_verb patient {patient_id} received any diagnosis [time_filter_global1] and also had a chest X-ray study indicating any ${category} in the ${object} within the same period? |
| Single | Image + Table | Has_verb patient {patient_id} received any diagnosis [time_filter_global1] and also had a chest X-ray study indicating ${attribute} within the same period? |
| Single | Image + Table | Has_verb patient {patient_id} received any procedure [time_filter_global1] and also had a chest X-ray study indicating ${attribute} in the ${object} within the same period? |
| Single | Image + Table | Has_verb patient {patient_id} received any procedure [time_filter_global1] and also had a chest X-ray study indicating any ${category} in the ${object} within the same period? |
| Single | Image + Table | Has_verb patient {patient_id} received any procedure [time_filter_global1] and also had a chest X-ray study indicating ${attribute} within the same period? |
| Single | Image + Table | Has_verb patient {patient_id} been prescribed any medication [time_filter_global1] and also had a chest X-ray study indicating ${attribute} in the ${object} within the same period? |
| Single | Image + Table | Has_verb patient {patient_id} been prescribed any medication [time_filter_global1] and also had a chest X-ray study indicating any ${category} in the ${object} within the same period? |
| Single | Image + Table | Has_verb patient {patient_id} been prescribed any medication [time_filter_global1] and also had a chest X-ray study indicating ${attribute} within the same period? |
| Single | Image + Table | Has_verb patient {patient_id} had a chest X-ray study indicating ${attribute} in the ${object} [time_filter_within] after having been diagnosed with {diagnosis_name} [time_filter_global1]? |
| Single | Image + Table | Has_verb patient {patient_id} had a chest X-ray study indicating any ${category} in the ${object} [time_filter_within] after having been diagnosed with {diagnosis_name} [time_filter_global1]? |
| Single | Image + Table | Has_verb patient {patient_id} had a chest X-ray study indicating any abnormality in the ${object} [time_filter_within] after having been diagnosed with {diagnosis_name} [time_filter_global1]? |
| Single | Image + Table | Has_verb patient {patient_id} had a chest X-ray study indicating ${attribute} [time_filter_within] after having been diagnosed with {diagnosis_name} [time_filter_global1]? |
| Single | Image + Table | Has_verb patient {patient_id} had a chest X-ray study indicating any ${category} [time_filter_within] after having been diagnosed with {diagnosis_name} [time_filter_global1]? |
| Single | Image + Table | Has_verb patient {patient_id} had a chest X-ray study indicating any abnormality [time_filter_within] after having been diagnosed with {diagnosis_name} [time_filter_global1]? |
| Single | Image + Table | Has_verb patient {patient_id} had a chest X-ray study indicating ${attribute} in the ${object} [time_filter_within] after having received {procedure_name} procedure [time_filter_global1] ? |
| Single | Image + Table | Has_verb patient {patient_id} had a chest X-ray study indicating any ${category} in the ${object} [time_filter_within] after having received {procedure_name} procedure [time_filter_global1] ? |
| Single | Image + Table | Has_verb patient {patient_id} had a chest X-ray study indicating any abnormality in the ${object} [time_filter_within] after having received {procedure_name} procedure [time_filter_global1] ? |
| Single | Image + Table | Has_verb patient {patient_id} had a chest X-ray study indicating ${attribute} [time_filter_within] after having received {procedure_name} procedure [time_filter_global1] ? |
| Single | Image + Table | Has_verb patient {patient_id} had a chest X-ray study indicating any ${category} [time_filter_within] after having received {procedure_name} procedure [time_filter_global1] ? |
| Single | Image + Table | Has_verb patient {patient_id} had a chest X-ray study indicating any abnormality [time_filter_within] after having received {procedure_name} procedure [time_filter_global1] ? |
| Single | Image + Table | Has_verb patient {patient_id} had a chest X-ray study indicating ${attribute} in the ${object} [time_filter_within] after having been prescribed with {drug_name} [time_filter_global1] ? |
| Single | Image + Table | Has_verb patient {patient_id} had a chest X-ray study indicating any ${category} in the ${object} [time_filter_within] after having been prescribed with {drug_name} [time_filter_global1] ? |
| Single | Image + Table | Has_verb patient {patient_id} had a chest X-ray study indicating any abnormality in the ${object} [time_filter_within] after having been prescribed with {drug_name} [time_filter_global1] ? |
| Single | Image + Table | Has_verb patient {patient_id} had a chest X-ray study indicating ${attribute} [time_filter_within] after having been prescribed with {drug_name} [time_filter_global1] ? |
| Single | Image + Table | Has_verb patient {patient_id} had a chest X-ray study indicating any ${category} [time_filter_within] after having been prescribed with {drug_name} [time_filter_global1] ? |
| Single | Image + Table | Has_verb patient {patient_id} had a chest X-ray study indicating any abnormality [time_filter_within] after having been prescribed with {drug_name} [time_filter_global1] ? |
| Single | Image + Table | Has_verb patient {patient_id} been diagnosed with {diagnosis_name} [time_filter_within] after having had a chest X-ray study indicating ${attribute} in the ${object} [time_filter_global1] ? |
| Single | Image + Table | Has_verb patient {patient_id} been diagnosed with {diagnosis_name} [time_filter_within] after having had a chest X-ray study indicating any ${category} in the ${object} [time_filter_global1] ? |
| Single | Image + Table | Has_verb patient {patient_id} been diagnosed with {diagnosis_name} [time_filter_within] after having had a chest X-ray study indicating any abnormality in the ${object} [time_filter_global1] ? |
| Single | Image + Table | Has_verb patient {patient_id} been diagnosed with {diagnosis_name} [time_filter_within] after having had a chest X-ray study indicating ${attribute} [time_filter_global1] ? |
| Single | Image + Table | Has_verb patient {patient_id} been diagnosed with {diagnosis_name} [time_filter_within] after having had a chest X-ray study indicating any ${category} [time_filter_global1] ? |

Continued on next page

Table C13: Full list of 75 *Image+Table*-related question templates in EHRXQA (Continued)

| Patient scope | Modality scope | Question Template |
|---|---|---|
| Single | Image + Table | Has_verb patient {patient_id} been diagnosed with {diagnosis_name} [time_filter_within] after having had a chest X-ray study indicating any abnormality [time_filter_global1] ? |
| Single | Image + Table | Has_verb patient {patient_id} received a {procedure_name} procedure [time_filter_within] after having had a chest X-ray study indicating ${attribute} in the ${object} [time_filter_global1] ? |
| Single | Image + Table | Has_verb patient {patient_id} received a {procedure_name} procedure [time_filter_within] after having had a chest X-ray study indicating any ${category} in the ${object} [time_filter_global1] ? |
| Single | Image + Table | Has_verb patient {patient_id} received a {procedure_name} procedure [time_filter_within] after having had a chest X-ray study indicating any abnormality in the ${object} [time_filter_global1] ? |
| Single | Image + Table | Has_verb patient {patient_id} received a {procedure_name} procedure [time_filter_within] after having had a chest X-ray study indicating ${attribute} [time_filter_global1] ? |
| Single | Image + Table | Has_verb patient {patient_id} received a {procedure_name} procedure [time_filter_within] after having had a chest X-ray study indicating any ${category} [time_filter_global1] ? |
| Single | Image + Table | Has_verb patient {patient_id} received a {procedure_name} procedure [time_filter_within] after having had a chest X-ray study indicating any abnormality [time_filter_global1] ? |
| Single | Image + Table | Has_verb patient {patient_id} been prescribed with {drug_name} [time_filter_within] after having had a chest X-ray study indicating ${attribute} in the ${object} [time_filter_global1] ? |
| Single | Image + Table | Has_verb patient {patient_id} been prescribed with {drug_name} [time_filter_within] after having had a chest X-ray study indicating any ${category} in the ${object} [time_filter_global1] ? |
| Single | Image + Table | Has_verb patient {patient_id} been prescribed with {drug_name} [time_filter_within] after having had a chest X-ray study indicating any abnormality in the ${object} [time_filter_global1] ? |
| Single | Image + Table | Has_verb patient {patient_id} been prescribed with {drug_name} [time_filter_within] after having had a chest X-ray study indicating ${attribute} [time_filter_global1] ? |
| Single | Image + Table | Has_verb patient {patient_id} been prescribed with {drug_name} [time_filter_within] after having had a chest X-ray study indicating any ${category} [time_filter_global1] ? |
| Single | Image + Table | Has_verb patient {patient_id} been prescribed with {drug_name} [time_filter_within] after having had a chest X-ray study indicating any abnormality [time_filter_global1] ? |
| Group | Image + Table | Count the number of patients aged [age_group] who had a chest X-ray study during hospital visit indicating ${attribute} in the ${object} [time_filter_global1]. |
| Group | Image + Table | Count the number of patients aged [age_group] who had a chest X-ray study during hospital visit indicating any ${category} in the ${object} [time_filter_global1]. |
| Group | Image + Table | Count the number of patients aged [age_group] who had a chest X-ray study during hospital visit indicating ${attribute} [time_filter_global1]. |
| Group | Image + Table | List the IDs of patients aged [age_group] who had a chest X-ray study during hospital visit indicating ${attribute} in the ${object} [time_filter_global1]. |
| Group | Image + Table | List the IDs of patients aged [age_group] who had a chest X-ray study during hospital visit indicating any ${category} in the ${object} [time_filter_global1]. |
| Group | Image + Table | List the IDs of patients aged [age_group] who had a chest X-ray study during hospital visit indicating ${attribute} [time_filter_global1]. |
| Group | Image + Table | Count the number of {gender} patients aged [age_group] who had a chest X-ray study during hospital visit indicating ${attribute} in the ${object} [time_filter_global1]. |
| Group | Image + Table | Count the number of {gender} patients aged [age_group] who had a chest X-ray study during hospital visit indicating any ${category} in the ${object} [time_filter_global1]. |
| Group | Image + Table | Count the number of {gender} patients aged [age_group] who had a chest X-ray study during hospital visit indicating ${attribute} [time_filter_global1]. |
| Group | Image + Table | List the IDs of {gender} patients aged [age_group] who had a chest X-ray study during hospital visit indicating ${attribute} in the ${object} [time_filter_global1]. |
| Group | Image + Table | List the IDs of {gender} patients aged [age_group] who had a chest X-ray study during hospital visit indicating any ${category} in the ${object} [time_filter_global1]. |
| Group | Image + Table | List the IDs of {gender} patients aged [age_group] who had a chest X-ray study during hospital visit indicating ${attribute} [time_filter_global1]. |

## C.3 QA dataset generation

### C.3.1 SQL/NeuralSQL annotation and QA pairs sampling

During the construction of our EHRXQA dataset, we sample *table*-related QA pairs by drawing from (Question, SQL) pairs and executing the SQL query to retrieve the answer. However, the process for *image*-related or *image+table*-related QA pairs is more complex, as we cannot sample QA pairs from (Question, SQL) without the label information for images. To overcome this, we create a new temporary table, `TB_CXR_PLUS`, which stores the label information for CXR images. The `TB_CXR_PLUS` table includes all the columns of the `TB_CXR` table, as well as additional columns that represent the 563 relationships between objects and attributes, as pre-processed in the MIMIC-CXR-VQA dataset. This table aids in annotating SQL queries to retrieve image information, effectively serving as an 'answer sheet' for QA dataset generation process. It is important to note that this temporary table is only used during data construction.

In keeping with our goal of retrieving rich information directly from the images themselves, we employ our new approach, NeuralSQL. As part of this, we use `TB_CXR_PLUS` to annotate SQL queries and `TB_CXR` to annotate NeuralSQL queries for all question templates related to *image* and *image+table*. An example of two queries can be seen in Figure C3. To sum up,

- For *table*-related question templates, we annotate the corresponding SQL query. During the dataset sampling process, we use these SQL queries to derive the answers. The final format of this part of the dataset is (Question, SQL, Answer).

- For *image*-related or *image+table*-related question templates, we annotate the corresponding SQL query using `TB_CXR_PLUS` and the NeuralSQL query using `TB_CXR`. During the dataset sampling process, we use the `TB_CXR_PLUS` query as an 'answer sheet' to dervie answers, and the NeuralSQL query to formulate questions directly over the image. The final format of this part of our dataset is (Question, NeuralSQL, Answer).

---

**Question Template:** Given the [time_filter_exact1] study of patient {patient_id} [time_filter_global1], **is there ${attribute} in the ${object}?**

**NeuralSQL Template:**

```
SELECT FUNC_VQA("is there ${attribute} in the ${object}?",T1.study_id)
FROM (
    SELECT tb_cxr.study_id
    FROM tb_cxr
    WHERE tb_cxr.subject_id = {patient_id}
    AND tb_cxr.hadm_id IN (
        SELECT admissions.hadm_id
        FROM admissions
        WHERE admissions.subject_id = {patient_id}
        time_filter_global1_event(admissions.admittime, admissions.dischtime)
    )
    time_filter_global1_absolute(tb_cxr.studydatetime)
    time_filter_exact1(tb_cxr.studydatetime)
) AS T1
```

**SQL Template:**

```
SELECT MAX(T1.relation = 1)
FROM (
    SELECT *
    FROM tb_cxr_plus
    WHERE tb_cxr_plus.study_id IN (
        SELECT DISTINCT tb_cxr_plus.study_id
        FROM tb_cxr_plus
        WHERE tb_cxr_plus.subject_id = {patient_id}
        AND tb_cxr_plus.hadm_id IN (
            SELECT admissions.hadm_id
            FROM admissions
            WHERE admissions.subject_id = {patient_id}
            time_filter_global1_event(admissions.admittime, admissions.dischtime)
        )
        time_filter_global1_absolute(tb_cxr_plus.studydatetime)
        time_filter_exact1(tb_cxr_plus.studydatetime)
    )
) AS T1
WHERE T1.object = "${object}" AND T1.attribute = "${attribute}"
```

Figure C3: Comparison of NeuralSQL and SQL templates given a *image*-related question template. Components highlighted in green indicate the same semantic meaning across the question, SQL, and NeuralSQL templates.

### C.3.2 NeuralSQL annotation details

- Four individuals, experienced in SQL and familiar with the EHRSQL dataset and its schema, participated in the SQL and NeuralSQL annotations. They were organized into teams to review each other's work.

- For NeuralSQL, if the sentence to be included in `FUNC_VQA` involved logical or set operations, we assigned the responsibility for these operations to SQL.

- In NeuralSQL, we aimed to maintain the natural language style of the original query in the VQA sentence to be included in `FUNC_VQA`.

### C.3.3 QA dataset split

We derive our train QA (*i.e.*, from silver database) and testing QA (*i.e.*, from gold database) sets from separate databases, with up to 80 samples for training and 10 for testing per template. The train QA set is divided further into a training QA set and a validation QA set, following a 7:1 ratio. This partitioning results in a validation set of a size comparable to that of the testing set.

### C.3.4 Paraphrasing

To generate paraphrases, we leveraged the OpenAI UI, applying Figure C4 to create 15 paraphrases per template using the GPT-4 model (version May 24, 2023). Following this, human reviewers pruned any paraphrases that strayed from the initial template's meaning. For the *Image*-related and *Image+Table*-related question templates, we used paraphrases crafted with GPT-4. In contrast, for the *Table*-related templates, we adopted machine paraphrases provided by EHRSQL. On average, each *Table*-related template contained 47.7 paraphrases, while each *Image* or *Image+Table*-related template contained 10.4 paraphrases. We randomly selected from these paraphrase pools and incorporated them into our datasets.

---

**Prompt Template for Paraphrasing: EHRXQA**

You are an AI paraphraser for the medical domain (radiology).
Write `{{num_of_paraphrase}}` paraphrases for the given question without changing its original meaning. The paraphrases must adhere to the following conditions.

Conditions:

- The paraphrased question should be similar to real-world questions asked by a medical doctor when given the EHR database.
- Keep the paraphrased question concise and straightforward.
- The answer to the paraphrased question should be identical to the answer to the original question.
- Maintain the placeholders in the format of ${placeholder} (e.g., ${object}, ${attribute}, ${category}, ${attribute_1}, ${object_1}).
- Maintain the placeholders in the format of [placeholder] (e.g., [time_filter_exact1], [time_filter_exact2], [time_filter_global1], [time_filter_global2]).
- Ensure that the paraphrased question maintains these placeholders.
- The unit-related placeholder will be replaced with the corresponding time expression, such as 'days, 'hours'.
- The gender-related placeholder will be replaced with the corresponding gender expression, such as 'male' or 'female'.
- The age-related placeholder will be replaced with the corresponding age expression, such as '30-40'.
- The exact time-related placeholder will be replaced with the corresponding time expression, such as 'first', 'second', or 'last'.
- The global time-related placeholder will be replaced with the corresponding time expression, such as 'on the first hospital visit', 'last year', or 'this month'.
- The procedure-related placeholder will be replaced with the specific procedure name, such as 'temporary tracheostomy' or 'venous cath nec'.
- The diagnosis-related placeholder will be replaced with the specific name of the diagnosis, such as 'PEG Insertion' or 'Invasive Ventilation'.
- The drug-related placeholder will be replaced with the specific medication name, such as 'danazol' or 'aspirin'.
- The comparison-related placeholder will be replaced with the corresponding comparison expression, such as 'still present' or 'newly detected'.
- The object-related placeholder will be replaced with the actual anatomical locations found in the image, such as 'left lung' or 'cardiac silhouette'.
- The attribute-related placeholder will be replaced with specific abnormalities that can be found in chest X-ray images, such as 'lung opacity' or 'lung cancer'.
- The category-related placeholder will be replaced with the corresponding category of the attribute, such as 'anatomical finding', 'disease', or 'tubes/lines'.
- Abnormality is a superset of four categories: anatomical finding, disease, device, and tubes/lines.
- Formulate the paraphrased questions to be answerable with `{{answer_type}}`.

Question: `{{question_template}}`
Paraphrased questions:

---

Figure C4: Prompt Template for Paraphrasing Question Templates for EHRXQA. Elements enclosed within double braces {{}} are substituted with values specific to each template.

## C.4 Data statistics

Table C14 presents detailed statistics of the EHRXQA dataset, providing a comprehensive breakdown of sample counts across various modality and patient categories in the training, validation, and test sets.

Table C14: Detailed Statistics of EHRXQA

| modality-based | patient-based | | # of samples | | |
|---|---|---|---|---|---|
| | | | train set | valid set | test set |
| Image | single | 1-image | 6,615 (18.3%) | 945 (18.3%) | 840 (17.5%) |
| | | 2-image | 3,410 (9.4%) | 488 (9.4%) | 468 (9.7%) |
| | | N-image | 1,890 (5.2%) | 279 (5.2%) | 240 (5.0%) |
| | group | | 945 (2.6%) | 135 (2.6%) | 120 (2.5%) |
| Table | none | | 396 (1.1%) | 54 (1.0%) | 50 (1.0%) |
| | single | | 8,219 (22.7%) | 1,151 (22.3%) | 1,080 (22.5%) |
| | group | | 4,346 (12.0%) | 647 (12.5%) | 586 (12.2%) |
| Image + Table | single | | 9,517 (26.3%) | 1,362 (26.3%) | 1,210 (25.2%) |
| | group | | 836 (2.3%) | 118 (2.3%) | 214 (4.5%) |

# D    NeuralSQL with visual question answering

To ensure compatibility with the original SQL grammar, we extend the production rules of the SQL query language for our API call `FUNC_VQA`. We achieve this by using the sqlgot[16] parser in the NeuralSQL interpreter. The sqlgot parser is designed to handle a wide range of SQL inputs and generate syntactically correct SQL in the targeted dialects. In our implementation, we perform batch-wise inference for the external VQA API to handle questions that involve multiple CXR images. This allows us to effectively process queries such as "Count the number of patients who had a chest X-ray study indicating...".

---

[16]https://github.com/tobymao/sqlglot

# E Experiments

## E.1 MIMIC-CXR-VQA

### E.1.1 MIMIC-CXR-VQA: Experimental settings

☑ **Standardization and Pre-processing of Pre-training Corpus**   To ensure fair comparisons, we standardize the pre-training corpus across all VLP models. Our strategy mitigates biases from varied data sources and ensures performance differences are attributed to the models' architectural characteristics and fine-tuning efforts rather than discrepancies in pre-training data. We adopt MedViLL's pre-processing strategy for both image and text data from MIMIC-CXR-JPG. For X-ray images, we remove the marginal space, adjust the resolution to fit the model's input size, and maintain the aspect ratio, discarding outliers not within 0.8 and 1.2. In processing the text data, we extract [17] the '*findings*' and '*impressions*' sections from the reports and then concatenate these sections for our models. If the token count, as tokenized by BERT, exceeds 253, we opt for the longer of the two sections. By adhering to the original MIMIC-CXR splits, we compile a corpus comprising 156,260 image-text pairs for the training set and 1,276 pairs for validation. We present results for both the original and controlled models, with the latter—pretrained on a carefully curated MIMIC-CXR corpus—denoted by an asterisk (∗).

☑ **Implementation details of VQA baselines**

- Prior (Most): This model is a prior model that outputs the most popular answer in the training and validation set, which is "yes".
- Prior (Question): This model is an advanced prior model that outputs the most popular answer in the training and validation set for each question (template).
- PubMedCLIP: We follow the original implementation code [18].
- MedViLL: We follow the original implementation code [19].
- $M^3AE$: We follow the original implementation code [20].

Table E15: Training, model configurations for VQA baselines along with resource information. Some model configurations are not reported if not applicable. For any other configurations that are not reported here, we followed the original paper.

| Name | $M^3AE^*$ | $M^3AE$ | MedViLL* | PubMedCLIP* | PubMedCLIP |
|---|---|---|---|---|---|
| ***Model configurations*** | | | | | |
| Visual encoder | ViT-B/16 | ViT-B/16 | RN50 | RN50 | RN50 |
| Text encoder | RoBERTa$_{base}$ | RoBERTa$_{base}$ | BERT$_{base}$ | Transformer / GRU | Transformer / GRU |
| ***Pre-training configurations*** | | | | | |
| Training epoch | 50 | 50 | 50 | 100 | 100 |
| Batch size | 256 | 256 | 128 | 64 | 64 |
| Learning rate | 5e-5 | 5e-5 | 1e-5 | 1e-5 | 1e-5 |
| ***Finetuninig configurations*** | | | | | |
| Training epoch | 50 | 50 | 20 | 20 | 20 |
| Batch size | 64 | 64 | 32 | 16 | 16 |
| Learning rate | 5e-6 | 5e-6 | 3e-5 | 1e-3 | 1e-3 |
| ***Resources*** **(pretraining / finetuning)** | | | | | |
| GPU device | A6000 × 4 / 1 | A6000 × 4 / 1 | A6000 × 4 / 1 | A6000 × 1 / 1 | A6000 × 1 / 1 |
| Training time | 35 / 16 hours | 69 / 16 hours | 32 / 66 hours | 226 / 62 hours | 226 / 62 hours |

---

[17] https://github.com/MIT-LCP/mimic-cxr/tree/master/txt
[18] https://github.com/sarahESL/PubMedCLIP
[19] https://github.com/SuperSupermoon/MedViLL
[20] https://github.com/zhjohnchan/M3AE

### E.1.2 MIMIC-CXR-VQA: Experimental results

Table E16: Comparison of performance of models on **MIMIC-CXR-VQA**.

| Model | Valid | | Test | |
|---|---|---|---|---|
| | Acc | F1 (micro) | Acc | F1 (micro) |
| Prior (Most) | 26.8 | 0.27 | 25.4 | 0.25 |
| Prior (Question) | 34.3 | 0.34 | 32.4 | 0.32 |
| PubMedCLIP | $55.1_{\pm 1.7}$ | $0.56_{\pm 0.02}$ | $54.9_{\pm 1.3}$ | $0.54_{\pm 0.02}$ |
| PubMedCLIP* | $56.6_{\pm 1.9}$ | $0.58_{\pm 0.02}$ | $56.5_{\pm 2.1}$ | $0.56_{\pm 0.02}$ |
| MedViLL* | $64.7_{\pm 0.2}$ | $0.69_{\pm 0.00}$ | $63.6_{\pm 0.1}$ | $0.67_{\pm 0.00}$ |
| M³AE | $68.9_{\pm 0.2}$ | $0.73_{\pm 0.00}$ | $68.9_{\pm 0.3}$ | $0.72_{\pm 0.00}$ |
| M³AE* | $70.2_{\pm 0.1}$ | $0.74_{\pm 0.00}$ | $69.2_{\pm 0.4}$ | $0.73_{\pm 0.00}$ |

Table E17: Comparison of performance (Acc) of models across content types on **MIMIC-CXR-VQA**.

| | Valid | | | | | | |
|---|---|---|---|---|---|---|---|
| Model | Plane | Gender | Size | Abnormality | Anatomy | Attribute | Presence |
| Prior (Most) | 16.7 | 16.7 | 50.0 | 24.8 | 0.0 | 0.0 | 50.1 |
| Prior (Question) | 50.0 | 50.0 | 29.5 | 12.8 | 15.7 | | 50.1 |
| PubMedCLIP | $84.5_{\pm 1.0}$ | $44.3_{\pm 6.6}$ | $75.2_{\pm 1.2}$ | $49.7_{\pm 2.0}$ | $34.6_{\pm 1.8}$ | $40.8_{\pm 3.0}$ | $69.1_{\pm 0.9}$ |
| PubMedCLIP* | $89.8_{\pm 4.2}$ | $53.5_{\pm 13.3}$ | $74.7_{\pm 1.9}$ | $50.9_{\pm 1.9}$ | $35.7_{\pm 2.5}$ | $43.0_{\pm 2.5}$ | $70.2_{\pm 0.9}$ |
| MedViLL* | $89.4_{\pm 5.2}$ | $65.0_{\pm 5.2}$ | $77.7_{\pm 0.2}$ | $60.3_{\pm 0.4}$ | $43.8_{\pm 0.6}$ | $55.0_{\pm 0.3}$ | $76.6_{\pm 0.3}$ |
| M³AE | $98.3_{\pm 1.2}$ | $92.9_{\pm 0.4}$ | $81.7_{\pm 0.3}$ | $64.4_{\pm 0.5}$ | $48.1_{\pm 0.6}$ | $59.4_{\pm 0.2}$ | $78.6_{\pm 0.1}$ |
| M³AE* | $97.7_{\pm 1.6}$ | $91.2_{\pm 1.1}$ | $82.5_{\pm 0.4}$ | $65.0_{\pm 0.5}$ | $49.1_{\pm 0.3}$ | $60.4_{\pm 0.3}$ | $81.0_{\pm 0.1}$ |
| | Test | | | | | | |
| Model | Plane | Gender | Size | Abnormality | Anatomy | Attribute | Presence |
| Prior (Most) | 17.1 | 16.7 | 43.3 | 23.9 | 0.0 | 0.0 | 50.4 |
| Prior (Question) | 48.7 | 50.0 | 43.3 | 29.0 | 11.7 | 12.4 | 50.4 |
| PubMedCLIP | $80.7_{\pm 2.0}$ | $44.3_{\pm 8.4}$ | $73.1_{\pm 0.1}$ | $49.6_{\pm 1.6}$ | $37.8_{\pm 1.0}$ | $42.3_{\pm 2.3}$ | $69.1_{\pm 0.8}$ |
| PubMedCLIP* | $87.5_{\pm 5.4}$ | $51.3_{\pm 11.0}$ | $74.1_{\pm 0.9}$ | $50.3_{\pm 1.8}$ | $38.5_{\pm 2.4}$ | $45.2_{\pm 2.4}$ | $70.1_{\pm 1.2}$ |
| MedViLL* | $90.2_{\pm 5.2}$ | $67.2_{\pm 5.0}$ | $75.1_{\pm 0.1}$ | $59.2_{\pm 0.6}$ | $45.3_{\pm 2.0}$ | $53.1_{\pm 0.3}$ | $76.0_{\pm 0.5}$ |
| M³AE | $98.0_{\pm 1.0}$ | $94.1_{\pm 0.8}$ | $79.1_{\pm 0.5}$ | $64.6_{\pm 0.5}$ | $51.6_{\pm 0.4}$ | $59.7_{\pm 0.6}$ | $78.3_{\pm 1.3}$ |
| M³AE* | $98.6_{\pm 1.0}$ | $90.9_{\pm 1.0}$ | $80.2_{\pm 1.1}$ | $63.9_{\pm 0.2}$ | $51.9_{\pm 1.7}$ | $60.2_{\pm 0.4}$ | $79.5_{\pm 0.7}$ |

### E.1.3 MIMIC-CXR-VQA: Relative metric for VQA grounding

☑ **Overview**  To estimate the achievable perception accuracy for single-image verification questions in MIMIC-CXR-VQA, we designed the reference model as a classification model. This model is capable of addressing our basic verification questions, which follow the template: "*Is there ${attribute} in the ${object}?*". The reference model is trained using a decomposed version of the MIMIC-CXR-VQA dataset. The details of the dataset construction and the reference model implementation are provided below.

☑ **Construction of train/valid/test (*verify*)**  The dataset is constructed with a focus on the basic verification template (*i.e.*, "Is there ${attribute} in ${object}?"). Note that all questions within the MIMIC-CXR-VQA dataset can be restructured as combinations of this basic verification template. For instance, a question such as "*Are there both ${attribute_1} and ${attribute_2} in the ${object}?*" can be divided into "*Is there ${attribute_1} in the ${object}?*" and "*Is there ${attribute_2} in the ${object}?*". Using this approach, we build the *verify* dataset, which consists of train (*verify*), valid (*verify*), and test (*verify*) sets. We decomposed the questions in each MIMIC-CXR-VQA dataset split to construct these subsets. We excluded category-related questions to avoid potential label imbalance.

☑ **Reference model structure**  The reference model focuses on local regions within the overall CXR image, guided by the bounding box data from Chest ImaGenome. Instead of considering the entire image as input, the model concentrates on these local regions in combination with attribute-specific headers to facilitate output for a grounding task. These headers identify the presence or absence of each attribute based on the local information provided. Through this strategy, the model can classify the relationship between objects and attributes, offering a distinct contrast to VQA models that rely on natural language questions to infer such relationships. The backbone of the reference model is the Vision Transformer (ViT), which is pre-trained using the Self-Distillation with No Labels (DINO) method. DINO is a self-supervised learning strategy that utilizes a momentum encoder and multi-crop training. This approach empowers the self-supervised ViT features to effectively capture explicit semantic segmentation information from an image. Considering that our VQA task requires attention to the anatomical locations specified in each question (*i.e.*, "*... in the ${object}?*") and classifying its attribute relationship (*i.e.*, "*Is there ${attribute} ...*"), the features derived from the DINO method offer significant advantages. Consequently, our reference model incorporates the DINO pre-trained ViT model as its backbone and adds a 3-layer MLP head for each attribute.

☑ **Experimental details**  We train our reference model following the setting of the previous work. We first pre-train the DINO model using our pre-training corpus (Appendix E.1.1). This pre-trained model is then fine-tuned using both the Train (*verify*) and Valid set (*verify*). The backbone of our model is ViT-S/16, and we use a 2D convolution layer with ReLU activation in each MLP head. During the pre-training phase, the model is trained using the AdamW optimizer with a batch size of 512, distributed over 8 GPUs. The learning rate increases linearly for the initial 10 epochs to a base value determined by the following linear scaling rule: $lr = 0.0005 \times \text{batchsize}/256$. Following this warm-up period, the learning rate decays according to a cosine schedule. For fine-tuning, we train the model for 100 epochs with a batch size of 1024. Here, we use the Adam optimizer with an initial learning rate of 1e-3.

**Experimental results with relative metric** The performance of our VQA models is evaluated by utilizing AUROC (Area Under the Receiver Operating Characteristic Curve) and relative AUROC metrics. Note that we only used object-attribute pairs with 10 or more instances where the corresponding object-attribute relationship was identified as positive (1) in the test set, to enhance the reliability of the evaluation. Thus, these metrics are computed across 82 specific (object, attribute) pairs within the MIMIC-CXR-VQA Test set (*verify*). These metrics deliver an all-encompassing view on the model's predictive precision and its ability to accurately identify attributes within objects.

Table E18: Comparison of performance of models across 82 (object, attribute) pairs on **MIMIC-CXR-VQA** Test set (*Verify*).

| object | attribute | support | AUROC | | | | $AUROC_{rel}$ | | |
| --- | --- | --- | --- | --- | --- | --- | --- | --- | --- |
| | | | *ref.* model | M³AE* | MedViLL | PubMedCLIP* | M³AE* | MedViLL | PubMedCLIP* |
| aortic arch | tortuous aorta | 69 | 0.719 | 0.880 | 0.852 | 0.645 | 1.225 | 1.185 | 0.898 |
| aortic arch | vascular calcification | 61 | 0.773 | 0.862 | 0.779 | 0.721 | 1.115 | 1.009 | 0.933 |
| cardiac silhouette | cardiac pacer and wires | 55 | 1.000 | 0.997 | 0.992 | 0.551 | 0.997 | 0.992 | 0.551 |
| cardiac silhouette | enlarged cardiac silhouette | 117 | 0.909 | 0.910 | 0.903 | 0.802 | 1.001 | 0.993 | 0.882 |
| cardiac silhouette | fluid overload/heart failure | 58 | 0.861 | 0.856 | 0.874 | 0.760 | 0.995 | 1.016 | 0.883 |
| carina | endotracheal tube | 64 | 0.671 | 0.916 | 0.883 | 0.754 | 1.369 | 1.319 | 1.126 |
| left costophrenic angle | costophrenic angle blunting | 62 | 0.681 | 0.873 | 0.762 | 0.689 | 1.286 | 1.123 | 1.016 |
| left costophrenic angle | lung opacity | 151 | 0.694 | 0.864 | 0.799 | 0.682 | 1.246 | 1.152 | 0.983 |
| left costophrenic angle | pleural effusion | 105 | 0.803 | 0.893 | 0.817 | 0.664 | 1.112 | 1.018 | 0.828 |
| left hemidiaphragm | hernia | 63 | 0.801 | 0.888 | 0.794 | 0.722 | 1.110 | 0.993 | 0.903 |
| left hilar structures | enlarged hilum | 60 | 0.671 | 0.778 | 0.709 | 0.634 | 1.161 | 1.059 | 0.945 |
| left hilar structures | lung opacity | 145 | 0.744 | 0.817 | 0.820 | 0.694 | 1.099 | 1.102 | 0.933 |
| left hilar structures | pulmonary edema/hazy opacity | 102 | 0.862 | 0.921 | 0.904 | 0.702 | 1.068 | 1.049 | 0.814 |
| left hilar structures | vascular congestion | 106 | 0.833 | 0.869 | 0.905 | 0.793 | 1.044 | 1.087 | 0.953 |
| left lower lung zone | aspiration | 63 | 0.897 | 0.898 | 0.820 | 0.746 | 1.001 | 0.914 | 0.831 |
| left lower lung zone | atelectasis | 113 | 0.833 | 0.841 | 0.735 | 0.646 | 1.010 | 0.883 | 0.776 |
| left lower lung zone | linear/patchy atelectasis | 69 | 0.661 | 0.766 | 0.643 | 0.643 | 1.159 | 0.972 | 0.973 |
| left lower lung zone | low lung volumes | 102 | 0.742 | 0.805 | 0.690 | 0.693 | 1.088 | 0.932 | 0.938 |
| left lower lung zone | lung opacity | 151 | 0.770 | 0.788 | 0.741 | 0.561 | 1.023 | 0.962 | 0.728 |
| left lower lung zone | pneumonia | 92 | 0.813 | 0.830 | 0.748 | 0.719 | 1.021 | 0.919 | 0.885 |
| left lung | aspiration | 75 | 0.901 | 0.871 | 0.839 | 0.707 | 0.967 | 0.931 | 0.785 |
| left lung | atelectasis | 123 | 0.871 | 0.874 | 0.779 | 0.684 | 1.004 | 0.895 | 0.786 |
| left lung | copd/emphysema | 52 | 0.904 | 0.904 | 0.866 | 0.678 | 1.000 | 0.958 | 0.751 |
| left lung | costophrenic angle blunting | 70 | 0.762 | 0.883 | 0.760 | 0.664 | 1.159 | 0.998 | 0.872 |
| left lung | enlarged hilum | 60 | 0.772 | 0.765 | 0.695 | 0.585 | 0.991 | 0.901 | 0.758 |
| left lung | fluid overload/heart failure | 60 | 0.827 | 0.864 | 0.893 | 0.753 | 1.045 | 1.080 | 0.910 |
| left lung | hyperaeration | 81 | 0.924 | 0.943 | 0.924 | 0.741 | 1.020 | 1.000 | 0.802 |
| left lung | linear/patchy atelectasis | 67 | 0.636 | 0.784 | 0.681 | 0.616 | 1.234 | 1.072 | 0.969 |
| left lung | low lung volumes | 107 | 0.904 | 0.913 | 0.865 | 0.714 | 1.010 | 0.957 | 0.790 |
| left lung | lung lesion | 85 | 0.685 | 0.685 | 0.654 | 0.577 | 1.004 | 0.958 | 0.845 |
| left lung | lung opacity | 158 | 0.802 | 0.785 | 0.739 | 0.662 | 0.979 | 0.921 | 0.825 |
| left lung | mass/nodule (not otherwise specified) | 87 | 0.661 | 0.763 | 0.588 | 0.548 | 1.159 | 0.893 | 0.832 |
| left lung | pleural effusion | 120 | 0.865 | 0.893 | 0.809 | 0.674 | 1.032 | 0.935 | 0.779 |
| left lung | pleural/parenchymal scarring | 86 | 0.782 | 0.824 | 0.768 | 0.602 | 1.056 | 0.984 | 0.772 |
| left lung | pneumonia | 99 | 0.820 | 0.766 | 0.711 | 0.668 | 0.936 | 0.868 | 0.815 |
| left lung | pulmonary edema/hazy opacity | 109 | 0.901 | 0.912 | 0.890 | 0.711 | 1.011 | 0.988 | 0.789 |
| left lung | vascular congestion | 103 | 0.869 | 0.871 | 0.885 | 0.786 | 1.001 | 1.018 | 0.904 |
| left mid lung zone | lung opacity | 148 | 0.685 | 0.821 | 0.712 | 0.523 | 1.200 | 1.041 | 0.765 |
| mediastinum | cardiac pacer and wires | 54 | 0.999 | 0.997 | 0.992 | 0.543 | 0.998 | 0.993 | 0.544 |
| mediastinum | enlarged cardiac silhouette | 122 | 0.905 | 0.899 | 0.875 | 0.793 | 0.994 | 0.968 | 0.876 |
| mediastinum | enteric tube | 66 | 0.890 | 0.998 | 0.978 | 0.777 | 1.122 | 1.099 | 0.873 |
| mediastinum | fluid overload/heart failure | 57 | 0.825 | 0.851 | 0.865 | 0.749 | 1.033 | 1.049 | 0.908 |
| mediastinum | hernia | 76 | 0.894 | 0.915 | 0.755 | 0.647 | 1.024 | 0.844 | 0.724 |
| mediastinum | ij line | 47 | 0.951 | 0.990 | 0.976 | 0.784 | 1.041 | 1.026 | 0.824 |
| mediastinum | superior mediastinal mass/enlargement | 68 | 0.752 | 0.804 | 0.689 | 0.625 | 1.070 | 0.916 | 0.831 |
| mediastinum | tortuous aorta | 74 | 0.904 | 0.867 | 0.837 | 0.619 | 0.959 | 0.926 | 0.685 |
| mediastinum | vascular calcification | 63 | 0.866 | 0.870 | 0.803 | 0.724 | 1.005 | 0.928 | 0.837 |
| right costophrenic angle | lung opacity | 149 | 0.828 | 0.888 | 0.827 | 0.699 | 1.074 | 1.000 | 0.845 |
| right costophrenic angle | pleural effusion | 102 | 0.733 | 0.844 | 0.804 | 0.618 | 1.151 | 1.096 | 0.843 |
| right hemidiaphragm | elevated hemidiaphragm | 58 | 0.912 | 0.968 | 0.775 | 0.547 | 1.062 | 0.850 | 0.600 |
| right hilar structures | enlarged hilum | 58 | 0.739 | 0.838 | 0.751 | 0.620 | 1.134 | 1.017 | 0.839 |
| right hilar structures | lung opacity | 152 | 0.790 | 0.855 | 0.831 | 0.689 | 1.082 | 1.051 | 0.872 |
| right hilar structures | pulmonary edema/hazy opacity | 96 | 0.867 | 0.908 | 0.896 | 0.693 | 1.047 | 1.034 | 0.799 |
| right hilar structures | vascular congestion | 100 | 0.860 | 0.857 | 0.884 | 0.798 | 0.997 | 1.028 | 0.929 |
| right lower lung zone | aspiration | 63 | 0.883 | 0.939 | 0.856 | 0.783 | 1.064 | 0.969 | 0.888 |
| right lower lung zone | atelectasis | 113 | 0.790 | 0.814 | 0.753 | 0.642 | 1.031 | 0.953 | 0.813 |
| right lower lung zone | linear/patchy atelectasis | 64 | 0.743 | 0.800 | 0.680 | 0.680 | 1.077 | 0.915 | 0.916 |
| right lower lung zone | low lung volumes | 106 | 0.827 | 0.833 | 0.713 | 0.708 | 1.010 | 0.864 | 0.858 |
| right lower lung zone | lung opacity | 154 | 0.786 | 0.769 | 0.741 | 0.600 | 0.979 | 0.943 | 0.764 |
| right lower lung zone | pneumonia | 91 | 0.835 | 0.789 | 0.768 | 0.612 | 0.945 | 0.920 | 0.733 |
| right lung | airspace opacity | 62 | 0.825 | 0.839 | 0.786 | 0.562 | 1.018 | 0.954 | 0.682 |
| right lung | aspiration | 65 | 0.936 | 0.953 | 0.864 | 0.740 | 1.019 | 0.924 | 0.791 |
| right lung | atelectasis | 129 | 0.809 | 0.833 | 0.758 | 0.685 | 1.031 | 0.938 | 0.848 |
| right lung | copd/emphysema | 51 | 0.912 | 0.916 | 0.877 | 0.702 | 1.004 | 0.962 | 0.770 |
| right lung | enlarged hilum | 62 | 0.768 | 0.830 | 0.715 | 0.586 | 1.082 | 0.931 | 0.763 |
| right lung | fluid overload/heart failure | 67 | 0.859 | 0.861 | 0.886 | 0.763 | 1.002 | 1.032 | 0.888 |
| right lung | hyperaeration | 78 | 0.961 | 0.941 | 0.940 | 0.766 | 0.979 | 0.978 | 0.797 |
| right lung | linear/patchy atelectasis | 65 | 0.804 | 0.806 | 0.751 | 0.728 | 1.002 | 0.935 | 0.905 |
| right lung | low lung volumes | 105 | 0.922 | 0.915 | 0.861 | 0.715 | 0.993 | 0.934 | 0.775 |
| right lung | lung lesion | 78 | 0.767 | 0.832 | 0.698 | 0.557 | 1.085 | 0.910 | 0.726 |
| right lung | lung opacity | 153 | 0.803 | 0.779 | 0.722 | 0.622 | 0.971 | 0.899 | 0.775 |
| right lung | mass/nodule (not otherwise specified) | 83 | 0.823 | 0.810 | 0.775 | 0.639 | 0.985 | 0.943 | 0.777 |
| right lung | pleural effusion | 113 | 0.841 | 0.851 | 0.803 | 0.632 | 1.012 | 0.955 | 0.751 |
| right lung | pleural/parenchymal scarring | 98 | 0.741 | 0.845 | 0.706 | 0.602 | 1.140 | 0.953 | 0.813 |
| right lung | pneumonia | 92 | 0.867 | 0.816 | 0.785 | 0.668 | 0.941 | 0.905 | 0.770 |
| right lung | pulmonary edema/hazy opacity | 111 | 0.928 | 0.924 | 0.895 | 0.727 | 0.995 | 0.964 | 0.783 |
| right lung | vascular congestion | 107 | 0.878 | 0.849 | 0.865 | 0.798 | 0.967 | 0.985 | 0.909 |
| right mid lung zone | lung opacity | 145 | 0.674 | 0.866 | 0.730 | 0.635 | 1.285 | 1.082 | 0.941 |
| trachea | endotracheal tube | 61 | 0.973 | 0.975 | 0.947 | 0.795 | 1.003 | 0.974 | 0.817 |
| upper mediastinum | superior mediastinal mass/enlargement | 65 | 0.774 | 0.821 | 0.712 | 0.644 | 1.062 | 0.920 | 0.832 |
| upper mediastinum | tortuous aorta | 67 | 0.843 | 0.881 | 0.845 | 0.621 | 1.045 | 1.003 | 0.737 |
| upper mediastinum | vascular calcification | 63 | 0.871 | 0.859 | 0.777 | 0.734 | 0.986 | 0.892 | 0.843 |

## E.2  MIMIC-CXR-VQA: Exploring data redundancy and the impact of paraphrasing

Given the brittleness of templates in prior EHR QA work, where the large sample size of emrQA (medication=220K, relation=900K) led to the redundancy issue, we decided to investigate the MIMIC-CXR-VQA dataset specifically, as it has a larger dataset size (377K).

First, we explore the degree to which additional templates actually improve the model to determine whether our dataset simply contains repetitive templates without any novelty. To this end, we conducted an ablation experiment with the MIMIC-CXR-VQA dataset. We evaluate the test set performance by randomly sampling training data at various template usage proportions (*i.e.*, how many unique templates are used for training), such as 5%, 10%, 20%, 50%, and 100%.

As shown in Table E19, our experiment with MIMIC-CXR-VQA demonstrated that using a higher number of questions (*i.e.*, training size increases) generated from templates questions generated from more diverse templates positively impacts the model's performance (*i.e.*, test Acc/F1) across all models. This observation suggests that MIMIC-CXR-VQA does not have the redundancy of questions, and question diversity generated from all templates contributes to the performance improvement.

Table E19: Results of the ablation experiment on the MIMIC-CXR-VQA test set. Comparison of performance across different training data proportions based on unique template usage: 5%, 10%, 20%, 50%, and 100%. We ran the experiments with one seed.

| template usage (%) | PubMedCLIP* test (Acc/F1) | MedViLL test (Acc/F1) | M$^3$AE* test (Acc/F1) |
|---|---|---|---|
| 5% | 49.1 / 0.39 | 44.8 / 0.44 | 56.9 / 0.56 |
| 10% | 48.4 / 0.50 | 54.4 / 0.55 | 61.3 / 0.64 |
| 20% | 51.2 / 0.50 | 60.0 / 0.62 | 65.0 / 0.67 |
| 50% | 55.8 / 0.56 | 62.6 / 0.65 | 68.9 / 0.72 |
| 100% | 56.5 / 0.56 | 63.6 / 0.67 | 69.2 / 0.73 |

Next, we explored the relationship between the variety of paraphrases produced using GPT-4 and model performance. To investigate whether increasing the diversity of paraphrases (thus reducing the redundancy of the question) improves the model's ability, we keep the dataset size constant and vary the number of paraphrases per template for the training dataset. We started with 48 seed templates as the training dataset (denoted as "seed template") and created two distinct training dataset variants named "low-diversity" and "high-diversity". These variants contained 20% and 100% paraphrased templates of the original template in the MIMIC-CXR-VQA training set, respectively. Note that each model, trained with three different datasets, was evaluated against the same original MIMIC-CXR-VQA test dataset. We conducted the experiments with three different seeds.

Our findings indicate that using paraphrased templates via GPT-4 improves model performance. This observation is evident when comparing the results of the "seed template" (without paraphrasing) with models trained in the variants "low-diversity" or "high-diversity". Further, as the diversity of paraphrases increases, the model performance might also improve. The effect of paraphrasing can differ based on the text encoder used. Models that incorporate a BERT-based language architecture, such as MedViLL and M$^3$AE, appear to benefit more from increased diversity. This suggests that even templates crafted through automatic paraphrasing can substantially boost performance, especially when there is adequate diversity.

Table E20: Results of the ablation experiment on the MIMIC-CXR-VQA test set. Comparison of performance across different degrees of paraphrasing diversity. We ran the experiments with three different seeds (mean ± std).

| Training Dataset | PubMedCLIP* test (Acc/F1) | MedViLL test (Acc/F1) | M$^3$AE* test (Acc/F1) |
|---|---|---|---|
| MIMIC-CXR-VQA (seed template) | 39.6 ± 0.4 / 0.36 ± 0.0 | 46.3 ± 0.3 / 0.48 ± 0.0 | 65.6 ± 0.5 / 0.68 ± 0.0 |
| MIMIC-CXR-VQA (low-diversity) | 56.5 ± 0.3 / 0.57 ± 0.0 | 62.5 ± 0.3 / 0.65 ± 0.0 | 69.2 ± 0.4 / 0.72 ± 0.0 |
| MIMIC-CXR-VQA (high-diversity) | 56.5 ± 2.1 / 0.56 ± 0.0 | 63.6 ± 0.1 / 0.67 ± 0.0 | 69.2 ± 0.4 / 0.73 ± 0.0 |

### E.3  EHRXQA

#### E.3.1  Implementation details of baselines

Our NeuralSQL-based approach integrates a large language model (LLM) as a parser with an external VQA API module. We use ChatGPT [42] (`gpt-3.5-turbo-0613`)[21] as our parser and utilize the M$^3$AE model, pre-trained on the MIMIC-CXR-VQA training set, as the VQA API module (frozen). We conduct in-context learning with few-shot samples, leveraging the capabilities of large language models, specifically in a 10-shot setting.

We have defined two prompting strategies: 1) Fixed: which uses 10-shot (Question, NeuralSQL) pairs; 2) BM25 (train): which retrieves 10 relevant pairs via BM25 from the training set for any given question. The fixed examples are randomly selected from the training set, but we ensured that at least one pair was sampled from each modality to provide the minimum required information (3 for *Table*-related, 4 for *Image*-related, and 3 for *Image+Table*-related). When executing NeuralSQL parsed by LLM, the batch size of the external VQA module is set to 16.

---

**Prompt for Fixed strategy.**

Generate NeuralSQL (*i.e.*, extended SQL with the following conditions) given the question to answer the question correctly. If the question can only be answered by examining a chest X-ray image and requires a VQA model, use the new syntax FUNC_VQA() to create a query. When the VQA sentence in FUNC_VQA() syntax contains logical operations such as union, difference, intersection, disjunction, or conjunction, decomposes the VQA statement into minimal semantic units and uses the SQL syntax to generate NeuralSQL. For example, decompose the original sentence "Are there any technical assessment or tubes/lines?" into "Are there any technical assessment?" and "Are there any tubes/lines?" by separating the logical disjunction (or) and creating two separate questions.

```
Q: {{1st question}}
NeuralSQL: {{1st query}}

...

Q: {{10th question}}
NeuralSQL: {{10th query}}

Parse the question into NeuralSQL.
Q: {{target question}}
NeuralSQL:
```

---

[21]Since the Codex API is no longer supported, we conducted all our experiments using ChatGPT (`gpt-3.5-turbo-0613`) instead.

**Prompt for BM25 strategy.**

Generate NeuralSQL (*i.e.*, extended SQL with the following conditions) given the question to answer the question correctly. If the question can only be answered by examining a chest X-ray image and requires a VQA model, use the new syntax FUNC_VQA() to create a query. When the VQA sentence in FUNC_VQA() syntax contains logical operations such as union, difference, intersection, disjunction, or conjunction, decomposes the VQA statement into minimal semantic units and uses the SQL syntax to generate NeuralSQL. For example, decompose the original sentence "Are there any technical assessment or tubes/lines?" into "Are there any technical assessment?" and "Are there any tubes/lines?" by separating the logical disjunction (or) and creating two separate questions. Q: `{{1st question retrieved from training QA set}}`
NeuralSQL: `{{1st query}}`

`...`

Q: `{{10th question retrieved from training QA set}}`
NeuralSQL: `{{10th query}}`

Parse the question into NeuralSQL.
Q: `{{target question}}`
NeuralSQL:

### E.3.2 EHRXQA: Experimental results

Table E21: Performance of ChatGPT + M$^3$AE (BM25 (train)) on the **EHRXQA** dataset, categorized by modality-based and patient-based scope.

| Modality-based | Patient-based | | $Acc_{LF}$ | $Acc_{EX|gt}$ | $Acc_{EX|pred}$ |
|---|---|---|---|---|---|
| Image | single | 1-image | 94.4 | 57.6 | 56.3 |
| | | 2-image | 73.3 | 51.1 | 50.0 |
| | | N-image | 90.8 | 39.6 | 39.6 |
| | group | | 85.0 | 5.0 | 1.7 |
| Table | none | | 98.0 | 100.0 | 98.0 |
| | single | | 87.1 | 100.0 | 95.6 |
| | group | | 44.7 | 100.0 | 87.5 |
| Image + Table | single | | 70.5 | 78.3 | 75.2 |
| | group | | 83.6 | 15.0 | 13.1 |

Table E22: Performance of ChatGPT + M$^3$AE (Fixed) on the **EHRXQA** dataset, categorized by modality-based and patient-based scope.

| Modality-based | Patient-based | | $Acc_{LF}$ | $Acc_{EX|gt}$ | $Acc_{EX|pred}$ |
|---|---|---|---|---|---|
| Image | single | 1-image | 1.4 | 57.6 | 27.7 |
| | | 2-image | 0.2 | 51.1 | 10.3 |
| | | N-image | 0.0 | 39.6 | 3.8 |
| | group | | 5.0 | 5.0 | 0.8 |
| Table | none | | 0.0 | 100.0 | 6.0 |
| | single | | 5.9 | 100.0 | 33.6 |
| | group | | 3.4 | 100.0 | 25.4 |
| Image + Table | single | | 3.3 | 78.3 | 40.8 |
| | group | | 13.1 | 15.0 | 6.5 |

# F Author statement

The authors of this paper bear all responsibility in case of violation of rights, etc. associated with the MIMIC-CXR-VQA and EHRXQA dataset.