# OpenReview forum: "EHRXQA: A Multi-Modal Question Answering Dataset for Electronic Health Records with Chest X-ray Images"
_NeurIPS.cc/2023/Track/Datasets_and_Benchmarks — NeurIPS 2023 Datasets and Benchmarks Poster_

### Official Review · Reviewer_cGyp · 2023-07-10
**A paper with both clinical and technical significance.**

**Rating:** 9
**Confidence:** 3
**Correctness:** The description of the dataset is acc…
**Clarity:** The manuscript is generally clear.

**Strengths:**

* They have constructed a QA dataset based on multimodal data combining table data and image data.
* The methodology for constructing the dataset is clearly stated, and is highly useful to the research community.
* They propose a NeuralSQL-based strategy using LLM and demonstrate its usefulness through comparison with existing methods.
* A methodology is presented that can flexibly and robustly answer a variety of questions, and it seems applicable and assured even in the era of LLM.

**Additional Feedback:**

There is no additional feedback.

**Documentation:**

The content of the dataset is detailed.

**Ethics:**

As it is built upon existing public datasets, ethical issues are minor.

**Limitations:**

There are no particularly serious limitations to point out.

**Opportunities For Improvement:**

* Since the approach relies on existing datasets, ensuring generality may be a challenge, as mentioned in the discussion section.

**Relation To Prior Work:**

The relation to prior research is also well described.

**Summary And Contributions:**

In this paper, the authors constructed a QA dataset for EHRs by combining structured EHR and chest X-ray datasets. Furthermore, to demonstrate its usefulness, they showed the effectiveness of an algorithm based on a NeuralSQL-based strategy.

---

> ### Author Response · Authors · 2023-08-21
> **Response to Reviewer cGyp**
>
> Thank you for your appreciation and valuable evaluation of our work.
>
> **Q1: Since the approach relies on existing datasets, ensuring generality may be a challenge, as mentioned in the discussion section.**
>
> A1: Thank you for sharing our view as described in the discussion section. We view this research as a stepping stone toward developing multimodal QA in medical records. We hope more institutions will decide to share their multi-modal EHR datasets in the future, as this will enable external validation and enhance the applicability of our work on a broader scale.

---

> > ### Comment · Reviewer_cGyp · 2023-08-28
> > **Response to authors**
> >
> > Thank you for your thoughtful remarks.

---

### Official Review · Reviewer_NgV3 · 2023-07-22
**The paper effectively introduces EHRXQA, a multimodal question answering dataset for electronic health records with chest X-ray images, but concerns arise regarding the integration of patient data, clarity in methodology, the scalability of SQL queries, and potential redundancies in using external APIs.**

**Rating:** 7
**Confidence:** 3

**Strengths:**

The paper is well-written and provides a detailed description of dataset generation and the proposed approach. The contribution of the paper is that the dataset and approach can enable joint reasoning over imaging and table modalities in EHR QA systems, which is currently underexplored.

**Additional Feedback:**

N/A

**Clarity:**

In addition, in Figure 2 stage 3 Time template, it’s not clear how to handle the three primary timeline scenarios. Is the time scenario handled in SQL table subquery or VQA API where clause? And what is [Time filter global1] mean? Not find clear explanation in the paper.

**Correctness:**

Yes. The submission is a dataset and the paper structured correctly by introducing the data preprocessing, categorization and generation methods.

**Documentation:**

The data pipeline is clear.

**Limitations:**

Use SQL query table might lead to performance issues for large datasets. As the dataset size increases, executing complex SQL queries on large tables can lead to performance bottlenecks. And SQL query might not the appropriate choice for QA task for better question sentence semantic understanding in the real-world scenario.

**Opportunities For Improvement:**

The image dataset could include text or table information to record patient visit information. The method of combining the dataset Chest Imagenome with external VQA API might be redundant. It might be possible to directly retrieve information from clinical notes about the corresponding image.

**Relation To Prior Work:**

Yes, this work is the first one to tackle the unique challenge of handling multi-modal questions within EHRs.

**Summary And Contributions:**

The paper presents a novel multi-modal question answering dataset called EHRXQA for electronic health records (EHRs) with chest X-ray images. The authors combine two uni-modal resources, MIMIC-CXR-VQA and EHRSQL (MIMIC-IV), to construct their multimodal EHR QA dataset.The paper proposes a NeuralSQL-based approach with external VQA API to handle multi-modal questions within EHRs.

---

> ### Author Response · Authors · 2023-08-21
> **Response to Reviewer NgV3 (1/3)**
>
> We are very grateful for your insightful comments. We address your comments by following Q-A pairs (from **Q1** to **Q6**):
>
> **Q1: The image dataset could include text or table information to record patient visit information. The method of combining the dataset Chest Imagenome with external VQA API might be redundant. It might be possible to directly retrieve information from clinical notes about the corresponding image.**
>
> A1: Thank you for the insightful review and your thought-provoking question regarding whether image information is necessary when corresponding textual information might exist. We appreciate that you have identified an aspect we have also considered.
>
> First, you are correct that textual information (e.g., “anatomical findings” section in the radiology report) indeed holds valuable image-related information. However, our investigation has found that textual information cannot entirely substitute the images for several reasons: 1) In describing original images into a report, not all information may be captured, or it may be too briefly summarized, e.g., “normality of a region”; 2) There may be clinical factors present in the image but absent from the accompanying report such as numeric information, e.g., “mediastinal-to-thoracic width ratio (MTR) and cardiothoracic ratio (CTR)”; 3) Images provide opportunities for additional spatial interpretation such as pixel heatmap visualization, which is challenging with text-based information alone.
>
> We believe that in an ideal QA scenario, we would utilize both image and text information complementarily as the context (not using only one modality). But at this stage, our focus is on exploring the image-table modality for EHR QA, and we are not including the text modality (e.g., clinical reports, radiology reports) in our current research scope. This integration (i.e., triple modality) remains an exciting avenue for future work, one we eagerly anticipate exploring.
>
> **Q2: Use SQL query table might lead to performance issues for large datasets. As the dataset size increases, executing complex SQL queries on large tables can lead to performance bottlenecks.**
>
> A2: We recognize the challenge of performance bottlenecks when executing complex SQL queries on large datasets. To address this issue, we have adopted strategies from our previous work on EHRSQL, specifically favoring nesting (using subqueries) over joining. This design decision aims to improve speed (Please refer to Appendix C.3 in the EHRSQL paper for details), and we applied this method to EHRXQA to mitigate the performance issues.
>
> Furthermore, in reality, SQL is still one of the most efficient ways to retrieve information from a large relational database. There are alternatives such as: 1) encoding raw tables with BERT-variants and retrieving the saved embeddings during inference; 2) Having LLMs read the tables. However, both alternatives have limitations that make them unsuitable for retrieving information from complex medical records. With alternative 1, it is practically impossible to execute complex queries that involve multiple nested queries and logical operations. With alternative 2, splitting large tables and having an LLM read all sub-tables will take even longer than executing SQL queries. Although there are other non-RDB options such as graph DBs or document-oriented DBs, it is beyond the scope of this work to analyze the pros and cons of storing EHR data in such DBs in terms of supported query functions, scalability, and execution latency.

---

> > ### Author Response · Authors · 2023-08-21
> > **Response to Reviewer NgV3 (2/3)**
> >
> > **Q3: And SQL query might not the appropriate choice for QA task for better question sentence semantic understanding in the real-world scenario.**
> >
> > A3: Since EHRs are commonly stored in relational databases with multiple tables, it is worth noting that SQL has been an appropriate choice for retrieving structured information from EHRs, as supported by several prior works [1-5].
> >
> > However, as you have rightly pointed out, we agree that typical SQL might not be a universal approach for all real-world EHR QA scenarios [6], such as when multi-modal QA is required over tables and images.
> >
> > For this reason, we have proposed the NeuralSQL approach in our work to handle these diverse information-seeking needs for real-world EHR QA scenarios. Our approach serves as a more flexible tool by leveraging the strengths of SQL for retrieving structured information from EHR tables, while also incorporating a visual module for handling image data. We believe that exploring such a versatile QA approach that can cover diverse question semantics in real-world scenarios is indeed a promising future direction.
> >
> > In response to your insightful observation and to further clarify the rationale behind our choice of SQL and the NeuralSQL approach, we have added Section 4.1.3 (”SQL/NeuralSQL Annotation”) in the revised manuscript.
> >
> > References:
> >
> > - [1] Wang, Ping, Tian Shi, and Chandan K. Reddy. "Text-to-SQL generation for question answering on electronic medical records." Proceedings of The Web Conference 2020.
> > - [2] Yu, Xiaojing, et al. "Dataset and enhanced model for eligibility criteria-to-SQL semantic parsing." *12th International Conference on Language Resources and Evaluation (LREC)*. 2020.
> > - [3] Raghavan, Preethi, et al. "emrkbqa: A clinical knowledge-base question answering dataset." Association for Computational Linguistics, 2021.
> > - [4] Bardhan, Jayetri, et al. "Drugehrqa: A question answering dataset on structured and unstructured electronic health records for medicine related queries." *arXiv preprint arXiv:2205.01290* (2022).
> > - [5] Lee, Gyubok, et al. "EHRSQL: A Practical Text-to-SQL Benchmark for Electronic Health Records." *Advances in Neural Information Processing Systems* 35 (2022): 15589-15601.
> > - [6] Kim, Daeyoung, et al. "Uncertainty-aware text-to-program for question answering on structured electronic health records." *Conference on Health, Inference, and Learning*. PMLR, 2022.
> >
> > **Q4: In addition, in Figure 2 stage 3 Time template, it’s not clear how to handle the three primary timeline scenarios.**
> >
> > A4: We appreciate your feedback and understand your concerns. It seems that there is some confusion about which sampling stage in the QA generation process, as illustrated in Figure 2, determines the type of the three primary timeline scenarios (described in the “Image+Table-related” paragraph of Section 4.1.2).
> >
> > To clarify, which timeline scenario to use for generating a question is decided in Stage 0, where the question template selection takes place. Once a template is selected, specific time conditions within that template are further concretized in Stage 3, the time template sampling process.
> >
> > For example, during Stage 0, the following template might be chosen: "Has_verb patient {patient_id} had a chest x-ray study indicating \\${attribute} in the \\${object} [time_filter_within] after being diagnosed with {diagnosis_name} [time_filter_global1]?" This selected template corresponds to the second timeline scenario, in which a CXR event follows a table event. In Stage 3, we concretize time conditions by sampling the appropriate time filters, such as [time_filter_within] to indicate "within the same hospital visit" and [time_filter_global1] to signify "in the last year."

---

> > > ### Author Response · Authors · 2023-08-21
> > > **Response to Reviewer NgV3 (3/3)**
> > >
> > > **Q5: Is the time scenario handled in SQL table subquery or VQA API where clause?**
> > >
> > > A5: The three primary timeline scenarios (described in the "Image+Table-related" paragraph of Section 4.1.2 (”Question Template Construction”)) are handled in SQL (sub-)queries, but not in the VQA API's WHERE clause. To be more specific, we explain each scenario as follows:
> > >
> > > - First Scenario (A table event and a CXR event co-occurs): Both the table and CXR events must occur within the same time frame. Therefore, each subquery (which represents a table/CXR event) includes the same SQL WHERE condition (e.g., `WHERE admissions.dischtime IS NOT NULL LIMIT 1`) to ensure that the two events co-occur within this time frame (e.g., "during the last hospital visit").
> > > - Second Scenario (A CXR event occurs after a table event): This scenario is determined in the SQL WHERE condition after joining two event tables. For example, `WHERE T1.charttime < T3.studydatetime` (where T1 represents a table (diagnoses) event, and T3 represents a CXR event).
> > > - Third Scenario (A table event follows a CXR event): This is handled similarly to the second scenario but the order is reversed.
> > >
> > > As outlined in Section 5 (”NeuralSQL with Visual Question Answering”), it's important to note that the current VQA API only deals with image sets $I$. Thus, the VQA API's WHERE clause (e.g., `WHERE FUNC_VQA(..., T1.study_id) = True`) is not explicitly related to handling the time scenarios.
> > >
> > > **Q6: And what is [Time filter global1] mean? Not find clear explanation in the paper.**
> > >
> > > A6: Thank you for pointing this out. We will clarify the time filters (e.g., [time_filter_global1]) by adding the following explanation in Section 4.1.2 (”Question Template Construction”) of the revised manuscript:
> > >
> > > Recognizing the critical role of time expressions in real-world questions in the hospital workplace, we further refined our question templates. We adopted the time filter concept from EHRSQL and applied it to all question templates.This enhancement allows our question templates to better meet the specific needs in clinical practice. Note that these time filters can be categorized into three types:
> > > 1) [time_filter_global] restricts the time range of interest, such as "last year" or "in 2022";
> > > 2) [time_filter_within], incorporating the keyword "within", pinpoints events happening within specific temporal boundaries, such as "within the same hospital visit" or "within the same day".
> > > 3) [time_filter_exact] refers to a precise temporal point, such as the "last CXR study" or a specific date and time like "2105-12-26 15:00:00".

---

> ### Author Response · Authors · 2023-08-28
> **Sincerely expecting further discussions with Reviewer NgV3**
>
> Dear Reviewer NgV3,
>
> We deeply appreciate the time and effort you have dedicated to reviewing our paper.
> As we are nearing the end of the discussion period, we would be grateful if you could let us know if our response addressed your concerns or if there are any further questions or feedback.

---

> > ### Comment · Reviewer_NgV3 · 2023-08-28
> >
> > Hi authors,
> >
> > I appreciate your detailed and completed statement that deals with my concerns. I have no further questions at all.

---

### Official Review · Reviewer_obQA · 2023-07-24
**Nice contribution to the EHR QA space**

**Rating:** 7
**Confidence:** 4

**Strengths:**

- Multimodal QA over structured and pixel data is a nice dataset contribution and unmet research need.
- The dataset is a nice synthesis of prior work on MIMIC CXR data

**Additional Feedback:**

- NA

**Clarity:**

- I was somewhat initially confused by the distinction between EHRXQA and MIMIC-CXR-VQA, since both involve image and table questions. Could the authors elaborate on the distinction between the two and the significant differences in their contributions?
- I think the presentation of the paper can be improved. Even a table breaking down the unimodal datasets by name, their included modalities and their remapping to new datasets would help anchor readers.

**Correctness:**

- The manscript does not have any clear errors

**Documentation:**

- Documentation is sufficient (most of the data is via PhysioNet)

**Ethics:**

- No ethics problems

**Limitations:**

- The dataset is largely a refactoring of multiple existing datasets (including some enriched templates to capture information contained in  Chest ImaGenome, etc).
- Its unclear how well these templated questions actually align with the types of information needs for EHR-based datasets.

**Opportunities For Improvement:**

- Questions generated from slot-filled templates tend to be easier for a model to learn and lack heterogeneity. https://aclanthology.org/2020.acl-main.410/ for example found only 5% of emrQA was needed to train a model. It would be nice to do an ablation experiment on the degree more templates are actually improving the model.
- For MIMIC-CXR-VQA, the task is formulated as a multi-label classification with 110 disjoint answers. It would be nice to see a more difficult baseline using an open world, generative assumption for question answering.
- More exploration of the GPT-4 based paraphrasing. This is mentioned at a few points (line 171, 246) and Appendix B.2.2, C.1.3. Given the brittleness of templates in prior EHR QA work, ablations around the interplay and benefits of more templates + diverse templates via LLM paraphrases would be a nice addition to the manuscript.

**Relation To Prior Work:**

- Prior work is nicely discussed (and in this case integral to synthesizing their new dataset).

**Summary And Contributions:**

This manuscript introduces EHRXQA, a multimodal QA dataset for structured EHRs and x-ray imaging. This dataset meets an unmet need for a multimodal (image/EHR) QA dataset by constructing 2 unimodal datasets (1) MIMIC-CXR-VQA (chest x-rays) and (2) EHRSQL. Both of these unimodal datasets involve some degree of refactoring and remixing to create EHRXQA. Combining these datasets to create a multimodal EHR QA dataset. The manuscript benchmarks five VQA baselines on the MIMIC-CXR-VQA, finding M3AE performs best. For EHRXQA they conduct experiments broken down by modality question (image/table/both).

---

> ### Author Response · Authors · 2023-08-21
> **Response to Reviewer obQA (1/5)**
>
> We are grateful for your valuable feedback and suggestions. We address your comments by following Q-A pairs (from **Q1** to **Q7**):
>
> **Q1: Questions generated from slot-filled templates tend to be easier for a model to learn and lack heterogeneity. https://aclanthology.org/2020.acl-main.410/ for example found only 5% of emrQA was needed to train a model. It would be nice to do an ablation experiment on the degree more templates are actually improving the model.**
>
> A1: Thank you for highlighting the significance of exploring redundancy in questions generated from slot-filled templates. Given that the large sample size of emrQA (medication=220K, relation=900K) led to the redundancy issue, we decided to investigate the MIMIC-CXR-VQA dataset specifically, as it has a larger dataset size (330K), making it more suitable for this analysis. Following your suggestion, we explored the degree to which additional templates actually improve the model (please see *Study 1*). We also carried out an in-depth analysis of the datasets to further investigate the underlying factors (please see *Study 2*). Our careful consideration of diverse templates and contexts in the MIMIC-CXR-VQA dataset indeed contributed to improved model performance, addressing the concerns you raised about redundancy and lack of heterogeneity.
>
> ---
>
> *Study 1 - Ablation Experiment*
>
> We conducted an ablation experiment with the MIMIC-CXR-VQA dataset to evaluate the test set performance by randomly sampling training data at various template usage proportions (i.e. how many unique templates are used for training), such as 5%, 10%, 20%, 50%, and 100%.
>
> |  | PubmedCLIP* | MedViLL* | M3AE* |
> | --- | --- | --- | --- |
> | template usage (%) | test (Acc/F1) | test (Acc/F1) | test (Acc/F1) |
> | 5% | 50.4 / 0.43 | 54.6 / 0.54 | 60.1 / 0.62 |
> | 10% | 50.5 / 0.43 | 57.3 / 0.58 | 63.2 / 0.66 |
> | 20% | 54.1 / 0.54 | 61.3 / 0.63 | 65.5 / 0.69 |
> | 50% | 57.1 / 0.56 | 63.3 / 0.67 | 68.8 / 0.72 |
> | 100% | 58.5 / 0.60 | 66.0 / 0.68 | 70.4 / 0.74 |
>
> Unlike the emrQA dataset, where 5% of the dataset yields the same test performance as 100% for emrQA-relation, our experiment with MIMIC-CXR-VQA demonstrated that using questions generated from more diverse templates positively impacts the model's performance (i.e., test Acc/F1) across all models. This observation suggests that MIMIC-CXR-VQA does not have the redundancy of questions observed in emrQA, and indeed, question diversity generated from all templates contributes to the performance improvement.
>
> ---
>
> *Study 2 - Dataset Analysis*
>
> We performed an in-depth comparison of the MIMIC-CXR-VQA and emrQA datasets to discern the underlying factors that might have caused the differences observed in *Study 1*. This table includes an analysis of Questions, Templates, and Contexts from both MIMIC-CXR-VQA and emrQA. Note: "# Question / Context" denotes how many questions are used to train with the identical context, and "# Question / Template" illustrates how many questions are sampled from the same template.
>
> |  | emr-QA; Medication | emr-QA; Relation | MIMIC-CXR-VQA |
> | --- | --- | --- | --- |
> | # Question | 222,957 | 904,592 | 377,726 |
> | # Context | 261 | 423 | 142,637 |
> | # Template (seed) | 15 | 19 | 48 |
> | # Template (after paraphrasing) | 80 | 139 | 1,363 |
> | # Question/Context | 854.24 | 2138.52 | 2.65 |
> | # Question/Template | 2786.96 | 6507.86 | 277.09 |
>
> Our analysis revealed that MIMIC-CXR-VQA uses a greater diversity of contexts (142,637) and templates (48 seed templates expanding to 1,363 after paraphrasing). Furthermore, the ratio of questions to contexts and templates in MIMIC-CXR-VQA (2.65 Questions/Context and 277.09 Questions/Template) is substantially lower than those in emrQA (854.24 and 2138.52 Questions/Context; 2786.96 and 6507.86 Questions/Template). These numbers illustrate that the MIMIC-CXR-VQA dataset's questions are not as redundant in relation to context and template, compared to emrQA.
>
> Reference:
>
> [1] Yue, Xiang, Bernal Jimenez Gutierrez, and Huan Sun. "Clinical reading comprehension: a thorough analysis of the emrQA dataset." *arXiv preprint arXiv:2005.00574* (2020).

---

> > ### Author Response · Authors · 2023-08-21
> > **Response to Reviewer obQA (2/5)**
> >
> > **Q2: For MIMIC-CXR-VQA, the task is formulated as a multi-label classification with 110 disjoint answers. It would be nice to see a more difficult baseline using an open world, generative assumption for question answering.**
> >
> > A2: Thank you for the suggestion. We have extended our experiment to include a more challenging baseline using an open-world, generative approach for question answering with MIMIC-CXR-VQA.
> >
> > Our experiment involves two distinct and very recent open-sourced baselines, aiming to compare both a general-domain multi-modal generative model, Open-Flamingo [2], and a medical domain multi-modal generative model, Med-Flamingo [3]. Both of these models are large-scale vision-language models, serving as multimodal few-shot learners with interleaved image-text sequences.
> >
> > Since the results from open-world generative models can be noisy and brittle when evaluated on our dataset compared to the classification setting, we conducted the baseline experiment on the part of our MIMIC-CXR-VQA dataset that includes verifying questions (i.e., "yes/no" questions), which constitute about 50% of our dataset. This allows us to more accurately evaluate the model's understanding.
> >
> > In the experiment setting, we use pre-trained 3B size for both models and for the few-shot setting, we use 6 static (medical images, question, answer) pairs, as this model (i.e., Flamingo [1]) family can boost up the performance using its in-context learning ability. As shown in the table, the current medical generative model is unable to achieve even 50% accuracy (i.e., equivalent to random guessing for “yes/no” questions). We anticipate that future work could explore further training, instruction, or preference tuning to attain the level of performance comparable to our best-performing model, M3AE.
> >
> > | model (open world, generative) | test - verify (Acc) | model (classification) | test - verify (Acc) |
> > | --- | --- | --- | --- |
> > | Open-Flamingo (zero shot) | 45.04 | PubMedCLIP* (fine-tuning) | 71.2 |
> > | Open-Flamingo (few shot) | 45.63 | MedViLL* (fine-tuning) | 77.3 |
> > | Med-Flamingo (zero shot) | 43.51 | M3AE* (fine-tuning) | 80.8 |
> > | Med-Flamingo (few shot) | 49.61 |  |  |
> >
> > References:
> > - [1] Alayrac, Jean-Baptiste, et al. "Flamingo: a visual language model for few-shot learning." Advances in Neural Information Processing Systems 35 (2022): 23716-23736.
> > - [2] Awadalla, Anas, et al. "OpenFlamingo: An Open-Source Framework for Training Large Autoregressive Vision-Language Models." arXiv preprint arXiv:2308.01390 (2023).
> > - [3] Moor, Michael, et al. "Med-Flamingo: a Multimodal Medical Few-shot Learner." arXiv preprint arXiv:2307.15189 (2023).

---

> > > ### Author Response · Authors · 2023-08-21
> > > **Response to Reviewer obQA (3/5)**
> > >
> > > **Q3: More exploration of the GPT-4 based paraphrasing. This is mentioned at a few points (line 171, 246) and Appendix B.2.2, C.1.3. Given the brittleness of templates in prior EHR QA work, ablations around the interplay and benefits of more templates + diverse templates via LLM paraphrases would be a nice addition to the manuscript.**
> > >
> > > A3: Thank you for emphasizing the need to explore GPT-4-based paraphrasing in LLM. We interpreted your comment as a suggestion to investigate how variations in template diversity, specifically through varying the number of paraphrases, can impact the model's overall performance. Therefore, we conducted an experiment to explore the correlation between the variety of paraphrases created using GPT-4 and model performance, keeping the dataset size constant and varying the number of paraphrases per template for the training dataset (please see Study 3). As you suggested, we have added the ablation study experiments exploring data redundancy (*Study 1* in A1) and the impact of paraphrasing (*Study 3* in A3) in Section E.2 (”MIMIC-CXR-VQA: Exploring data redundancy and the impact of paraphrasing”) of the supplementary materials in the revised manuscript.
> > >
> > > ---
> > >
> > > *Study 3 - Ablation Experiment*
> > >
> > > In the MIMIC-CXR-VQA dataset, we started with 48 seed templates as the training dataset (denoted as "seed template") and created three training dataset variants with different degrees of paraphrasing diversity, namely "para-small", "para-moderate", and "para-large". These variants contained 3.9, 10.4, and 21.5 paraphrased templates per original template, respectively, while keeping the dataset size constant. Note that each model, trained with four different datasets, was evaluated against the same original MIMIC-CXR-VQA test dataset. We ran the experiments with three different seeds.
> > >
> > > |  | PubMedCLIP* | MedViLL* | M3AE* |
> > > | --- | --- | --- | --- |
> > > | Training Dataset | test  (Acc/F1) | test  (Acc/F1) | test  (Acc/F1) |
> > > | MIMIC-CXR-VQA (seed template) | 43.6 ± 1.7 / 0.38 ± 0.0 | 47.3 ± 1.8 / 0.50 ± 0.0 | 67.1 ± 1.4 / 0.70 ± 0.0 |
> > > | MIMIC-CXR-VQA (para-small) | 58.5 ± 0.2 / 0.60 ± 0.0 | 65.0 ± 0.8 / 0.68 ± 0.0 | 69.5 ± 0.1 / 0.73 ± 0.0 |
> > > | MIMIC-CXR-VQA (para-moderate) | 58.4 ± 0.5 / 0.60 ± 0.0 | 65.0 ± 0.2 / 0.68 ± 0.0 | 69.6 ± 0.1 / 0.73 ± 0.0 |
> > > | MIMIC-CXR-VQA (para-large) | 58.3 ± 0.3 / 0.59 ± 0.0 | 65.9 ± 0.2 / 0.69 ± 0.0 | 70.1 ± 0.3 / 0.73 ± 0.0 |
> > >
> > > Our findings indicate that using paraphrased templates through GPT-4 improves model performance, as evidenced when comparing the first result (seed template) to the other three (para-small, para-moderate, and para-large). Further, the model performance increases with the diversity of the paraphrase. However, the impact of paraphrasing can vary depending on the text encoder employed. Models that utilize a BERT-based language model, such as MedViLL and M3AE, benefit more from this increased diversity. These results show that even templates generated through automatic paraphrasing can significantly enhance performance, provided sufficient diversity.

---

> > > > ### Author Response · Authors · 2023-08-21
> > > > **Response to Reviewer obQA (4/5)**
> > > >
> > > > **Q4: The dataset is largely a refactoring of multiple existing datasets (including some enriched templates to capture information contained in Chest ImaGenome, etc).**
> > > >
> > > > A4: While it may appear that our EHRXQA dataset is a simple recompilation of existing datasets, this characterization does not fully capture the essence of our work. Like EHRSQL, which utilizes MIMIC-III as components and focuses its contribution on crafting question templates for retrieving structured information from EHRs, we invested significant effort into creating complex and diverse question templates to reflect the information-seeking needs across image and table modalities within EHRs. This included exploring clinical scenarios involving CXR images and EHR tables and ensuring clinical relevance through evaluation and validation with a medical expert. Therefore, even though our QA dataset is based on existing datasets, it is hardly fair to say it is just a refactoring of multiple existing datasets. Please refer to response A5 for further elaboration on the alignment of these templated questions with the information-seeking needs for EHR-based datasets.
> > > >
> > > > **Q5: Its unclear how well these templated questions actually align with the types of information needs for EHR-based datasets.**
> > > >
> > > > A5: To align our templated questions with real-world information-seeking needs, we crafted the EHRXQA question templates with the following process: Our process included (1) exploring actual clinical needs across image and table modality via consulting a medical expert, (2) grounding our templates in these needs for both CXR images and EHR tables, and (3) ensuring clinical relevance through expert validation.
> > > >
> > > > - For the image modality, we discovered that existing VQA question scenarios (e.g., ”Is there pneumonia?”) often lacked usability and were too simplified for EHR contexts. We identified many potential question scenarios within EHRs (e.g., ”For patient 42, Is there pneumonia in the last hospital visit?”) that could provide valuable insights into visual diagnoses. To encapsulate these complex scenarios, we considered various types of inquiries, including single patient inquiries, consecutive comparisons, and imaging analysis across multiple CXR images. We grounded these needs in our 168 question templates in EHRXQA, marking a departure from traditional single-image medical VQA settings.
> > > > - For the table modality, our 174 question templates do align well with real-world information-seeking needs. These templates are rooted in the EHRSQL answerable question templates, reflecting actual information-seeking needs gathered by conducting a survey on a diverse pool of hospital staff.
> > > > - In considering both CXR images and structured EHRs, we found that there are multi-modal information-seeking needs across images and tables. After consulting with a medical expert, we unearthed new potential scenarios that cover three timeline scenarios (i.e., after, before, co-occurring between two modalities) and cohort analysis (i.e., combining demographic factors such as gender and age with CXR findings).
> > > >
> > > > While our question templates may not cover all possible information-seeking needs, we have invested significant effort into aligning them with clinical relevance. We hope our EHRXQA dataset will serve as a stepping stone for uncovering more diverse and complex multi-modal information-seeking needs within EHRs in the future. To further clarify and enhance the description of our template construction process, we have revised a few paragraphs in Section 4.1.2 (”Question Template Construction”) of our manuscript.

---

> > > > > ### Author Response · Authors · 2023-08-21
> > > > > **Response to Reviewer obQA (5/5)**
> > > > >
> > > > > **Q6: I was somewhat initially confused by the distinction between EHRXQA and MIMIC-CXR-VQA, since both involve image and table questions. Could the authors elaborate on the distinction between the two and the significant differences in their contributions?**
> > > > >
> > > > > A6: To elaborate, we summarized the key distinctions (please see *A.*) between EHRXQA and MIMIC-CXR-VQA in the table below. Also, we present the significant difference (please see *B.*) in their contributions. To further aid in understanding the contributions of our datasets, we have added Section 4.3 ("Comparisons with other EHR QA datasets") in the revised manuscript.
> > > > >
> > > > > ---
> > > > >
> > > > > *A. Distinction Between the Two Datasets*
> > > > >
> > > > > | Features | EHRXQA | MIMIC-CXR-VQA |
> > > > > | --- | --- | --- |
> > > > > | Setting (Input → Output) | Database (Images, Tables), Question → Answer | Single image, Question → Answer |
> > > > > | Modality | Image, Table, Image + Table | Image |
> > > > > | Time Expression | Present | Absent |
> > > > > | Example | “After being prescribed nitroglycerin, did patient 568 undergo a CXR during the same visit that revealed any abnormalities in the cardiac silhouette?” | “Is there any abnormality in the cardiac silhouette?” |
> > > > >
> > > > > The term "time expression" denotes time-related expressions that could indicate specific events within a particular timeframe (e.g., last year, current hospital visit). For detailed examples from each dataset, please refer to Table 1 of the main paper (EHRXQA) and Table B4 in the supplementary section (MIMIC-CXR-VQA). If this does not fully address your concern, could you please specify the part of the paper that made you think the MIMIC-CXR-VQA dataset includes both image and table questions? Your feedback will help us clarify and improve our text.
> > > > >
> > > > > ---
> > > > >
> > > > > *B. Differences in Contributions*
> > > > >
> > > > > EHRXQA and MIMIC-CXR-VQA are both EHR QA datasets, but they offer contributions from distinct perspectives.
> > > > >
> > > > > - MIMIC-CXR-VQA is a new image-based EHR QA dataset that can be utilized as a medical VQA benchmark dataset. The contribution of our dataset comes from the diversity and complexity of its question templates compared to other medical VQA datasets (Please see Table B8 of the supplementary). Specifically, our templates involve compositional templates formulated through set/logical operations, which are not typically found in other datasets.
> > > > > - EHRXQA is the first multi-modal EHR QA dataset with both image and table modalities. While existing EHR QA works (Please see Table C15 of the supplementary) mainly focus on questions regarding individual modalities, our dataset not only involves questions about each separate modality but also includes multi-modal questions combining image and table data. It provides more than just an aggregation of unimodal questions, including complex queries demanding cross-modal reasoning.
> > > > >
> > > > > &nbsp;
> > > > > &nbsp;
> > > > >
> > > > > **Q7: I think the presentation of the paper can be improved. Even a table breaking down the unimodal datasets by name, their included modalities and their remapping to new datasets would help anchor readers.**
> > > > >
> > > > > A7: Thank you for your suggestion. We consider the following table to show the relationship between datasets and modalities. If this table doesn't address your concern, please let us know if you have specific requests or if another format would be more helpful.
> > > > >
> > > > > | Dataset Name | Included Modality | Source Datasets |
> > > > > | --- | --- | --- |
> > > > > | MIMIC-CXR-VQA | Image | MIMIC-CXR, Chest ImaGenome |
> > > > > | EHRSQL | Table | MIMIC-IV |
> > > > > | EHRXQA | Image, Table | MIMIC-CXR-VQA, EHRSQL |
> > > > >
> > > > > Moreover, we will refine Figure 1 in our main paper to clearly illustrate the relationship among datasets. We aim to make the modalities of each dataset more evident in the figure and will adjust the caption to more explicitly depict the mapping process among the datasets.

---

> ### Author Response · Authors · 2023-08-28
> **Sincerely expecting further discussions with Reviewer obQA**
>
> Dear Reviewer obQA,
>
> Thank you for your careful review of our paper. Your insights have been invaluable.
> As the discussion period is nearing its end, we would like to inquire if our response addressed your concerns.
> Please let us know if you have any further questions or feedback.

---

> > ### Comment · Reviewer_obQA · 2023-08-29
> >
> > I thank the authors for their additional experiments and details in the rebuttal. This addresses my main questions. I have increased my score as a result.

---

### Decision · Program_Chairs · 2023-09-22

**Decision:**

Accept (Poster)

**Comment:**

The authors have provided detailed responses to address the reviewers' valuable feedback. I appreciate the authors' sincere efforts to improve their work.

A key question was around the need for images when textual reports may exist. The authors explained why images still provide complementary value - some aspects like spatial patterns are not captured in text. Their focus is on image-table QA, but integrating text in the future is promising.

Regarding SQL performance, the authors adopt optimization strategies like nesting to mitigate bottlenecks. The NeuralSQL approach also complements SQL to handle diverse real-world QA needs. The rationale for choosing SQL is further clarified.

Overall, the authors have satisfactorily responded to the reviewers' concerns around image relevance, SQL usage, timeline handling, and terminology. One reviewer also helpfully pointed out the need to ensure generalizability across institutions in the future.

I recommend accepting this paper based on its novelty as the first multimodal QA dataset combining images and structured EHR data. This can enable joint reasoning over the two modalities for tasks like clinical decision support. The proposed NeuralSQL approach also demonstrates a way to handle such diverse QA needs.